# Quantitative essentiality in a reduced genome: a functional, regulatory and structural fitness map

Samuel Miravet-Verde [ID] [1,4,7 ✉], Raul Burgos [ID] [1,5,7 ✉], Eva Garcia-Ramallo[1], Marc Weber[1,6] & Luis Serrano [ID] [1,2,3 ✉]

## Abstract

**Essentiality studies have traditionally focused on coding regions, often overlooking other small genetic regulatory elements. To address this, we combined transposon libraries containing promoter or terminator sequences to obtain a high-resolution essentiality map of a genome-reduced bacterium, at near-single-nucleotide precision when considering non-essential genes. By integrating temporal transposon-sequencing data by k-means unsupervised clustering, we present a novel essentiality assessment approach, providing dynamic and quantitative information on the fitness contribution of different genomic regions. We compared the insertion tolerance and persistence of the two engineered libraries, assessing the local impact of transcription and termination on cell fitness. Essentiality assessment at the local base-level revealed essential protein domains and small genomic regions that are either essential or inaccessible to transposon insertion. We also identified structural regions within essential genes that tolerate transposon disruptions, resulting in functionally split proteins. Overall, this study presents a nuanced view of gene essentiality, shifting from static and binary models to a more accurate perspective. Additionally, it provides valuable insights for genome engineering and enhances our understanding of the biology of genome-reduced cells.**

**Keywords** Essentiality; Transposon Sequencing; High-Resolution; Regulatory Elements; *Mycoplasma*
**Subject Categories** Biotechnology & Synthetic Biology; Computational Biology

## Introduction

Genes are the basic units that define the genome of an organism. Nonetheless, other genomic elements are also expected to contribute to cell function. These may include structural regions governing chromosome replication, or transcriptional and translational regulatory elements that allow coordination of gene activity. Notably, a comprehensive analysis of a genome-reduced organism like *Mycoplasma pneumoniae* has shown the importance of alternative regulatory mechanisms of transcriptional activity that do not depend on transcription factors (Yus et al, 2019), such as for example RNA degradation, the importance of the first nucleotide of the transcript, supercoiling, or the influence of chromosomal structure (Trussart et al, 2017). Hence, studies that aim to fully understand how genetic information impacts the fitness of an organism should consider other genomic features aside from genes.

Transposon mutagenesis is a powerful genetic tool to produce random mutations in genomes (Hamer et al, 2001). The appearance of ultra-deep sequencing technologies has allowed the screen of large pools of transposon mutants simultaneously in a given growth condition (Basta et al, 2017; Price et al, 2018; Salama et al, 2004; Christen et al, 2011; Langridge et al, 2009), opening the possibility to perform diverse genome-wide analyses of gene essentiality in a high-throughput manner (Barquist et al, 2013a). In a typical transposon-insertion sequencing (Tn-Seq) experiment, transformed cells with insertions in genes required for growth are lost after a short period of growth selection. This results in a transposon integration map where essential regions remain free of transposon insertions, allowing the identification of essential (E) and non-essential genes (NE). However, NE genes can be classified in subgroup categories depending on how the disruption of these genes causes a competitive defect under a defined set of conditions. These genes, typically known as fitness (F), may be classified as NE or E depending on the growth conditions and rounds of selection. In addition, essential genes may tolerate insertions in specific locations such as N- and C-terminal regions that generally do not form part of the functional unit (Miravet-Verde et al, 2020).

Traditionally, essentiality studies have focused on the gene level with few studies looking at the essentiality of other genomic features, including small genetic entities such as regulatory signals (including promoters and terminators) and non-transcribed regions of the chromosome (Mann et al, 2012; Capel et al, 2016; Barquist et al, 2013b; Zhang et al, 2012). This is due to the high transposon insertion density required to provide sufficient statistical evidence on the essentiality of small DNA segments (e.g., a promoter region). Ideally, this resolution

[1]Centre for Genomic Regulation (CRG), The Barcelona Institute of Science and Technology, Dr. Aiguader 88, Barcelona 08003, Spain. [2]Universitat Pompeu Fabra (UPF), Barcelona, Spain. [3]ICREA, Pg. Lluís Companys 23, Barcelona 08010, Spain. [4]Present address: Department of Biology, Institute of Microbiology and Swiss Institute of Bioinformatics, ETH Zurich, Zurich, Switzerland. [5]Present address: Pulmobiotics S.L., Dr Aiguader 88, 08003 Barcelona, Spain. [6]Present address: Flomics Biotech S.L., C/ Roc Boronat 31, 08005 Barcelona, Spain. [7]These authors contributed equally: Samuel Miravet-Verde, Raul Burgos. ✉E-mail: smiravet@ethz.ch; raul.burgos@crg.eu; luis.serrano@crg.eu

should approach one insertion per base, with careful consideration given to the design of the transposon used in the analysis. For example, mariner transposable elements depend on TA dinucleotide targets, limiting the resolution that can be obtained, especially in high GC-content genomes (Zhang et al, 2012). Additionally, the presence of promoter and terminator sequences within the transposon can bias the insertion preference depending on the transcriptional context. This can be particularly important when assessing the essentiality of large operons constituted by NE genes followed by E ones, or when gene overlaps happen. In this respect, approaches have been developed placing outward-facing promoters in the transposon vector to induce the expression of E genes and minimize polar effects when an operon is disrupted (Coe et al, 2019).

In this study, we leveraged the small genome of *M. pneumoniae* (816 kb), its high transformation efficiency, and the extensive available -omics datasets (Güell et al, 2009; Yus et al, 2009; Kühner et al, 2009; Maier et al, 2011, 2013; Chen et al, 2016; Lluch-Senar et al, 2013; Trussart et al, 2017; Burgos et al, 2020; Lluch-Senar et al, 2015), to define a comprehensive and high-resolution fitness map of its genome. We achieved this by designing and transforming the bacterium with two different Tn4001-based transposons, which insert randomly into the genome with only a slight TA dinucleotide preference that can be computationally corrected (Miravet-Verde et al, 2020). One design presents outward-facing promoters at both ends, designed to minimize polar effects and to explore the influence of transcriptional changes on fitness. The other transposon features rho-independent intrinsic terminators at each end, allowing us to investigate the impact of termination or reduced transcription in different genomic loci. By combining both datasets, we found 453,897 unique insertions covering ~55% of the entire genome, reaching a transposon insertion coverage close to absolute saturation for NE genes (92.4% average linear density (LD), calculated as number of insertions/gene length for each genomic locus; ~1 insertion per bp resolution). This resolution, along with the comparison of the two transposon libraries, enabled us to evaluate the significance of genomic regions beyond genes, such as regulatory signals and structural elements. It also revealed the influence of genome accessibility on transposon sequencing data. Additionally, we assigned essentiality at the protein domain level and identified, aside from NE termini extensions, regions within E genes that tolerate disruptions, leading to functional split proteins. Furthermore, we assessed the fitness impact of gene disruptions and transcriptional perturbations by tracking insertion decay after serial passaging selection. This approach allowed us to develop a novel methodology for assessing essentiality, using a k-means unsupervised clustering method that treats essentiality as a dynamic phenomenon.

Collectively, this study provides a resource for understanding the biology of a genome-reduced cell, offering insights into the minimal functions required to sustain life and their regulation, while also providing guidance for the design of proteins or entire biological systems for biotechnological applications.

## Results

### Design and generation of Tn-Seq libraries of *M. pneumoniae* at 1 bp resolution

We designed two transposon vectors based on the mini-transposon pMTnCat (Burgos and Totten, 2014), which contains a mobile *cat*

resistance marker enclosed by inverted repeats (IR), and a transposase derived from Tn4001 (Lyon et al, 1984). One vector (pMTnCat_BDPr) was engineered to contain two outward-facing promoters (P438) adjacent to the two IR (Fig. 1A). This configuration was implemented to minimize transcriptional polar effects on neighboring genes regardless of transposon orientation. The P438 promoter comprises 22 bases and includes two Pribnow boxes promoting constitutive and strong transcription of leaderless mRNAs (Pich et al, 2006b). Considering most mRNAs in *M. pneumoniae* lack Shine–Dalgarno sequences, translation initiation at the first methionine of the transcript seems the preferent mechanism for protein production (Nakagawa et al, 2010; Montero-Blay et al, 2019). Hence, this transposon design provides a system to evaluate the essentiality of potential ORF products as long as the P438 promoter inserts close to a translation initiation codon. A complementary vector (pMTnCat_BDter) containing outward-facing intrinsic rho-independent terminator sequences overlapping part of the two IR sequences was also constructed (Fig. 1A). In contrast to pMTnCat_BDPr, this genetic configuration aims to terminate or diminish transcription of any transcript originating from, or passing through, the transposon sequence once inserted in the genome. For this purpose, we used an endogenous rho-independent terminator (ter625) that silences the expression of the MPN626 sigma factor (Fig. EV1A). Termination activity of the ter625 sequence was assessed in two different genetic contexts using gene reporters expressed as polycistronic or fusion transcripts (Fig. EV1B). In both cases, we detected a reduction of expression of the reporter in the presence of ter625 as compared to a mutated version affecting the terminator hairpin structure. These results confirmed the functionality of the ter625 sequence.

Using the two mini-transposon vectors described above, we constructed two independent libraries of transposon mutants in the *M. pneumoniae* M129 strain ("P" for pMTnCat_BDPr, and "T" for pMTnCat_BDter). To eliminate dead cells that could interfere with the analysis and to define more precisely the contribution of NE genes in cell fitness, we completed up to ten consecutive passages equivalent to approximately ten cell divisions each (samples P1 or T1, to P10 or T10). Two biological replicates per sample, in addition to two technical replicates for passages from 2 to 4, were processed for next-generation sequencing, and insertion sites were then identified using FASTQINS (Miravet-Verde et al, 2020) (Dataset EV1). To ensure mapping accuracy, we used as a reference the genome sequence obtained from our laboratory strain, which was re-annotated based on available transcriptomic and proteomic data (see Methods; Dataset EV2). In terms of linear density (LD: number of insertions normalized by length of the region considered) and RPKM (reads mapping an insertion per kilobase per million reads), we observed a good correlation between these metrics measured at gene level for biological replicates across passages for both the P and T libraries ($R^2 \geq 0.87$ or $R^2 \geq 0.91$ in all conditions, respectively; Appendix Fig. S1). This shows no significant sampling batch effects.

At the genome-wide level, the P library showed an average LD in P1 of $37.1 \pm 8.1\%$ (Fig. 1B), or 46.3% when aggregating insertion events from the two independent P1 datasets (Dataset EV3). For the T1 library, the average LD at the genome level was slightly lower, $34.4 \pm 8.4\%$ and 44.2%, respectively. In total, we detected 285,015 insertions in common for both libraries, while 93,344 and 75,538 insertions were uniquely detected in the P and T libraries,

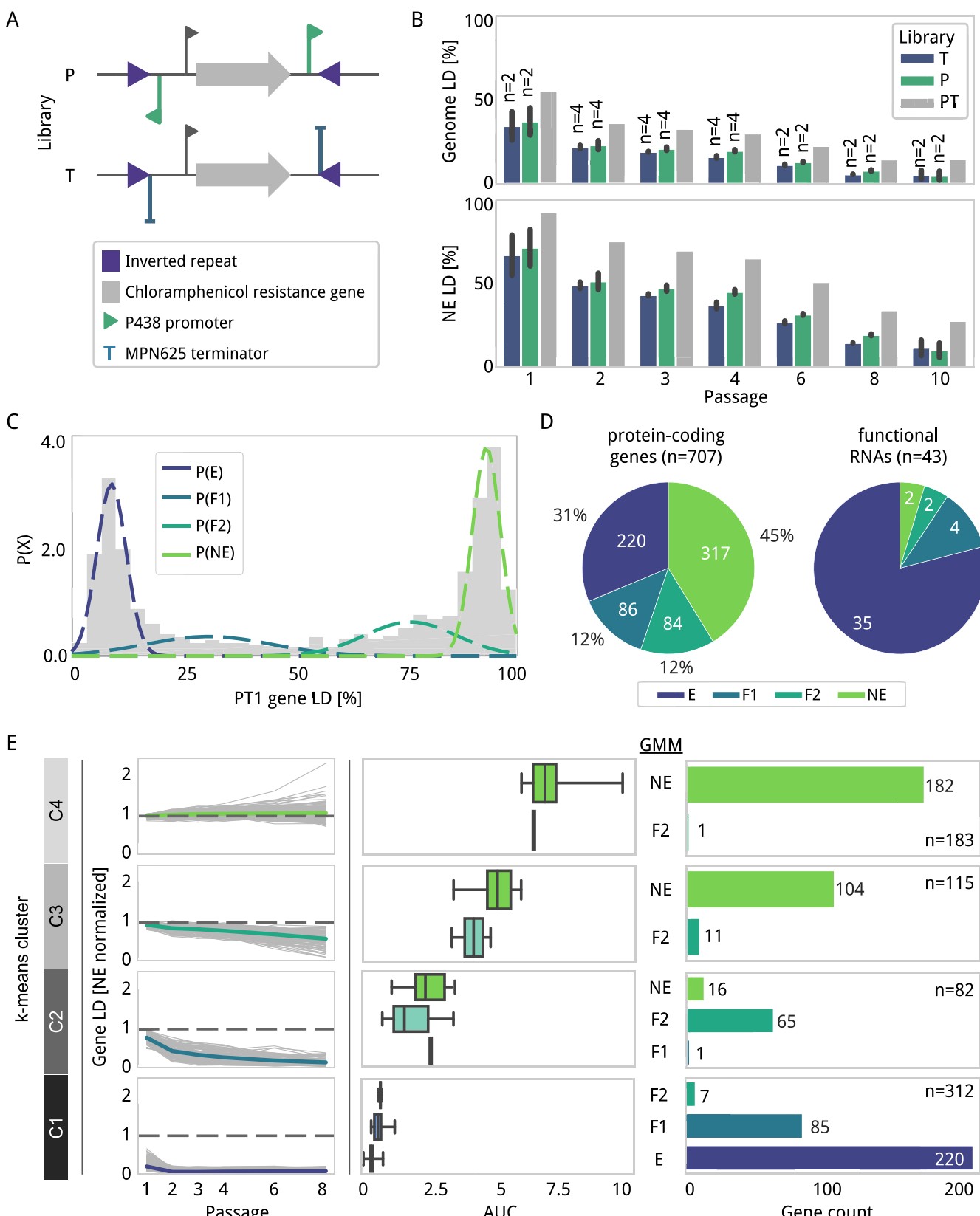

**Figure 1.  Transposon library design and gene essentiality analysis.**

(A) Schematic of the transposon mobile region in the two libraries designed here. P (top) containing outward-facing P438 promoters (green flags); and T (bottom) containing ter625 terminators (blue flags). Both constructs carry the chloramphenicol resistance gene as a marker (gray arrow) flanked by inverted repeat (IR) sequences (purple triangle). Notice that in the case of T libraries, the terminator partially overlaps with the IR sequence '*TACGGACTTTATC*'. (B) Bar plots comparing the transposon insertion coverage (Y-axis), as linear densities (LD; number of insertions normalized by the size of the genomic region considered), obtained along different serial passages (X-axis) for P (green), T (blue), and the combined P and T (PT) libraries (gray). The top plot shows the coverage of the full genome, whereas the bottom plot considers only known NE genes. Except for PT, which are the result of merging independent samples, the height of the bar represents the average obtained between replicates of the same condition. Error bars represent the standard deviation. The number of samples sequenced per condition is annotated on top of each bar. (C) Gaussian mixture model (GMM) of the distribution of LD values associated with annotated genes (X-axis) against the frequency (Y-axis) in PT1 samples (corresponding to passage 1). Dashed overlying lines represent the probability distribution for each essentiality category: E (dark blue), F1 (blue), F2 (turquoise), and NE (green). (D) Pie charts showing the essentiality category of annotated genes, including protein-coding genes (left) and functional RNAs (right). (E) Comparative analysis between k-means clusters, area under the curve (AUC) values, and GMM categories for protein-coding genes. The first left column shows the decay trajectories (gray lines) in LD (Y-axis) measured from samples PT along passages 1 to 8 (X-axis). For relative comparison purposes, LD decays were normalized to a gold set of known NE genes. The mean trajectory is plotted with a specific color in comparison to a gray dashed line centered at 1, corresponding to the average LD for a gold set of NE genes. The center column represents the AUC values (X-axis) assigned to the genes in each cluster and essentiality category (Y-axis) as defined in Fig. 1C. Box plots show the median (center line), 25th and 75th percentiles (box bounds), and minimum and maximum values (whiskers). The bar plots in the right column show the number of genes in each cluster and their essentiality category, as estimated by the GMM. The total number of genes is indicated next to the annotated "*n*". Note that for the clustering analysis, we excluded 15 NE genes with a high percentage of sequence repetition, as these interfere with the clustering prediction (see Methods).

respectively. The combination of both libraries, referred here as 'PT', resulted in a total genome LD of 55.6% (453,897 unique insertions of a total of 816,357 possible ones, with an average of 267.1 reads per insertion), and 92.4% when considering a set of known NE genes (Miravet-Verde et al, 2020) (20,385 unique insertions along 22,047 bp; Fig. 1B). This represents ~1 transposon insertion at every base. As expected, the high LDs observed at P1 and T1 samples decreased with increasing rounds of growth selection, highlighting the loss of mutants with reduced fitness (Fig. 1B).

## Gene essentiality estimation in highly saturated transposon libraries

To assign essentiality categories to genes annotated in *M. pneumoniae* (Dataset EV2), we used an unsupervised clustering approach based on a Gaussian mixture model (GMM) (Miravet-Verde et al, 2020). For this, we used the combined datasets obtained at passage 1 (PT1) since they had the maximal LDs. The GMM analysis highlighted a model fitting in four categories (Fig. 1C; Appendix Fig. S2). As expected, the LD distributions showed two dominant gene populations in the left- and right-ends representing E and NE, respectively. The high insertion coverage of the datasets made possible a classification of the population in the middle (defined as F genes) into two categories, referred to herein as F1 (quasi-E) and F2 (quasi-NE).

Considering all the annotated regions of the genome, including protein-coding genes (707), functional RNAs (6), and tRNAs (37), we found a total of $n_E = 255$, $n_{F1} = 90$, $n_{F2} = 86$, and $n_{NE} = 319$ (Fig. 1D; Dataset EV4). Functional RNAs, including ribosomal RNAs (MPNr01, MPNr02, and MPNr03), MPNs01 (4.5S RNA), MPNs03 (RNaseP), and MPNs04 (tmRNA), were essential for survival. Most annotated tRNAs ($n = 37$) were classified as E ($n_E = 29$, $n_{F1} = 4$, $n_{F2} = 2$, $n_{NE} = 2$), except for MPNt08, MPNt16, MPNt23, and MPNt24 (F1), MPNt28 and MPNt36 (F2), and MPNt26 and MPNt15 (NE) that exhibited transposon densities consistent with F and NE categories (Dataset EV2). MPNt26 appears annotated as a serine TCG tRNA, but sequence analysis using ARAGORN (Laslett and Canback, 2004) predicts a lack of tRNA structure, suggesting it represents a pseudo-tRNA of

MPNt25, with whom MPNt26 shares an identity of 89.9%. This is supported as well by the fact that MPNt26 is the only tRNA that does not classify as E after eight passages (Dataset EV4). MPNt15 seems to be dispensable out of all the tRNAs decoding arginine codons, consistent with the fact that MPNt37 may complement it due to the wobble effect (Murphy and Ramakrishnan, 2004), but not the other way around (Dataset EV2). A similar explanation seems true for MPNt28 lysine decoding tRNA that can be replaced by MPNt20, and MPNt36 leucine decoding tRNA that could be replaced by MPNt19 (Dataset EV4). Overall, these results highlight the accuracy of the transposon data to predict different degrees of fitness contribution. Additionally, we looked at the essentiality of the 186 ncRNAs annotated in *M. pneumoniae* (Güell et al, 2009). Of these, 18 and 12% were classified as E and F1, respectively. Except ncMPN336, that is classified as F1 and overlaps with an F2 gene (MPN415), the other E/F1 ncRNAs overlap with E/F1 genes (Dataset EV4). Considering ncRNAs are mostly NE and originate from transcriptional noise in *M. pneumoniae* (Lloréns-Rico et al, 2016), these overlapping ncRNAs are likely to be dispensable for the cell as well. Regarding protein-coding genes, we found a classification consisting of $n_E = 220$, $n_{F1} = 86$, $n_{F2} = 84$, $n_{NE} = 317$ (Fig. 1D). Among the 220 E coding genes, 185 were found conserved in the minimal synthetic bacterium JCVI-syn3.0 (531,490 bp; 438 protein-coding genes) (Hutchison et al, 2016) (Dataset EV4). Also, we found conserved in JCVI-syn3.0 other 165 F1/F2/NE genes of *M. pneumoniae*, 105 of them presenting at least 50% of LD (Appendix Fig. S3; Dataset EV4). Although gene essentiality depends on genetic context and potential interactions involving the encoded protein, these findings suggest that some of these genes may be dispensable in a synthetic genome or, if deleted, could still yield a viable organism, albeit with impaired growth.

The assignment of two F categories revealed sets of genes that can be disrupted but with different impacts on cell viability. To further explore this, we measured the apparent decay in LD for each gene along the passages. This data was first analyzed by clustering the gene LD trajectories using a k-means algorithm (Fig. 1E; Appendix Fig. S4). The number of clusters ($k = 4$) was selected based on the elbow method (see Methods; Appendix Fig. S4), which indicated that four clusters captured the main patterns in the data trajectories (Fig. 1E). Although increasing the number of

clusters could further partition the data, our tests did not substantially improve the interpretability or biological relevance using additional clusters. The four clusters selected (C1 to C4) generally aligned well with the predicted essentiality categories obtained using the GMM model, where gene essentiality was inferred from the single time point PT1 (Fig. 1E). For example, cluster C1 was characterized by constant E LDs from passage 2 onward, and comprised almost exclusively E and F1 genes. In contrast, cluster C4 showed stable LDs up to passage 6, mostly including NE genes. However, this analysis revealed additional layers of complexity for F1, F2, and NE genes. In particular, we identified genes that, while starting at similar LDs and having a similar GMM category, exhibited different decays and cluster assignments.

## Establishing a method for quantitative assessment of essentiality

The above observations prompted us to define a quantitative method of essentiality. For this, we calculated the area under the decay curve (AUC) for each gene, as an indicator of the cell fitness contribution of each gene (Fig. 1E). Using this approach, we observed a good correlation of AUC independently of the passages selected (Appendix Fig. S5). We identified, for instance, 72 F2 and 16 NE genes with high LDs at passage 1, but exhibiting low AUC values (belonging to clusters 1 and 2; Fig. 1E) owing to the low persistence of their insertions. This observation suggests that although these genes are classified as NE/F2 at first passage, they have, in fact, a major contribution to cell fitness. These decay differences could be explained in part by the fact that NE/F2 genes clustered in C2 encode proteins with higher copy numbers in the cell, as compared to genes clustered in C3 and C4 (Fig. EV2). These differences in protein abundance could impact the fitness effect of genes after disruption, as a certain degree of cellular function may persist during the initial passages until the protein is diluted and/or degraded, given that the average protein half-life in *M. pneumoniae* exceeds 60 h (Maier et al, 2011; Burgos et al, 2020). Interestingly, we observed a positive correlation between protein copy numbers and essentiality as previously observed in *Escherichia coli* (Ishihama et al, 2008), suggesting that E genes can be wrongly classified as NE in the first passages. These observations indicate that the number of cell divisions needs to be considered to ensure a proper classification of essentiality.

In summary, we present in Dataset EV4 two complementary essentiality analysis approaches and metric scores that provide a dynamic quantitative assessment of essentiality. This represents an improvement of classification performance compared to previous studies relying on 2 or 3 static categories of essentiality (Barquist et al, 2013a; Lluch-Senar et al, 2015), thus providing a more accurate view on the specific contribution of each gene to cell fitness.

## Essentiality analysis of transcriptional and translational regulatory elements

We then examined the essentiality of potential transcriptional and translational regulatory sequences. For this analysis, we took into consideration different genomic elements that define transcriptional and translational units (Dataset EV2). These include predicted promoters associated with transcription start sites (TSS) experimentally determined (Yus et al, 2012, 2019), predicted intrinsic transcription termination sites (TTS), defined 5′ and 3′ untranslated regions (5′UTR and 3′UTR, respectively), and ribosome binding site (RBS) predicted motifs (see Methods). Regions located between two non-overlapping genes and expressed from the same operon were also considered as possible regulatory regions (named here as inter-genic-intra-operon, or "iGiO"). For the analysis, we discarded elements smaller than 5 bp, as well as those overlapping with E/F1 genes or containing repetitive sequences, as these are known to interfere with the transposon mapping (Miravet-Verde et al, 2020). In total, 1050 out of 3189 regulatory regions remained for downstream analyses.

The majority of the 1050 selected regulatory regions analyzed exhibited a transposon insertion coverage consistent with a NE nature in the PT1 library (Appendix Fig. S6A; Dataset EV5). These results indicate that genes in *M. pneumoniae* do not normally require strict regulation at either the transcriptional or translational level, consistent with the limited transcriptional variation observed for this organism (Yus et al, 2019). The GMM model predicted 43 regions consistent with E and F1 categories (Fig. 2A). After applying a more stringent cut-off criterion of LD ≤ 0.4, we kept 25 essential regulatory regions associated with 22 different genes. These genes with apparent essential regulation, including 6 tRNAs, were mostly related to translation and transcription (Table EV1). The 5′UTR and iGiO elements were the most frequent regulatory features associated with these E regions (Fig. 2A; Table EV1), suggesting that translation initiation, transcript stability, and/or processing, are important contributor factors to the regulation of these genes (Tietze and Lale, 2021). For example, we identified essential iGiO sequences separating rRNAs and tRNA genes, which are probably required for transcript processing and maturation (Shepherd and Ibba, 2015). Similarly, we detected depletion of insertions in the promoter and downstream sequences of the mature tmRNA (MPNs04), consistent with the requirement of these sequences to complement a tmRNA null mutant (Burgos et al, 2021). We also identified cases of E 3′UTR regions, yet these elements often overlap with other genomic features, making it difficult to ascertain their specific contribution (Fig. EV3). One example is the 3′UTR of *mpn517* (uncharacterized protein), which overlaps with the 5′UTR of *mpn516*, encoding the RNA polymerase subunit β. We also observed a depletion of insertions in the TTS and 3′UTR regions of *mpn061* (Fig. EV3). This E gene encodes the signal recognition particle and is oriented in the opposite direction of *mpn060*, an E gene encoding the *S*-adenosylmethionine synthetase (MetK). The *mpn061* terminator could play a role in preventing collisions during convergent transcription and avoid transcriptional interference. Alternatively, sequences present in the 3′UTR may be crucial to modulate mRNA stability or the *mpn061* function (Ren et al, 2017).

Next, we looked for possible enrichments of promoters versus terminators transposon insertions in regulatory regions (Dataset EV5). As expected, no significant differences were found between P and T libraries in regulatory elements exclusively associated with F2 and NE genes (Appendix Fig. S6B). In contrast, we detected significant and persistent enrichment of promoter insertions compared with terminators in upstream regulatory regions of genes belonging to E and F1 categories, including promoter, 5′UTR, and iGiO sequences; and for RBSs at later passages (Appendix

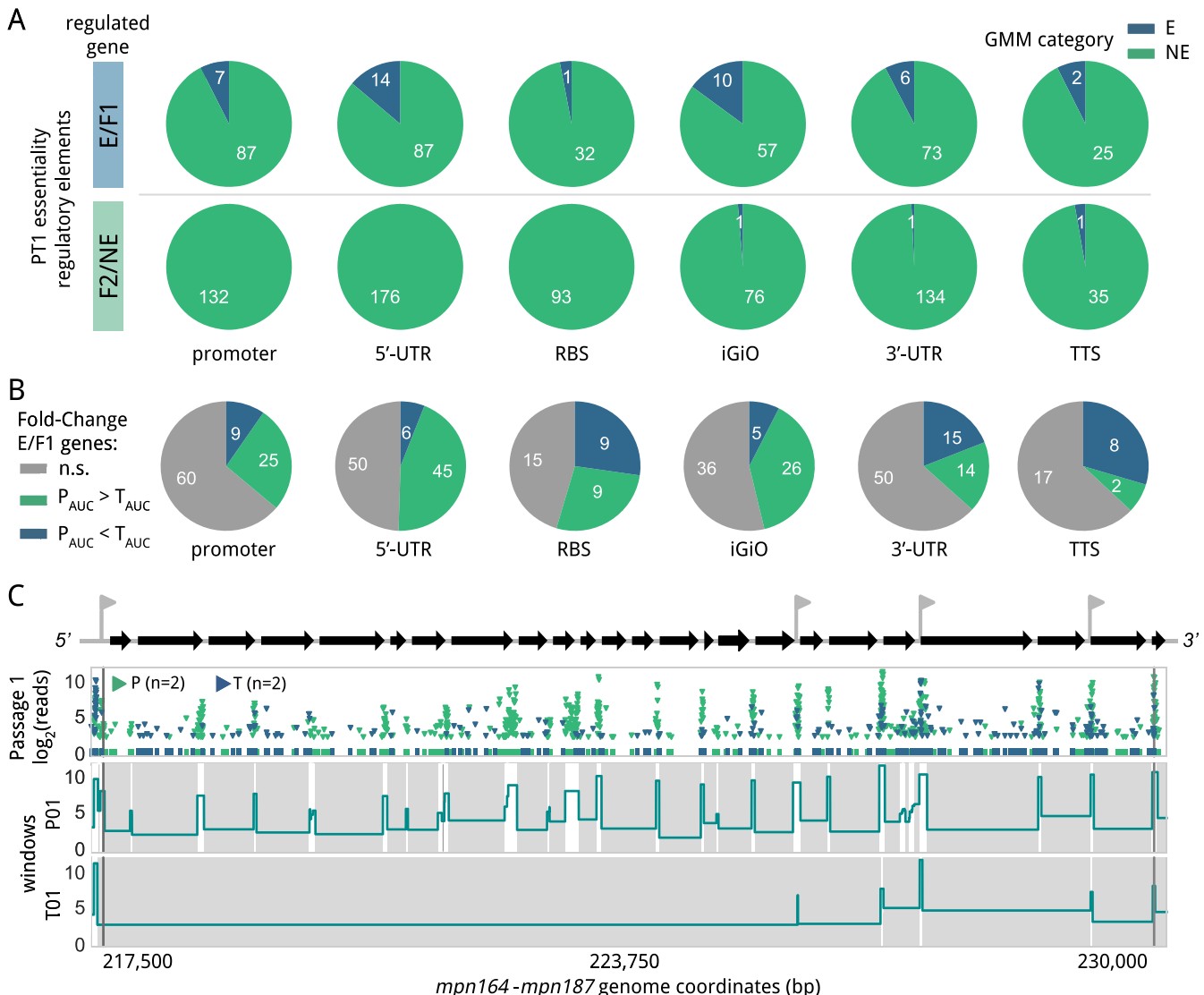

**Figure 2. Essentiality analysis of putative regulatory elements.**

(A) Essentiality assignment of regulatory elements based on a GMM model. Pie charts represent the number of E (blue) and NE (green) elements for each type of regulatory element. The top row shows regulatory elements associated with E/F1 genes, while the bottom row shows those associated with F2/NE genes. Only elements of at least 5 bp, non-repeated and not overlapping with E/F1 genes, are represented. (B) Fold-changes in AUC comparing the persistence of transposons containing promoters (P) versus terminators (T) in different types of regulatory elements. Pie charts represent the distribution of regulatory elements according to the AUC enrichment tendency. In gray, no significant (n.s.), AUC enrichment for P in green, and AUC enrichment for T in blue. (C) Transposon insertion profile at passage 1 across an operon containing 23 genes (*mpn164* to *mpn187*; top black arrows) mainly encoding ribosomal proteins. Each transposon insertion and its read count (Y-axis, in log2) is represented by an inverted triangle, with transposons containing promoters or terminators labeled in green or blue, respectively. The top scheme also shows predicted endogenous promoters (gray flags). Below the scheme, the window essentiality domain at first passage highlights the differential insertion bias for the P and T libraries at passage 1 (P1 and T1, respectively). Gray and white areas represent the E and NE regions, respectively. The height of each domain corresponds to the log2 transformation of the average of read counts per insertion in each window.

Fig. S6C). We did not find statistically significant differences between the two libraries in TTS or 3′UTR regions, in which terminator enrichments could be expected (Appendix Fig. S6B,C). However, when looking at the persistence of the insertions across passages, our AUC analysis showed that while promoters persisted longer than terminators in regulatory elements upstream of E/F1 genes, this trend was inverted in 3′UTR and intrinsic terminators (Fig. 2B). These results suggest that transcription termination in *M.*

*pneumoniae* has a minor influence in cell fitness but could be important for achieving optimal growth in some cases. One example is the enrichment of transposons containing terminators in the endogenous *mpn625* terminator sequence (Fig. EV1A), consistent with the fact that overexpression of the MPN626 ortholog in *M. genitalium* is toxic (Torres-Puig et al, 2015).

We also detected a particular preference for P library insertions close to the TSS and N-terminus of E protein-coding regions at the

nucleotide level (Fig. EV4). As an example, Fig. 2C shows a large transcription unit mainly constituted by E ribosomal genes, in which inter-genic regions mainly accept transposons containing promoters. Remarkably, this pattern was also almost completely maintained up to the sixth passage (Appendix Fig. S7). In contrast, we found that terminator sequences are better tolerated in the intersections delineating sub-operon structures, where internal promoters can rescue transcription (Appendix Fig. S8A).

Despite these general observations, our global genome analysis revealed that 44.7% of 5′UTR regions ($n = 105$) and iGiO elements ($n = 39$) associated with E genes, and lacking clear downstream promoters to rescue transcription, tolerate insertions from the T library with relatively high AUC values ($\geq 2$). These observations suggest either that transcription termination is leaky depending on the context (Reynolds and Chamberlin, 1992), or that weak, unpredicted promoters allow the expression of these genes. In fact, the *M. pneumoniae* sigma 70 factor recognizes a lax consensus promoter sequence, being the Pribnow box the only required element (Güell et al, 2009). Furthermore, we found that terminator sequences inserted in regions lacking apparent mechanisms of transcriptional rescue tended to be associated with highly expressed transcripts (Appendix Fig. S8B), suggesting that transcription termination may not be totally efficient in these cases, as we have shown experimentally (Fig. EV1B). Interestingly, we also observed a progressively faster depletion, after subsequent passages, of the transposons located closer to the 5′ end of polycistronic mRNAs, indicating that disruptions occurring earlier in the transcript induce more severe growth defects (Appendix Figs. S7 and S9).

Altogether, the results show that *M. pneumoniae* generally lacks essential regulatory features and its transcriptome is quite resistant to genomic perturbations. Despite this, there is still a need for preserving transcription and translation of E genes when no additional regulatory elements can rescue their expression after disruption. In addition, the results also highlight that the maintenance of proper levels of gene expression is important to achieve optimal fitness.

## Mapping local-level essentiality using an annotation-independent approach

To expand our essentiality analysis to other non-annotated regions of the genome, we used an annotation-independent approach based on a sliding window analysis applied at the whole-genome level. For this, we examined transposon insertion densities of informative sliding windows of 31 bp (see Methods). This analysis segmented the genome into 4255 regions, defining contiguous E segments surrounded by NE (Dataset EV6). After filtering out segments containing repeated sequences and shorter than 5 bp (see Methods), we detected in the PT1 library samples a total of 914 E segments with an average size of 328 bp, extending to a maximum size of 5594 bp (Dataset EV6). Most of these DNA segments ($n = 879$) overlapped with annotated coding regions, especially F1 and F2 genes, consistent with the discontinuous insertion profile typically observed in these genes (Fig. 3A; Dataset EV7). As expected, the longest E fragments mainly overlapped with E genes and tended to delineate precisely their sequence boundaries as seen for the aforementioned ribosomal operon (Appendix Fig. S7). This highlights the accuracy of the algorithm to define E regions and its usefulness to assist in the annotation of genomes.

Consistent with the compact genome organization of *M. pneumoniae*, we only detected 35 E segments mapping to non-coding regions. Of these, 30 overlapped with putative gene regulatory elements, while five were mapped to other non-annotated areas. Of note, four of them are located in a non-coding region, and one overlaps a putative promoter region, that separates *mpn688* (*soj*) and *mpn001* (*dnaN*) genes (Fig. 3B), which contains the origin of replication (*oriC*) of *M. pneumoniae*, as this region, together with short stretches of the flanking genes, supports replication of suicide plasmids (Blötz et al, 2018). Strikingly, only five out of the ten putative DnaA boxes predicted in the *oriC* were found to overlap with E segments (Blötz et al, 2018), suggesting that many of these predicted DNA boxes are either redundant or not functional, or the requirements for initiation of replication in *M. pneumoniae* are rather relaxed. In fact, two of the E DnaA boxes lie within the E coding sequences of *soj* and *dnaN*, thus making it difficult to ascertain their true essentiality. The discontinuous E profile of the *oriC* region could reflect the importance of the specific arrangement of individual DnaA boxes and their different affinity properties. In other words, insertions between specific DnaA boxes could alter the local distance, indirectly affecting the nucleation process and proper assembly of DnaA oligomers (Wolański et al, 2015). Some of the identified E segments may also be important for binding other essential proteins or regulating certain aspects of DNA replication. In this regard, it is intriguing that the presence of two ncRNA within the *oriC* region. Of note, the 5′ends of these ncRNAs overlap with E segments, raising the possibility that transcription of these ncRNAs may play a role in the initiation of replication *in M. pneumoniae*. Supporting this, ncRNAs have been involved previously in regulating different steps of DNA replication of some bacterial plasmids, or in the recruitment of the origin recognition complex of *Tetrahymena thermophila* (Mohammad et al, 2007).

## Impact of genome accessibility on Tn-Seq data

Next, we wonder whether the insertion profiles of small chromosome regions are affected by the binding of DNA-associated proteins, which could prevent transposon accessibility. To evaluate this effect, we analyzed regions of the genome where we detected bound proteins as revealed by ChIP-seq and protein occupancy data (Yus et al, 2019). In general, we did not find differences in terms of the number of transposon insertions in these regions, but we found some exceptions (Appendix Fig. S10; Dataset EV8). For example, two of the DNA segments in the *oriC* region found to be E could be protected as shown by DNA protection studies (POD) (Yus et al, 2019), or specifically methylated (Lluch-Senar et al, 2013). Additionally, ChIP-seq showed binding between nucleotides 1 and 100 for the SMC-ScpAB complex (crucial role in the organization and segregation of the chromosome (Soppa et al, 2002)), the RNA polymerase complex (around position 550), and DnaC (around position 650) (Yus et al, 2019). Thus, we see that the E regions detected in the *oriC* seem to be bound by different protein complexes. This raises the question of whether some of these small E regions detected are truly essential or if they are merely protected from transposon insertion (see below for further discussion). To further explore whether DNA protection could affect transposition, we looked at the promoters regulated by the three known transcription repressors of *M. pneumoniae* (HrcA, Fur, and WhiA)

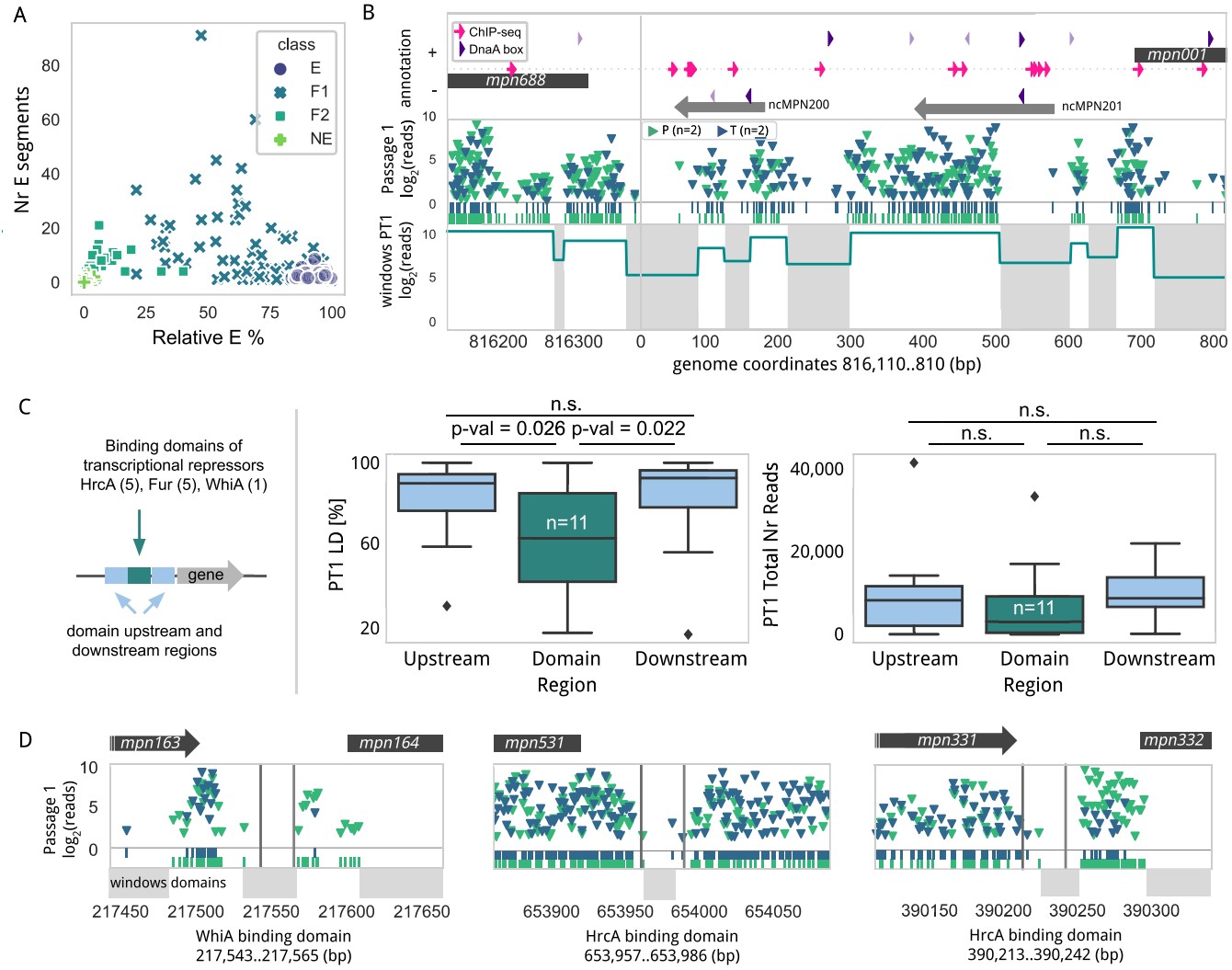

**Figure 3. Local-level essentiality analysis using a sliding window approach.**

(**A**) Scatter plot relating the percentage of each gene being essential by our sliding window approach (X-axis) and the number of E segments (Y-axis) with different coloring and shape based on the essentiality category assigned in the GMM analysis. (**B**) Transposon insertion linear density (LD) in the promoter (P) and terminator (T) libraries at passage 1 (PT1) for the *oriC* genome region. Transposon insertions and read count (Y-axis, in log2) are represented (promoters or terminators labeled in green or blue, respectively). Below, the window essentiality domain highlights the differential segments (gray - E and white - NE). The first row shows the genomic context in both orientations, with gray arrows representing ncRNAs and black representing the N-termini of *mpn001* and *mpn688*. ChIP-seq peaks are represented as pink arrows, while the purple triangles label putative DnaA boxes that are darker when overlapping an E domain. (**C**) Left - Schema of the analysis performed. Center - Box plot analysis of the preferential insertions using the PT1 library in the domain region where a transcription repressor is bound (a total of 11 cases, labeled as "n"), compared with the equivalent size upstream and downstream regions by one-tailed paired *T*-test. Right - same representation, including the total number of insertion reads. The underlying data points are all included in Dataset EV9. Box plots show the median (center line), the 25th and 75th percentiles (box bounds), and the minimum and maximum values (whiskers), or display outliers as diamonds, following the default settings of the Seaborn's boxplot function. (**D**) Transposon insertion profile as shown in panel A for DNA-binding regions of transcriptional repressors. Windows essentiality domains (gray - E and white - NE) are represented below.

(Yus et al, 2019), as these are likely to bind in a more constitutive manner to their DNA target sites and some of them regulate NE genes. Inspection of a total of 11 binding sites for these transcriptional repressors shows, in general, a significant reduction in LD and a decreased trend in total transposon reads, when compared to adjacent regions of the same length (Fig. 3C; Dataset EV9). This is especially clear for the main targets of HrcA (*mpn021*, *mpn332*, and *mpn531*), for two main targets of Fur (*mpn043* and *mpn162*), and one of WhiA (*mpn164*) (Fig. 3D). Of note, *mpn332*,

which encodes the essential Lon protease, is co-transcribed with its upstream gene *mpn331*. Apart from this, it has its own promoter, which is regulated by the HrcA transcriptional repressor. We have previously demonstrated that this regulatory region, where HrcA binds, can be replaced by an inducible system platform (Burgos et al, 2020). This demonstrates that this HcrA-binding domain, which is free of transposon insertions (Fig. 3D), is in fact NE under normal growth conditions if there is an active promoter transcribing *mpn322*. The fact that *mpn531*, *mpn043*, and *mpn162* encode

NE proteins, also supports the conclusion that these regulatory regions are NE and that Tn-seq data can be affected by the competition or interference of DNA-binding proteins.

Overall, these results suggest that the lack of transposon insertions in regulatory regions may, in some cases, be due to the tight binding of regulatory proteins. Similarly, some of the E regions in the *ori* may result from nucleoid organization by the SMC complex (Yus et al, 2019). Therefore, genome accessibility must be considered when interpreting Tn-seq essentiality data of small regions.

## Detection of non-essential regions and domains in essential protein-coding genes

Next, we exploited our sliding window analysis to identify possible NE regions within E and quasi-E (F1) protein-coding genes. We found that E genes tolerate insertions affecting, on average, five and nine amino acids at their N- and C-terminal regions, corresponding to 2% and 4% of the protein length, respectively (Dataset EV7). For F1 genes, the tolerance increased to 13 and 18 amino acids (7% and 9%, respectively; Fig. 4A).

Sequence conservation analysis of orthologs in closely related species revealed that in *M. pneumoniae*, out of 345 E/F1 genes, 193 of them have acquired N- or C-terminal extensions through evolution (Dataset EV7). Notably, ribosomal proteins were highly represented among this group (Appendix Fig. S11), perhaps reflecting strategies for ribosome evolution. Although the selection process underlying these protein extensions is unclear, our data indicate that they are disruptable or at least less essential than the rest of the coding gene. For example, we detected within the *nusA* gene of *M. pneumoniae* (mpn154), a high tolerance of insertions across the 3′ end of this E gene, corresponding to a long C-terminal extension of 124 amino acids (Fig. EV5A). As expected, when analyzing P and T libraries separately, transposons carrying promoters predicted longer N- and C-terminal NE extensions for both E and F genes (Fig. 4B). For example, we found cases like *mpn229* (Fig. EV5B) and *mpn116* (in this case, an F2 gene; Fig. EV5C), in which the NE extensions are covered almost exclusively by transposons containing promoters, which highlights the importance of preserving transcription of downstream genes. Importantly, although the protein extensions identified are apparently NE, our results indicate that the persistence of insertions upon passages in these regions is generally lower than those found in NE coding genes (Fig. 4C). These observations support the notion that these NE extensions do have a contribution to cell fitness in many cases.

Then, we focused our attention on genes containing protein domains exhibiting distinct profiles of essentiality that could suggest modular functionality within the protein. Although our sliding window analysis was efficient in identifying the transposon-tolerant regions within E genes, the algorithm tended to define an excess of E and NE segments across F1 and F2 genes, likely due to statistical noise arising from the lower number of insertions relative to NE genes (see above). To circumvent this, we adopted a manual approach for selecting examples, by analyzing transposon profile plots of each gene individually. After visual inspection, we identified a total of 16 genes presenting an apparent distinct profile of essentiality across the length of the coding region, considering only those genes presenting distinct domains covering

at least more than 15% of the total protein (Table EV2). The boundaries of these domains were further defined based on the position and number of transposon insertions that more clearly separated the differential essentiality profile of these domains. Note that since visual inspection can be susceptible to judgment subjectivity, the exact proposed domain boundaries presented in Table EV2 need to be considered as an approximation. In support of our analysis, many of the identified regions corresponded to known conserved domains assigned to COG or Pfam families. An example is *mpn362*, which encodes a N5-glutamine methyltransferase (HemK), containing a C-terminal protein domain conserved in the YrdC/Sua5 protein family, which is responsible for the formation of the threonylcarbamoyladenosine (t(6)A) modification found at position 37 of ANN decoding tRNAs (El Yacoubi et al, 2009). HemK in *E. coli* methylates the release factor RF1 and RF2 (Heurgue-Hamard, 2002). This post-translational modification has been shown to induce an important stimulatory effect on the release activity of RF2 (Dinçbas-Renqvist et al, 2000), which recognizes UGA and TAA stop codons. In *M. pneumoniae*, the UGA codon is assigned to tryptophan instead of a stop codon. Therefore, only two codons (TAA and TAG) are used to terminate translation, which can be recognized by RF1. Due to this unusual codon usage, *M. pneumoniae* has lost RF2, and only conserves RF1, which is encoded by the gene located upstream of *mpn362*. We found that the HemK domain tolerated transposon insertions while the YrdC/Sua5 domain remained mostly free, indicating that the fitness assigned nature of *mpn362* (classified as F1) can be explained by having a NE domain, but in fact it should be considered as E due to the activity of the YrdC/Sua5 domain (Fig. 4D; Table EV2). To validate these results, we used the SURE-editing genome-engineering tool (Piñero-Lambea et al, 2022) to replace the endogenous *mpn362* gene by N-terminal deletion variants containing only the YrdC/Sua5 domain (Appendix Fig. S12A). We identified two possible in-frame start codons close to the junction of both domains: TTG and ATG, encoding respectively residues L264 and M290. To express the YrdC/Sua5 domain alone, we tested both translation start codons under the control of the P438 promoter. Only positive gene replacements with the construct using the TTG start codon were obtained (Appendix Fig. S12B), consistent with the enrichment of transposons containing promoters detected just before this alternative start codon (Fig. 4D). Then, we assessed whether MPN362 is expressed as a domain fusion protein only, or YrdC/Sua5 can be expressed as a separate domain as well. To test this, we added a FLAG tag at the C-terminus of MPN362 and replaced the endogenous gene with this tagged variant. This FLAG-tagged variant was viable and expressed only detectable levels of the full-length isoform based on Western blot analyses using anti-FLAG antibodies (Appendix Fig. S12C). Altogether, these results demonstrate that the Hemk domain of *M. pneumoniae* is dispensable for cell viability, yet it is expressed as a fusion protein with an essential YrdC/Sua5 domain. These results suggest that either an alternative modifying enzyme of RF1 exists, or *M. pneumoniae* RF1 may function independently of post-translational modifications.

Although differential domain essentiality was generally detected at passage 1, we also found cases in which this distinct behavior was not observed after several passages. For example, *mpn119* that encodes the ortholog of the MG200 gliding motility protein of *M. genitalium* (Pich et al, 2006a; Cloward and Krause, 2009) is fully

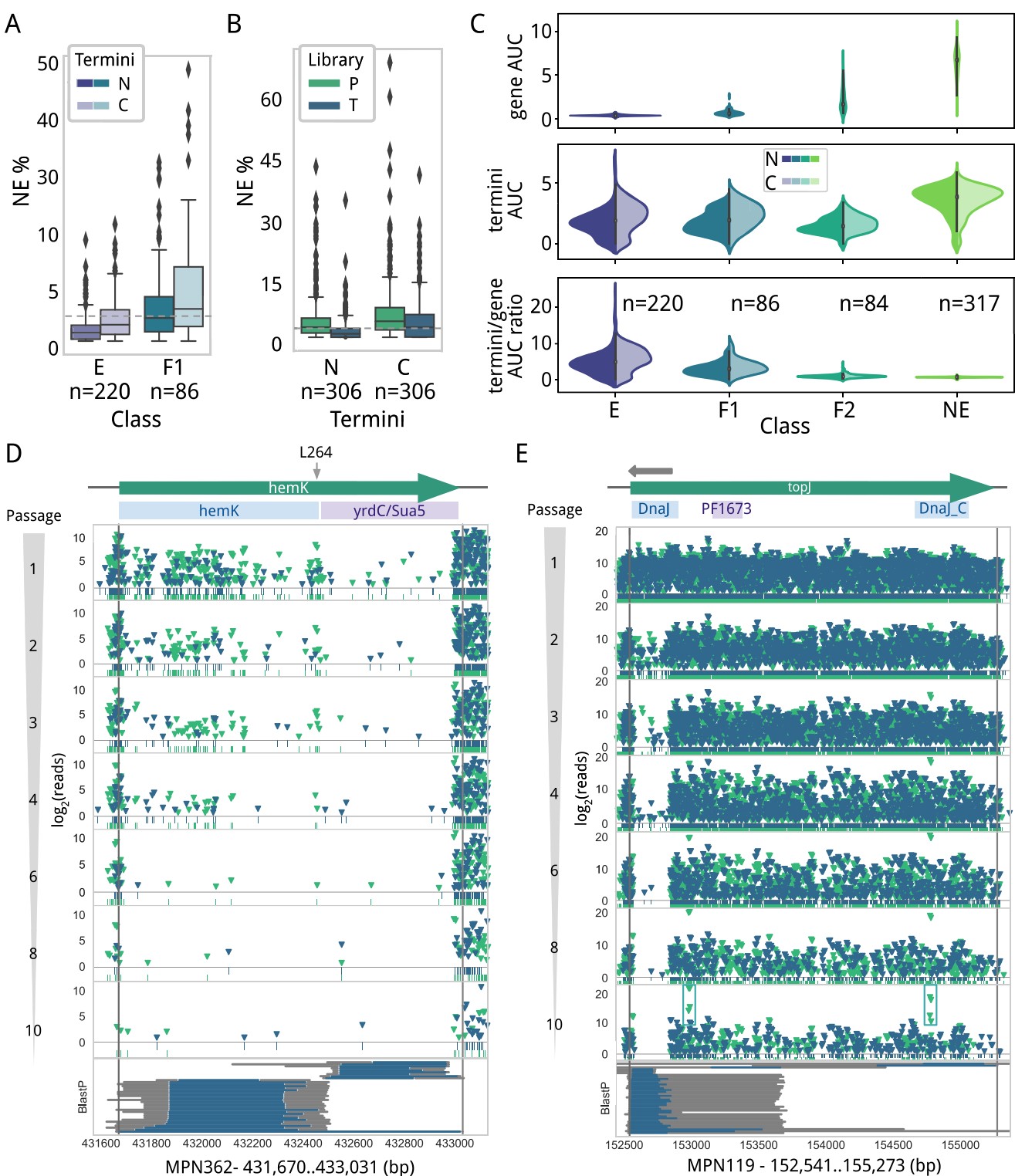

disruptable across the whole coding sequence at passage 1. However, we detected a decrease in LD affecting only the first 100 amino acids, reaching a full essential LD profile at passage 3 (Fig. 4E). This N-terminal region encodes for a conserved DNAJ-like domain, indicating that in addition to the role of MPN119 in

gliding motility, this domain may have an important contribution to cell fitness as well. However, we cannot rule out the possibility that the expression of a putative small protein of 68 amino acids within this region may be responsible for the distinct essentiality profile observed, albeit we did not identify peptides by mass

**Figure 4. Analysis of regions with different essentiality in coding genes.**

(A) Box plot comparing the relative percentage with respect to total gene length being disrupted by insertions when combining P and T libraries at first passage (PT1) for E and F1 genes. The darker box indicates the percentage of the protein at the N-terminus that is found to be NE, and the clear box the same for the C-termini. Notice a dashed gray line marking the 5%, generally used as a threshold for permissive insertions in essentiality studies. Box plots show the median (center line), the 25th and 75th percentiles (box bounds), and the minimum and maximum values (whiskers), or display outliers as diamonds, following default settings of the Seaborn's boxplot function. (B) Same representation as in (C) panel but combining E and F1 genes and coloring by the library (green for promoter (P) and blue for terminator (T), respectively). Box plots show the median (center line), the 25th and 75th percentiles (box bounds), and the minimum and maximum values (whiskers), or display outliers as diamonds, following the default settings of the Seaborn's boxplot function. (C) Violin plots comparing the AUC calculated for genes (top), N- and C-termini (middle), and the ratio of these two values (bottom) by essentiality category from the GMM model. In the termini and ratio panels, the solid (left) and transparent (right) colors of the split violin plots represent the AUC and ratio distributions calculated for N- and C-termini, respectively. Within each violin, a box-and-whisker plot is included, showing the median (center dot) and the minimum and maximum values (whiskers), following the default settings of the Seaborn's violinplot function. (D) Transposon insertion analysis of MPN362 reveals two domains with different essentiality (hemK and yrdC/Sua5). Insertion profiles for promoter (green) and terminator (blue) from passages 1 to 10 are displayed. The last row represents the BLASTP results, in blue the aligned region and gray the non-aligned sequence for the homologous hit. It can be noticed that the two domains are commonly found split in other bacteria. (E) Same representation as in panel (D) for MPN119, which encodes a DnaJ-like protein associated with the attachment organelle. Supported by BLASTP analysis, it is shown a DnaJ domain conserved region at the N-terminus that tolerates transposon insertions during the first passages, but these are rapidly lost after consecutive passages, indicating a different fitness contribution as compared to the rest of the gene. Interestingly, this region also contains a small open reading frame that could encode for a 68 amino acid protein, which is shown by a gray arrow. In addition, two promoter regions (represented in the green boxes at passage 10) are selected with high efficiency, suggesting split variants with enhanced fitness that could match the domains PF16713 (enriched in aromatic and glycine residues box) and a DnaJ C-terminal domain. From the BLASTP analysis, hits mapping each of the three splits suggested by promoter insertions can be retrieved in other bacteria. Note passages 8 and 10 Y-axis are in a different scale, with insertion reaching a log$_2$(reads) value over 20.

spectrometry analysis (Miravet-Verde et al, 2019, 2024). Overall, these observations highlight the importance of considering multiple passages to accurately assess the fitness contribution of genes and protein domains. More importantly, when assigning essentiality to a gene, we need to consider the density of insertions across its length, since genes resulting from the fusion of an E and NE domains, could be classified as F, while they carry an E function.

## Structural essentiality and the discovery of proteins that can be split

We then set out to characterize small NE internal regions within E coding sequences from which we could infer structural protein information. We identified 9 E genes that appear to maintain functionality when expressed apparently as split proteins (Table EV3). One such example is *mpn106*, which encodes the β subunit (PheT) of the phenylalanyl-tRNA synthetase (PheRS). This essential enzyme catalyzes the covalent attachment of Phe to its cognate tRNA and has a complex tetrameric organization composed of two α subunits (PheS) and two β subunits in a (αβ)$_2$ configuration (Mosyak et al, 1995). As expected, both *mpn105* (*pheS*) and *mpn106* (*pheT*) exhibited a clean essential transposon profile across their gene lengths, except for a segment of about 34 bp in *pheT* that tolerated transposons containing promoters (23 ± 5 insertions with 129 ± 16 read count average; $n = 2$), but rarely tolerated transposons containing terminators (3.5 ± 1.5 insertions with 10 ± 5 reads; $n = 2$). Notably, mutants with these transposon insertions were still tolerated after consecutive passages (Fig. 5A). Sequence alignment analysis indicated that this disruptable region coincides with a linker that separates the structural domains B1 and B3 (Fig. 5B) (Mosyak et al, 1995). Inspection of this region revealed three possible in-frame alternative translation start sites, at residues L196, L202, and M208, suggesting that PheT may retain functionality when expressed as two separate protein fragments. To corroborate these observations, we constructed a split *pheT* variant in which we introduced a premature stop codon after residue T207, followed by the P438 promoter sequence to drive the expression of a second

PheT fragment starting at residue M208 (Fig. 5C). This design results in the expression of an N-terminal fragment of 207 amino acids corresponding to the structural domain B1-B2-B1, and a second fragment of 598 amino acids encompassing domain B3 to B8. Using the SURE-editing genome-engineering tool (Piñero-Lambea et al, 2022) we successfully replaced the endogenous *pheST* locus with the mutant variant, demonstrating that *pheT* can be split into two protein fragments without compromising *M. pneumoniae* viability, suggesting that the enzyme retains activity (Fig. 5D). Of note, the β subunit of PheRS contains recognition and binding sites for the tRNA (Roy and Ibba, 2006; Goldgur et al, 1997), suggesting that the split fragments are capable to structurally accommodate the tRNA and interact with the catalytic α subunit. Supporting this, orthogonal aminoacyl-tRNA synthetases/tRNA pairs have been designed by generating split forms of these enzymes (Jiang et al, 2023).

These results support the suitability of transposon mutagenesis to identify structural regions and guide the design of protein-fragment complementation assays.

## Assessment of transposon insertion events improving cell fitness

The transposons used in this study are expected to change the transcription strengths of the genes surrounding the transposon insertion. Although many of these transcriptional perturbations are expected to negatively impact cell fitness, we wondered whether there may be cases promoting an improvement. To identify these cases, we computed the enrichment of normalized read values for both the P and T libraries between passages 1 and 10. To constrain the analysis, only reproducible insertions between replicates were considered. We identified a total of 958 and 1605 positions whose read counts were significantly enriched from passages 1 to 10 exclusively in the P or T libraries, respectively. An additional set of 29 positions belonged to both libraries (Fig. 6A; Dataset EV10). To explore the potential effect of these insertions, we mapped them based on their distance and position to the corresponding gene and non-coding annotations. As expected, the majority of enriched

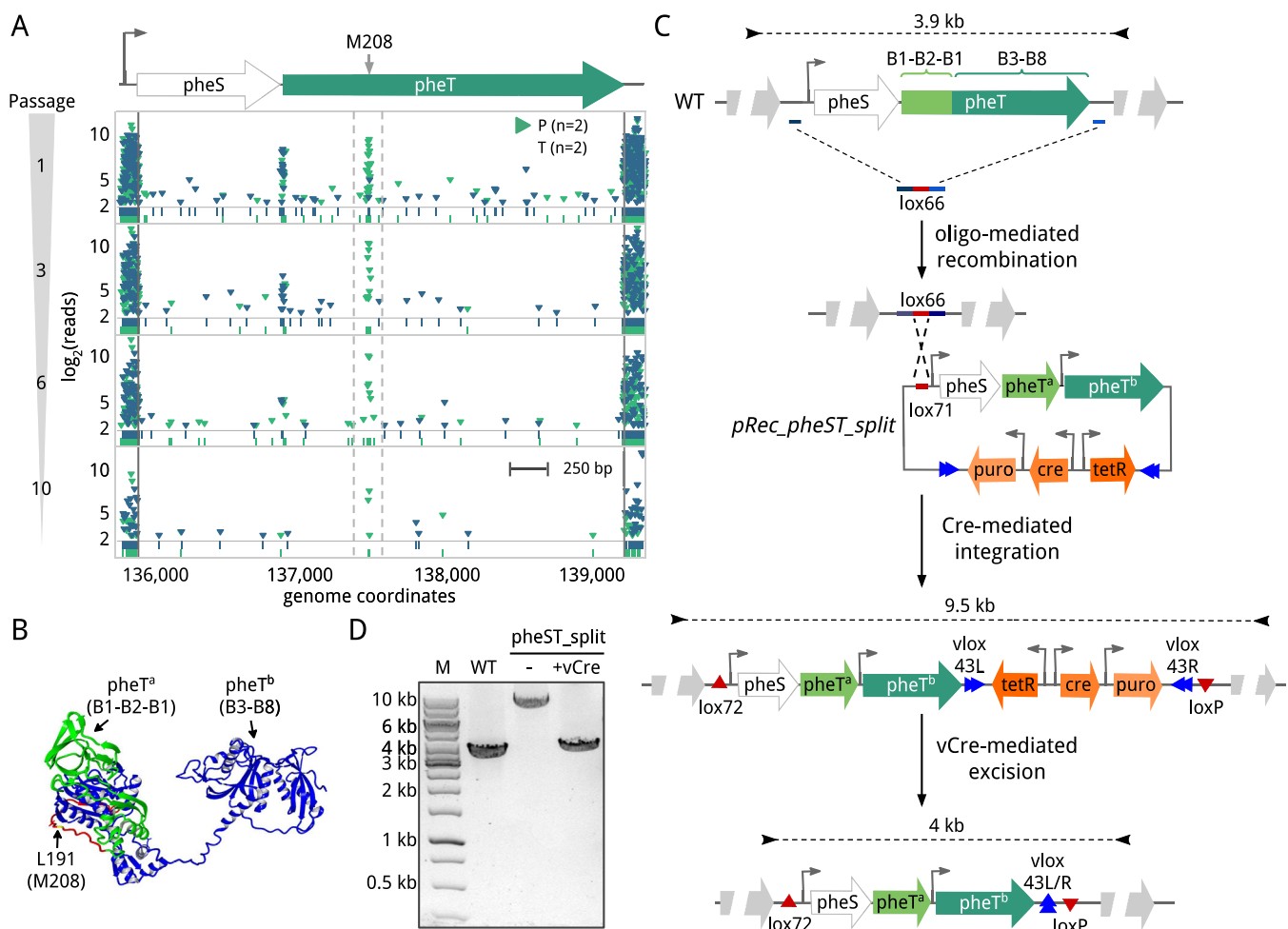

**Figure 5. Identification and validation of a functional split protein.**

(A) Transposon insertion analysis of the *mpn105* (*pheS*) and *mpn106* (*pheT*) genes. The four rows show the transposon insertion profile (green for promoter (P) and blue for terminator (T)) across these E genes at different passages (Left Y-axis). The narrow region bounded by gray discontinuous lines shows preferential insertion of the P library upstream of the M208 residue of the *pheT* coding sequence, which could serve as an alternative translation start codon. (B) Three-dimensional structure of the phenylalanyl-tRNA synthetase beta subunit (PheT) from *Thermus thermophilus* (PDB 1PYS). Domains B1-B2-B3 (in green) and B3-B8 (in blue) are separated by a linker (in red) that tolerates preferential insertions of the P library. Within this linker, we show residue L191 (in yellow), which corresponds to residue M208 of *M. pneumoniae* PheT protein based on sequence homology. (C) Schematic diagram showing the constructs and SURE-editing procedure used to generate a *M. pneumoniae* strain with a split PheT variant. Briefly, deletion of the *pheST* endogenous locus is mediated by oligo recombineering using an oligo containing a lox66 recombination site (in red) flanked by homologous regions (in blue). Gene complementation with the mutant variant is then mediated by Cre-mediated integration of plasmid pRec_pheST_split into the lox66 site, generating lox72 and loxP sites. The plasmid backbone (containing tetR repressor, Cre and puromycin resistance marker) is then removed from the genome by vCre-mediated recombination using the vlox sites present in the pRec_pheST_split plasmid sequence. Sizes of the PCR products expected for each intermediate strain are shown above. (D) PCR analyses using genomic DNA of the WT and intermediate strains before (labeled as "−") and after (labeled as "+") vCre excision are shown. The size of the expected PCR products is shown in panel (C).

insertions overlapped with F2/NE genes (71.4% in P and 73.8% in T; Appendix Fig. S13). We then analyzed the RNA expression of the 147 and 185 genes (including those encoding for functional RNAs), showing enrichment of P transposon insertions in upstream positions and T insertions in downstream positions, respectively (Dataset EV11). In general, we found that these genes presented lower expression, when compared to those NE genes lacking promoter enrichments in P transposon insertions (Fig. 6B). This observation may indicate that the strong P438 promoter is selected in those positions where it could lead to an increase in expression that could improve fitness.

In parallel, we explored the function of these genes by means of a COG enrichment analysis (see Methods). We found an enrichment in genes belonging to the COG categories G - carbohydrate transport and metabolism, D - cell cycle control, cell division, chromosome partitioning; T - signal transduction mechanisms; and S - unknown function. When doing this analysis for genes showing enrichment of insertions containing terminators in upstream positions, we only detected COG enrichment for the S category after applying a hypergeometric test and adjusting it using the Benjamini–Hochberg method (Fig. 6C; Dataset EV12). As examples, we identified enrichment of transposons containing

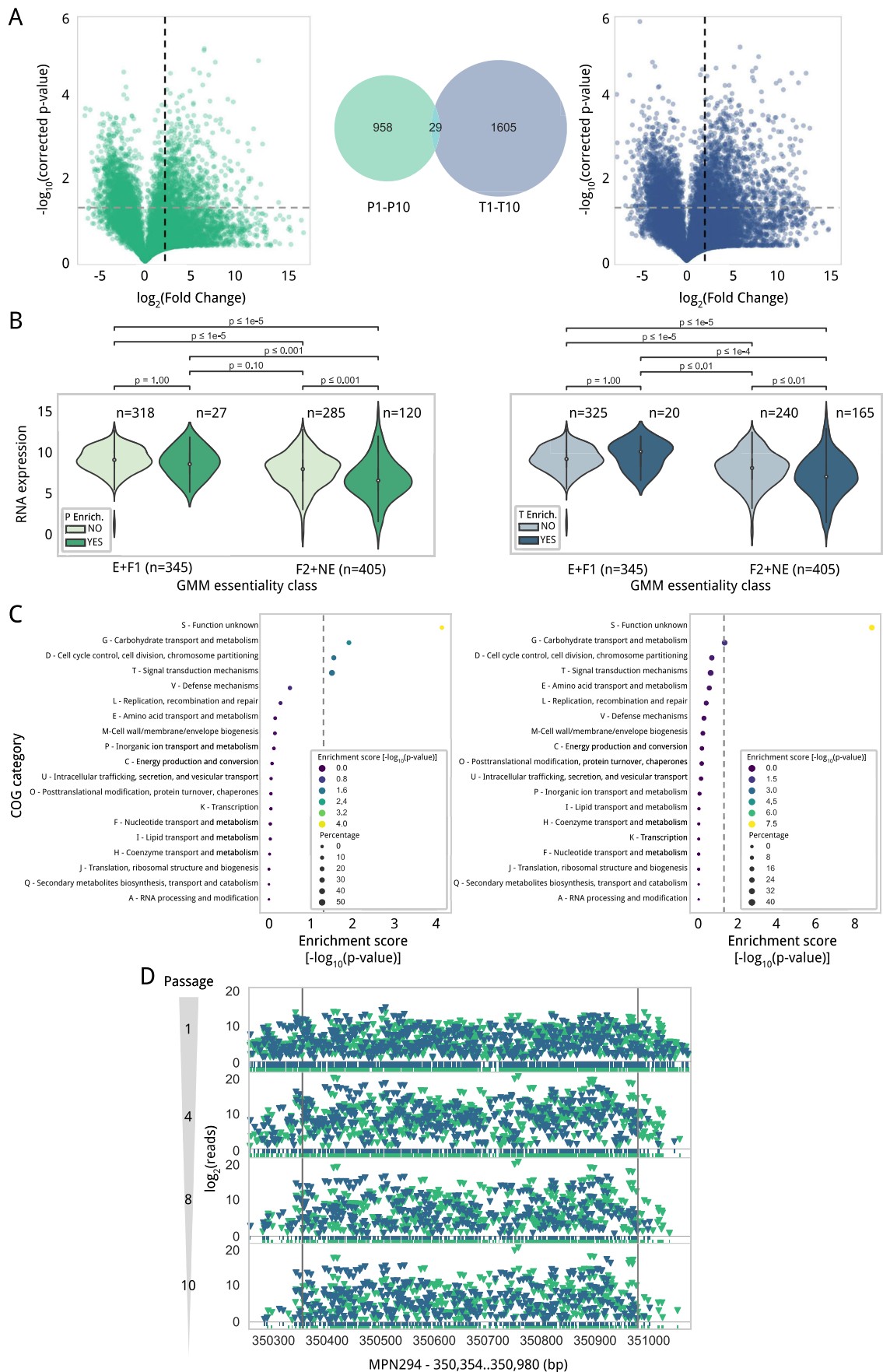

◀ **Figure 6.  Transposon insertion enrichment analysis after 10 serial passages.**

(A) Statistical analysis to identify insertion positions in the genome that are enriched after serial passages. We show the volcano plot analysis for the P (green; $n = 12{,}748$ valid positions that were then statistically tested) and T (blue; $n = 17{,}160$ valid positions statistically tested) libraries (see Methods) as well as the number of positions significantly enriched in each library and their intersection as a Venn diagram (same color code). The source data for this analysis is provided in Dataset EV10. (B) Violin plots comparing the RNA expression, measured in wild-type *M. pneumoniae*, of the 147 genes showing enrichment of P transposon insertions in 5′ upstream positions (darker color) compared with those not having such enrichment (lighter color). NE genes with an insertion selected present a significantly lower expression in wild-type, when compared to those genes lacking P insertion enrichments (all statistical comparisons tested by two-tailed *T*-test). On the right, the same comparative is shown for the 185 genes that present an enriched insertion in the T transposon library (referring this time to 3′ downstream positions with respect to the analyzed genes) compared to the rest of the genes in the same category. Note that in this analysis, we consider both genes encoding proteins and functional RNAs, and we group E and F1 genes together, and F2 and NE genes together. Within each violin, a box-and-whisker plot is included, showing the median (center dot) and the minimum and maximum values (whiskers), following the default settings of the Seaborn's violinplot function. (C) Enrichment in COG categories for genes showing enrichment of P transposon insertions in 5′ upstream positions for the P and T libraries (left and right, respectively). (D) Transposon insertion profiles in the *mpn294* gene across passages for promoter (green) and terminator (blue) libraries. Passages 1, 4, 8, and 10 are displayed.

promoters upstream of the transcriptional regulator MraZ, which regulates the cell division operon, RpoE (MPN024) and RnaseJ1 (MPN280), which modulate transcription levels, and Tig (MPN331) and GroES (MPN574), which are chaperone proteins. Among others, we also found enrichment of transposons putatively affecting the expression of several genes belonging to the PTS system, a group of paralog genes of the MPN039 family with unknown functions, and an operon encoding several lipoproteins (MPN640 to MPN648) (Dataset EV11).

Apart from transcriptional perturbations, we also looked for gene disruptions that could improve cell fitness. In the absence of sampling effects during the passing process, we would expect, in this case, an increase in transposon reads across the whole coding sequence, instead of enrichments in specific positions. To discard cases where an increase in transposon reads could be derived from a few insertions, we subtracted the three insertions with the highest reads at passage 10. After applying these criteria, only *mpn294* with homology to proteins belonging to the DJ-1/PfpI superfamily, was found to contain a consistent enrichment in transposon insertion reads across the whole coding sequence (Fig. 6D).

Overall, these results highlight the power of transposon analysis to infer possible strategies to adjust cell fitness during the design of a bacterial chassis.

# Discussion

This study presents a comprehensive analysis of transposon sequencing data using engineered transposons containing promoter or terminator sequences to explore complex genomic features. On one hand, we provide the essentiality map of a bacterial genome with the highest insertion coverage achieved in a study of this kind, close to 1 nucleobase resolution for NE genes. This improvement in resolution is relevant because essentiality studies using lower insertion coverage can lead to overestimation of E and F genes, significantly reducing the accuracy of essentiality predictions (Miravet-Verde et al, 2020), especially for small genomic regions. Furthermore, we introduce an AUC-based approach that leverages multiple passages to provide a quantitative metric capturing not just a static snapshot of essentiality, but also fitness measured as the persistence of insertions over time. This revealed a complex landscape of gene essentiality, far from simplistic models relying on static and binary classifications. While previous studies have shown that gene essentiality is a conditional trait that depends on the

genetic and environmental context (Larrimore and Rancati, 2019), our study highlights that genes classified in a particular category have, in fact, distinct degrees of fitness influence. This argues for the need for more accurate methods to improve the resolution of categorization.

The high transposon coverage obtained in this study allowed us to establish four essentiality categories based on a standard GMM model, corresponding to essential (E), non-essential (NE), quasi-essential (F1), and quasi-non-essential (F2); whereas previous essentiality studies in *M. pneumoniae* were able to identify only three categories, with ~4 insertions per bp for NE genes in *M. pneumoniae* (Lluch-Senar et al, 2015). When we looked at the persistence of the transposons across passages, we defined four clusters that in general aligned well with the GMM essentiality categories. Interestingly, however, we noted that some genes exhibited different profiles of transposon persistence, despite having the same essentiality category based on the GMM model. These results indicate that genes initially predicted with the same essentiality category have, in fact, a distinct fitness influence. Indeed, our analyses suggest that the initial concentration of the gene products may be a possible contributing factor explaining these differences. These observations motivated us to implement an alternative clustering method based on a k-means algorithm that integrates temporal datasets by computing transposon decay curves over time. This approach offers a dynamic and quantitative estimation of the impact of loss or gain of function, thus enabling us to distinguish the contribution of each gene, including those that are apparently dispensable for cell growth. This quantitative information can be particularly useful for engineering purposes and the optimization of cellular processes.

Taking advantage of the 1-base resolution of the produced transposon data, we applied this methodology to perform a systematic interrogation of a bacterial genome to also infer essentiality information for less explored genomic features. To this end, we applied a sliding window analysis to detect highly essential sequences across the genome (Zhang et al, 2012; Aseev et al, 2024). Probably because of the compact genome organization of *M. pneumoniae*, we did not find many essential elements in non-annotated regions except for defined sequences in a few regulatory regions, mostly related to translation and transcription, and in the origin of replication. However, the essentiality of these small regions should be considered with caution since we found that tight binding of proteins to DNA can prevent transposon insertions. As an example, regions like the CIRCE element of the Lon protease

(MPN332), which is targeted by the HcrA repressor, is wrongly assigned as essential in standard growth conditions based on the transposon sequencing data. Thus, these data should be integrated with ChIP-seq or POD data to clearly distinguish potentially essential regulatory regions from those that cannot be defined without further experimentation. It is also possible that some E regions, like those at the putative origin of replication, could be due to nucleoid structural constraints preventing access of the transposon, although we did not find any bias in the average density of insertions over the genome.

When we compared the promoter and terminator libraries, we detected significant and persistent insertions of the P library in upstream regulatory regions of genes belonging to E and F1 categories, including promoter, 5′UTR, and iGiO sequences, and also, for potential RBSs at later passages. In contrast, we could not see a general enrichment of insertions from the T library at predicted TTS and 3′UTR regions. However, there are a few examples where we clearly observed preferential insertions of the terminator library at a TTS, like in the case of the TTS of *mpn625*, which prevents expression of MPN626. The importance of this termination site is consistent with the fact that overexpression of the MPN626 ortholog in *M. genitalium* is toxic (Torres-Puig et al, 2015). When looking at the persistence of the insertions across passages, we found that while insertions from the P library persisted longer than those of the T library upstream of E/F1 genes, this trend was inverted at 3′UTR and predicted TTS. This again illustrates the importance of analyzing multiple passages when analyzing the essentiality of genomes and suggests that transcription termination in *M. pneumoniae* has a minor influence on cell fitness, but can be important in some genetic contexts for achieving optimal growth. It is also important to emphasize that while, as described above, we see very few cases of preferential insertions from the T library, we see many cases where there is a negative selection of terminators while P insertions are maintained. One strikingly clear case is the main and essential ribosomal operon, thus showing that the terminator sequence introduced in the T library transposon is functional.

When looking at annotated regions, we identified some protein-coding genes exhibiting different levels of essentiality across the coding sequence, suggesting that they likely represent multi-functional genes that originated from the fusion of separate protein domains, or from elongation of the coding sequence. We could define non-essential N- and C- terminal extensions of essential genes, which were found to be quite common among ribosomal proteins. Although these extensions are not required for cell survival, our analysis across passages suggests that their presence confers, in many cases, a certain fitness advantage. In this regard, it has been suggested that some extensions of eukaryotic ribosomal proteins are involved in the recruitment and binding of key initiation factors (Ghosh and Komar, 2015). Additionally, some ribosomal proteins have acquired other functions beyond the ribosome (Zhou et al, 2015), opening the possibility that these extensions may have evolved to fulfill these functions. For example, the transcription elongation factor NusA interacts with the RNA polymerase via its N-terminal domain, but it has been recently shown that it also interacts with ribosomes via its C-terminal region, suggesting that NusA could mediate transcription-translation coupling in *M. pneumoniae* (O'Reilly et al, 2020). Although this C-terminal extension was previously established as essential (O'Reilly et al, 2020), our data indicate that it is fully disruptable at passage 1, yet transposons are negatively selected in this region after consecutive passages. This suggests a contribution of this C-terminal extension to protein function, likely enhancing cell fitness. Remarkably, we could also infer structural information on essential proteins with a modular domain architecture, where one of the domains is essential while the other is not, confirming that they have arisen from gene fusion. An example that we validated experimentally is the *mpn362* gene, which contains an NE (Hemk) and an E (YrdC/Sua5) domain. Thus, this type of analysis shows its application when trying to define a minimal genome since it can identify NE functions present as fusion proteins that could result in incorrect assignment of essentiality (i.e., assigning F when the gene is actually E). Finally, both the inclusion of promoter sequences in the transposon design and the 1 bp resolution of our transposon data allowed us to identify regions in proteins where they could be split while maintaining their functionality. An example that we validated in this study is the Phenylalanyl-tRNA synthetase.

In addition, the use of engineered transposons bearing distinct transcription regulatory elements enabled us to assess the impact of transcription and translation disturbance on cell fitness. We found that gene expression in *M. pneumoniae* is, in general, resistant to transposon perturbations, indicating a lack of strict regulation in normal growth conditions. This would also suggest coding and non-coding regions could evolve independently, as shown in prokaryotes (Rogozin et al, 2002) and eukaryotes (Kowalczyk et al, 2022). Despite this, the expression of essential genes is preserved in mutant libraries, and most of the insertions are negatively selected after consecutive passages, emphasizing the importance of maintaining proper levels of expression to achieve optimal fitness. By contrast, we also identified insertions conferring competitive benefits in specific positions of the genome, especially those from the promoter library in upstream positions of some low-expressed genes, suggesting possible engineering strategies to obtain highly adapted strains. We also found that disruption of *mpn294* was highly selected in our growth competition assay. This gene encodes a protein of unknown function belonging to the DJ-1/PfpI superfamily, which includes some chaperones, proteases, and other stress response proteins (Smith and Wilson, 2017). In *Pseudomonas aeruginosa*, *pfpI* defective mutants exhibit higher spontaneous mutation rates and are more sensitive to different types of stress (Rodríguez-Rojas and Blázquez, 2009). Thus, although MPN294 may play a direct role, it is tempting to speculate that its disruption may cause increased mutation rates, leading to a larger repertoire of mutants. In this scenario, mutations providing a competitive advantage during growth selection are expected to be positively selected along with MPN294 inactivation.

In conclusion, we describe a novel methodology of essentiality assessment using a k-means unsupervised clustering method that, in combination with multiple passages, provides quantitative information about the contribution of different genetic elements in the overall fitness of a genome-reduced bacterium. Given the widespread use of Tn-Seq in microbiology (Goodall et al, 2018; Christen et al, 2011; Wong et al, 2022; Akusobi et al, 2022; Meeske et al, 2016), we envision that the methodology described here can be applied to other model organisms, only limited by the transposon insertion coverage that can be achieved in each particular cellular system. Altogether, we show that Tn-Seq data

can be very informative as a resource for fundamental biology research and synthetic biology. Future work aiming to determine conditional essentiality across different genetic and environmental contexts, such as cell stress or in vivo conditions (Danchin and Fang, 2016; Luhua et al, 2013; Cain et al, 2020; Zhu et al, 2018), may reveal that the essentiality status of specific genes can change under such conditions, potentially providing insights on the roles of these genes.

# Methods

### Reagents and tools table

| Reagent/resource | Reference or source | Identifier or catalog number |
|---|---|---|
| **Experimental models** | | |
| *Escherichia coli* DH5α | NEB | C2987H |
| *M129_GP35* | Piñero-Lambea et al, 2022 | |
| *M129_pMTnCat_BDPr* | Miravet-Verde et al, 2020 | |
| Other *Mycoplasma* strains | This study | Table EV4 |
| **Recombinant DNA** | | |
| pMTnCat | Burgos and Totten, 2014 | |
| pMTnTetM438 | Pich et al, 2006b | |
| pMTnCat_BDPr | Miravet-Verde et al, 2020 | |
| pLoxPuroCre | Piñero-Lambea et al, 2022 | |
| pGentaVcre | Piñero-Lambea et al, 2022 | |
| Other plasmids | This study | Table EV5 |
| **Antibodies** | | |
| Rabbit polyclonal anti-CAT | Abcam | ab50151 |
| Rabbit polyclonal anti-DsRed | Takara | 632496 |
| Rabbit polyclonal anti-RL7 | Richard Herrmann lab | |
| Mouse monoclonal anti-FLAG M2 | Sigma | F3165 |
| Sheep polyclonal HRP-conjugated anti-mouse IgG | Sigma | A6782 |
| Goat polyclonal HRP-conjugated anti-rabbit IgG | Sigma | A0545 |
| **Oligonucleotides and other sequence-based reagents** | | |
| Primers | This study | Table EV6 |
| **Chemicals, Enzymes and other reagents** | | |
| Puromycin | Gibco | A1113803 |
| Chloramphenicol | Sigma | C0378 |
| Tetracycline | Sigma | T7660 |
| Gentamicin | Sigma | G1397 |
| Anhydrotetracycline | Takara | 631310 |
| *Bam*HI | NEB | R0136S |

| Reagent/resource | Reference or source | Identifier or catalog number |
|---|---|---|
| *Eco*RV | NEB | R0195S |
| Phusion High-Fidelity DNA Polymerase | Thermo Scientific | F530S |
| HEPES | Sigma | H4034 |
| Sucrose | Sigma | 84097 |
| Glucose | Sigma | G-8270 |
| PPLO broth | Difco | 255420 |
| Phenol red | Sigma | P3532 |
| Heat-inactivated horse serum | Life Technologies | 26050088 |
| Skim milk | Millipore | 70166-500 G |
| Tween-20 | Sigma | P7949-500ML |
| NuPAGE ™ LDS loading buffer | Invitrogen | NP0007 |
| **Software** | | |
| FASTQINS v1.0 | CRG-CNAG/fastqins: Pipeline for transposon sequencing processing Miravet-Verde et al, 2020 | |
| ANUBIS v1.0 | CRG-CNAG/anubis: Python package for the analysis of tn-seq data Miravet-Verde et al, 2020 | |
| SPAdes v3.14.1 | Bankevich et al, 2012 | |
| QUAST v5.0.2 | Gurevich et al, 2013 | |
| snippy v4.6.0 | https://github.com/tseemann/snippy | |
| NucDiff v2.0.3 | Khelik et al, 2017 | |
| **Other** | | |
| BCA Protein Assay Kit | Pierce | 23225 |
| MasterPure DNA Purification Kit | Epicentre | MCD85201 |
| NuPAGE 4–12% Bis-Tris precast polyacrylamide gels | Invitrogen | WG1402BX10 |
| Supersignal West Femto Chemiluminescent Substrate | Thermo Scientific | 34096 |
| NEBNext® Ultra™ DNA Library Prep Kit for Illumina | Illumina | E7370L |
| AgenCourt AMPure XP beads | Beckman Coulter | A63882 |
| QIAquick Gel Extraction Kit | Qiagen | 50928706 |
| NEBNext® Multiplex Oligos for Illumina | Illumina | E7335L |
| KAPA Library Quantification Kit | KapaBiosystems | KK4835 |
| BioAnalyzer | Agilent | |
| MiSeq Sequencing System | Illumina | |

| Reagent/resource | Reference or source | Identifier or catalog number |
|---|---|---|
| HiSeq 2500 sequencing platform | Illumina | |
| Tecan Spark plate reader | Tecan | |
| Gene Pulser XCell™ electroporation system | Bio-Rad | |
| iBlot™ dry blotting system | Invitrogen | |
| LAS-3000 Imaging System | Fujifilm | |
| Bioruptor sonication system | Diagenode | |

## Bacterial strains and growth conditions

Bacterial strains used in this study are summarized in Table EV4. Wild-type *M. pneumoniae* lab strain M129 (referred to here as M129_LSR) and its derivatives were grown in a modified Hayflick medium at 37 °C under 5% $CO_2$ in tissue culture flasks. This medium was supplemented with puromycin (3 μg/ml), chloramphenicol (20 μg/ml), tetracycline (2 μg/ml), or gentamicin (100 μg/ml) for the selection of transformants, and 0.8% agar when solid medium was required. To induce the Ptet promoter system, we used anhydrotetracycline (aTc) at 5 ng/ml. For cloning purposes, we used *Escherichia coli* strain Dh5α (New England Biolabs), which was grown at 37 °C in LB broth or LB agar plates containing ampicillin (100 μg/ml).

## Molecular cloning

Plasmids used in this study are summarized in Table EV5 and were constructed by Gibson assembly using the primers listed in Table EV6 as follows:

Vectors pMTnCat_BDPr and pMTnCat_BDter were designed to generate two distinct mycoplasma transposon libraries. Construction of pMTnCat_BDPr plasmid was previously described (Miravet-Verde et al, 2020). The pMTnCat_BDter transposon vector was obtained by amplifying the *cat* gene using the ter_cat_F and ter_cat_R primers, and the resulting PCR product was cloned by Gibson assembly into a pMTnCat vector (Burgos and Totten, 2014) opened by PCR using primers p_ter_F and p_ter_R. Note that the inverted repeats of the transposon are designed to include stop codons in all three possible frames of insertion. Therefore, all transposon insertions generate protein fusions that, depending on the frame, can include the addition of only one amino acid (W or R or G), four amino acids (IKSV), or seven amino acids (DKVRIIV).

To assess the termination activity of the ter625 promoter, we used the *cat* and *cherry* gene reporters expressed as polycistronic or fusion transcripts, respectively. To test the polycistronic context, we generated the plasmid pMTnTc_ter625_RBS_CAT, which contains the *tetM438* gene followed by the ter625 terminator sequence, an RBS sequence, and the CAT coding sequence. To obtain this plasmid, we amplified the *cat* gene using two consecutive PCR reactions to include the ter625 and the RBS sequences. For the first reaction, we used the pair of primers ter625_RBS_CAT_F and p_CAT_R and then primers p_ter625_F and p_CAT_R. The final PCR product was cloned by Gibson assembly into a digested *Bam*HI pMTnTetM438 vector (Pich et al, 2006b). We also generated the plasmid control pMTnTc_ter625mut_RBS_CAT, in which we mutated the ter625 sequence to disrupt the terminator hairpin structure. For this, we used the same cloning strategy but using the primer ter625_RBSmut_CAT_F instead of the ter625_RBS_CAT_F primer. To test the fusion context, we generated the pMTnTc_P438_MP200_ter625_cherry plasmid. This construct carries an in-frame N-terminal fusion of a 29 amino acid peptide of mp200 ORF to the Cherry coding sequence (Burgos et al, 2021), but separated by the ter625 terminator sequence, which was included in-frame with the fused coding sequences. To obtain this plasmid, we amplified the mp200 peptide using two consecutive PCR reactions, first with the pair of primers p438_MP200_F and ter625_MP200_R, and then with p_P438_F and ter625_MP200_R. The final PCR product was cloned by Gibson assembly into a digested *Eco*RV pMTnTetM438 vector together with the Cherry coding sequence, which was amplified with primers ter625_Ch_F and p_Ch_R. As a control, we generated the pMTnTc_P438_MP200_ter625mut_cherry plasmid, in which the ter625 sequence was mutated to disrupt the terminator hairpin structure. For this, we used the same cloning strategy but using the primer ter625mut_Ch_F instead of the ter625_Ch_F primer.

To confirm the viability of a subset of transposon insertions, we performed *M. pneumoniae* genome editions using the SURE-editing tool (Piñero-Lambea et al, 2022) and plasmids pRec_ΔNt_L_hemK, pRec_ΔNt_M_hemK, pRec_hemK_FLAG, and pRec_pheST_Stop_P438. All these plasmids are based on the selector plasmid pLoxPuroCre that allows the selection of edited cells after oligo recombineering (Piñero-Lambea et al, 2022). Since the genome editions attempted in this study affected essential regions, we modified the pLoxPuroCre selector plasmid to include essential regions and the intended gene mutations to perform DNA replacements of large genomic regions. For example, the pRec_ΔNt_L_hemK selector plasmid contains the essential genes *mpn360*, *mpn361* plus an N-terminally truncated gene version of *mpn362* (previously annotated as *hemK*), in which L264 was used as a start codon and its expression driven by the P438 promoter (Pich et al, 2006b). To obtain this plasmid, two PCR fragments were obtained using genomic DNA as a template, and the pair of primers p_MPN360_F / P438_MPN361_R and P438_ΔNt_L_hemK_F / p_hemk_R. To obtain the final construct, the two PCR products were cloned by Gibson assembly into the pLoxPuroCre vector opened by PCR using primers p_pRec_F and p_pRec_R. A similar strategy was used to obtain the pRec_ΔNt_M_hemK plasmid, which contains instead an N-terminally truncated version of the *mpn362* gene starting at residue M290. In this case, the second PCR product was replaced and obtained by using primers P438_ΔNt_M_hemK_F and p_hemk_R. Similarly, the pRec_hemK_FLAG plasmid contains the essential genes *mpn360*, *mpn361* plus a C-terminally FLAG-tagged version of *mpn362*. To obtain this construct, we performed two consecutive PCR reactions, first with the pair of primers p_MPN360_F and hemk_FLAG_R, and then with primers p_MPN360_F and p_hemk_R. The resulting PCR product was cloned by Gibson assembly into the pLoxPuroCre vector, opened by PCR using primers p_pRec_F and p_pRec_R. Finally, the pRec_pheST_Stop_P438 plasmid contains the essential

genes *mpn105* (pheS) and *mpn106* (pheT), with the exception that the PheT coding sequence was prematurely interrupted by placing a stop codon after residue T207, followed by the P438 promoter sequence to drive the expression of a second PheT fragment starting at residue M208. To obtain this construct, two PCR fragments were obtained using genomic DNA as a template, and the pair of primers p_pheST_F/ pheT_StopP438_R and StopP438_pheT_F/p_pheST_R. To obtain the final construct, the two PCR products were cloned by Gibson assembly into the pLoxPuroCre vector opened by PCR using primers p_pRec_F and p_pRec_R.

## Validation of transposon data by constructing specific mutant strains

To assess the quality of the transposon data, we constructed specific mutants to mimic the effect of a subset of transposon insertions and thus confirm the viability of these gene disruptions. As examples, we constructed *hemK* and *pheT* mutants using the SURE-editing tool, a genome-engineering method that combines genome editing by oligo recombineering and selection of edited cells by plasmid integration mediated by site-specific recombinases (Piñero-Lambea et al, 2022). A major advantage of this combined strategy is that it allows the introduction of gene platforms, thus enabling precise gene replacements even in essential regions of the genome. We took advantage of this improvement to introduce specific mutations affecting *hemK* or *pheT* genes without disrupting the essential genomic context. To obtain the *hemK* mutant derivatives (M129_ΔNt_L_hemK, M129_ΔNt_M_hemK, and M129_ hemK_FLAG) and the *pheT* mutant (M129_pheST_Stop_P438), we co-electroporated the M129-GP35 strain (Piñero-Lambea et al, 2022) with 0.5 nmols of editing oligo (ssDNA_hemK for hemK mutant derivatives or ssDNA_pheST for pheT mutant) and 2 μg of the corresponding selector plasmid described above (pRec_ΔNt_L_hemK, pRec_ΔNt_M_hemK, pRec_hemK_FLAG, and pRec_pheST_Stop_P438). The editing oligo contains two 40-nucleotide-long regions homologous to the adjacent sequences of the intended deletion area, and separated by a lox66 recognition site that is used to select the edited cells via Cre-integration of the selector plasmid. To allow expression of the puromycin marker and induce Cre expression from the selector plasmid, cells were recovered after electroporation at 37 °C for 2 h in the presence of aTc. Transformed cells were then selected during 24 h in 25 ml cultures supplemented with aTc and puromycin to ensure oligo recombineering and subsequent selection of the edited cells by the integration of the selector plasmid. As described above, the selector plasmid was engineered to contain a DNA platform aimed to complement the deletion of essential regions mediated by the editing oligo, and to introduce the desired mutations. Finally, puromycin-resistant colonies were isolated from agar plates supplemented with puromycin, and DNA rearrangements were verified by Sanger sequencing and PCR analysis using primers listed in Table EV6. The backbone of the selector plasmid was further excised from the genome by electroporating the mutants with 2 μg of pGentaVcre suicide vector (Piñero-Lambea et al, 2022), which contains the vCre recombinase. PCR analysis and Sanger sequencing were performed using the primers listed in Table EV6 to confirm the intended genome edition.

## Immunoblot analysis

Mycoplasma cells were washed twice with 1× PBS, scraped off from the flasks, and centrifuged at 13,100 × *g* for 10 min. The cell pellet was then resuspended in 1% SDS lysis buffer, and disrupted using a Bioruptor sonication system (Diagenode) with an On/Off interval time of 30/30 s at high frequency for 10 min. Cell lysates were quantified by using the Pierce™ BCA Protein Assay Kit and mixed with NuPAGE™ LDS loading buffer (Invitrogen). Ten μg of each cell extract was subjected to electrophoresis through NuPAGE™ 4–12% Bis-Tris precast polyacrylamide gels (Invitrogen), and proteins transferred onto nitrocellulose membranes using an iBlot™ dry blotting system (Invitrogen). For immunodetection, membranes were blocked with 5% skim milk (Sigma) in PBS containing 0.1% Tween-20 solution and probed with the following antibodies. For the assessment of the ter625 termination activity, we used rabbit polyclonal antibodies anti-CAT (Abcam, 1:2000), anti-DsRed (Takara, 1:2000), or anti-RL-7 (kind gift of Dr. Herrmann, Heidelberg University, 1:5000) as loading control. For the detection of putative HemK isoforms, we used the mouse monoclonal anti-FLAG M2 (Sigma, 1:5000). Anti-mouse IgG (1:10,000) or anti-rabbit IgG (1:5000) conjugated to horseradish peroxidase (Sigma) were used as secondary antibody. Blot signals were detected using the Supersignal™ West Pico or Femto Chemiluminescent Substrate Detection Kit (Thermo Scientific) and the LAS-3000 Imaging System (Fujifilm).

## Genome sequencing and assembly of the *M. pneumoniae* strain used in this study

To increase the accuracy of the transposon insertion mapping, we sequenced the genome of our *M. pneumoniae* lab strain (M129_LSR). For this, genomic DNA was extracted using the MasterPure™ DNA Purification Kit (Epicentre, Cat. No. MCD85201). Libraries were prepared using the NEBNext® DNA Library Prep Reagent Set for Illumina® kit (ref. E7370L) according to the manufacturer's protocol. Briefly, 500 ng of DNA was fragmented to ~600 bp and subjected to end repair, addition of "A" bases to 3′ ends, ligation of NEBNext hairpin adapter, and USER excision. All purification steps were performed using AgenCourt AMPure XP beads (ref. A63882, Beckman Coulter). Library size selection was done with 2% low-range agarose gels. Fragments with an average insert size of 660 bp were cut from the gel, and DNA was extracted using the QIAquick Gel extraction kit (ref. 50928706, Qiagen) and eluted in 15 μl EB. The adapter-ligated size-selected DNA was used for final library amplification by PCR using NEBNext® Multiplex Oligos for Illumina. Final libraries were analyzed using Agilent DNA 1000 chips to estimate the quantity and check size distribution, and were then quantified by qPCR using the KAPA Library Quantification Kit (ref. KK4835, KapaBiosystems). Libraries were loaded at a concentration of 15 pM onto a flow cell together with other samples at equal concentration (half a run), and were sequenced 2 × 300 on Illumina's MiSeq. Reads preprocessing consisted of trimming reads using SeqPurge v0.1-478-g3c8651b (Sturm et al, 2016) with a minimum read length of 80 and default parameters for base calling quality threshold. Genome assembly was performed with the resulting reads using SPAdes genome assembler v3.14.1 using default parameters (Bankevich et al, 2012). Quality of the assembly was assessed using QUAST v5.0.2 (quality assessment tool for genome assemblies) (Gurevich et al, 2013). Then, both single-nucleotide polymorphisms and indel variants were called using snippy v4.6.0 (https://github.com/tseemann/snippy) with default parameters. Snippy uses FreeBayes v1.3.2 (arXiv:1207.3907)

internally to call variants, and subsequently filters variants with low support. All variants passing the default filters (minimum coverage of 10, minimum quality of 100, minimum fraction of reads supporting the alternative allele of 0.9) were considered as high confidence. Additional manual curation was then performed by aligning the resulting genome with the *M. pneumoniae* M129 reference genome available at NCBI under accession NC_000912.1. This was performed using NucDiff v2.0.3 (Khelik et al, 2017), a tool which allows comparison of closely related sequences, and rigorous analysis of local differences and structural rearrangements. Raw sequencing files and the resulting assembled genome have been deposited in ENA under the study accession number PRJEB80886 and are accessible at the link http://www.ebi.ac.uk/ena/data/view/PRJEB80886.

## Comprehensive genome annotation of the *M. pneumoniae* laboratory strain

The genome annotation was updated based on the new genome sequence (816,357 bp; Dataset EV2; deposited under accession PRJEB80886 in ENA), followed by manual curation to define the most likely protein-coding regions. For this, we used gene orthology analysis and experimental data, including transcription start sites (TSS) and peptide data collected by multiple mass spectrometry experiments (Yus et al, 2019, 2012; Miravet-Verde et al, 2019). This genome annotation includes 707 protein-coding genes and several types of functional RNAs (3 rRNAs, 1 tmRNA, 1 RNaseP RNA, 1 4.5S RNA, and 37 tRNAs) and 186 ncRNAs (Güell et al, 2009). For the analysis of the transposon data, we considered the annotation of other genomic features, such as operons ($n = 323$, considering 2 or more protein-coding genes and/or functional RNAs in the same transcriptional unit), TSS ($n = 856$) and predicted promoters ($n = 1430$) (Lloréns-Rico et al, 2015; Yus et al, 2012, 2019; Miravet-Verde et al, 2024). Transcription terminator sites (TTS, $n = 434$) were also defined as the position where an intrinsic terminator is predicted in the 500 bp downstream of a gene stop codon (Naville et al, 2011). The 5′ and 3′ untranslated regions (UTR5, $n = 483$; and UTR3, $n = 323$) were defined from the closest annotated TSS or TTS to the gene start or end, respectively. When no TTS was found, we tracked the 150 bp downstream of the related gene. The regions in between non-overlapping genes and expressed from the same operon were also tracked and referred to as inter-genic-intra-operon regions (iGiO; $n = 240$). Finally, we also included predicted ribosome binding sites (RBS, $n = 279$; included as a binary value where 1 indicates the presence of any of the possible Shine–Dalgarno sequences known to act as an RBS (Miravet-Verde et al, 2019)) located no more than 15 bp upstream of a start codon, regions with experimental signal for protein occupancy and chromatin immunoprecipitation sequencing (ChIP-seq) data for different transcriptional regulators (Yus et al, 2019) ($n = 2454$), and a set of genomic regions lacking any gene annotation after excluding all putative ORFs ("noann"; $n = 134$) (Miravet-Verde et al, 2024). Finally, 155 regions were annotated as "noexp" sharing a low transcriptional signal in both strands (i.e., $\log_2[\text{CPM}] < 3$ in both genomic strands).

All the annotations described above are detailed in Dataset EV2, including genomic coordinates and functional information, such as conservation studies and proteomic-related information, including protein copies per cell, as previously reported (Miravet-Verde et al,

2019). Motivated by the nature of this study, we also reanalyzed the RNA-sequencing data for the new version of *M. pneumoniae* M129_LSR genome using the experiment found at the ArrayExpress database under accession identifier E-MTAB-6203 (two replicates after 6 h of growth at 37 °C) to associate expression values with each genomic region considered in this study. Finally, for each of these annotations, the percentage repeated is calculated by taking as reference a set of 85,077 bp genome positions labeled as 'repeated'. This is done by running a sliding window of 21 bp, extracting the sequence, and looking for perfect matches in the direct and reverse-complementary genome sequence of *M. pneumoniae*. If the subsequence appears at least twice, the position of the window will be labeled as repeated and these positions are assumed to have low and/or limited mapping quality (Miravet-Verde et al, 2020) (Datasets EV1 and EV2).

## Transposon mutant libraries preparation

Transposon mutant libraries were prepared as previously described (Miravet-Verde et al, 2020). Briefly, *M. pneumoniae* wild-type cells were transformed by electroporation (Weber et al, 2020) with 2 μg of mini-transposon plasmid DNA pMTnCat_BDPr or pMTnCat_BDter. Mutant libraries (referred to herein "P" and "T") were selected in 5 ml cultures supplemented with chloramphenicol, and the resulting transformants were scraped off the flasks in 1 ml of fresh medium to obtain a cell stock referred to as passage 0 (P0 and T0, respectively). Transposon mutants were then serially cultured through ten consecutive passages to eliminate dead cells and further assess the impact on growth fitness of the transposon insertions represented in the libraries. For this, we inoculated 25 μl of P0 or T0 into 5 ml of Hayflick medium supplemented with chloramphenicol. After 4 days of culture, when cells reached the stationary phase (approximately ten cell divisions, Appendix Fig. S14), cells were scraped off from the flasks in the same culture medium to also recover non-adherent cells. One milliliter of this cell suspension (referred to as passage 1: P1 and T1) was used for genomic DNA isolation using the MasterPure™ DNA Purification Kit (Epicentre, Cat. No. MCD85201), and 25 μl was used to inoculate the next passage. This procedure was repeated until passage 10 (P10 and T10), including sample collection for all the intermediate passages. To account for sampling batch effects, cell passaging and sample collection were performed in duplicate.

## Transposon insertion sequencing, insertion calling, and visualizations

Illumina transposon sequencing libraries were constructed from genomic DNA of "P" and "T" mutant pools extracted at different passages, using the NEBNext Ultra DNA Library Prep kit (#E7370L) according to previously published protocols (Miravet-Verde et al, 2020). Transposon libraries were sequenced on a HiSeq 2500 platform using HiSeq v4 sequencing chemistry and $2 \times 125$ bp paired-end reads. The raw data were submitted to the ArrayExpress database (http://www.ebi.ac.uk/arrayexpress) with the accession identifiers E-MTAB-8918 (P library) and E-MTAB-14533 (T library).

The software FASTQINS was used to define the genome positions and the read count associated with each transposition

event from raw sequencing files by looking for reads containing the IR sequence *TACGGACTTTATC* (Miravet-Verde et al, 2020). The insertion points are then determined by mapping to *M. pneumoniae* M129_LSR, allowing one mismatch in the genomic sequence. The final output consists of each genome base found inserted and the read count measuring how many times each transposon insertion is mapped (Dataset EV1). General mapping was assessed by calculating each sample genome insertion coverage (i.e., genome-wide LD) as the total number of insertions found in each sample normalized by the genome length of *M. pneumoniae* M129_LSR (816,357 bp). Also, a set of known NE genes ($n = 29$; 21,599 bp in total, Dataset EV2) was used to provide an average of the maximum saturation as the total number of insertions found mapping in these genes and divided by the region covered by these same genes (Dataset EV3) (Lluch-Senar et al, 2015).

For downstream analysis, including the calculation of different metrics and the visualization of genomic insertion profiles, we employed the framework ANUBIS (Miravet-Verde et al, 2020). For these analyses, insertions with a single read count were disregarded.

## Metric and essentiality estimation by a GMM model

A reference of gene essentiality was estimated using both P1 and T1 samples and merging the two replicates of each library (PT1). To normalize differences in transposon saturation and the coverage of the sequencing technique, the top and bottom 5% of insertions in terms of read count in each sample were not considered in the merging process. Then, gene linear densities (LD) were calculated as the number of insertions mapped to a given genomic annotation normalized by its base pair length. Repeated regions where the ambiguity prevents efficient mapping of insertion were ignored, and GC preferences were corrected as done in previous studies (Miravet-Verde et al, 2020). The list of positions is referenced in Dataset EV1.

Essentiality assignment in essential (E), fitness (F1 and F2), and non-essential (NE) was performed by using the Gaussian Mixture Model from the ANUBIS package (Miravet-Verde et al, 2020). This approach applies a probabilistic estimation of groups in the data, assuming they are generated from a mixture of $k$ number(s) of Gaussian distributions. To define $k$, we took into consideration the evolution of the Akaike Information Criterion (AIC) and Bayesian Information Criterion (BIC) to find the best compromise between goodness of fit and number of parameters (Appendix Fig. S2). Also, this approach was shown to outperform other estimation methods and presents the advantage of not requiring a reference "gold" set of genes with known essentiality categories (Miravet-Verde et al, 2020). The relation between genomic annotations found in *M. pneumoniae* M129_LSR and their estimated essentiality category by analysis condition can be found in Dataset EV4.

For structural and regulatory regions, we performed the same GMM analysis using the PT1 combined sample with two and four components. As elements categorized as F1 always correspond to E in the two-components model, and the same occurs for F2, which were associated with NE, we decided to simplify the prediction to consider only E and NE classes. This is, in fact, expected due to the way the GMM algorithm minimizes differences between components (i.e., the most explanatory threshold with two components

separates E from NE, with four E + F1 from F2 + NE elements). For the manual curation and plotting in the analyses related to LD, we filtered out annotations shorter than 5 bp or having less than 25% of their sequence repeated. The complete list of selected elements, associated information, and whether they were selected or not for downstream analyses is included in Dataset EV5.

## Essentiality decay exploration and k-means clustering

Decays in LD associated with each genomic annotation were explored using a k-means clustering algorithm to group annotations based on their response to passaging (Garreta and Moncecchi, 2013). As the number of decay groups is initially unknown, we determined the minimum number of plausible clusters by inspecting the distortion values as the number of clusters increased. This distortion value is calculated for each specified number of clusters and corresponds to the sum of squared distances between observations and their dominating associated centroids. As was similarly done for assigning essentiality categories with a GMM, we selected four as the number of clusters to predict as this is the value that minimizes the distortion (i.e., sum of squared errors) and thus expected to provide a balanced compromise between goodness of fit, observed decay shapes, and number of parameters without overfitting the prediction (Appendix Fig. S4A). Furthermore, this parameter was the maximum while still retrieving clusters containing at least 50 genes, ensuring the trends were general enough. For each explored annotation (e.g., genes), we calculated a metric (e.g., LD) and its value along subsequent passages to represent the persistence of insertions in a specific annotation, or fitness impact on cell growth. This value tends to decay due to both cell death and sampling effects. Thus, we normalized values by the average found for NE positions to avoid the biases derived from the original insertion coverage rate at the sample level. Consequently, values can range from 0 to values above 1 (e.g., when an annotation is more enriched in insertions than NE regions in that same sample). In addition, the definite integral of normalized linear densities along passages 1 to 4, 6, 8, and 10, corresponding to the area covered by the decay curve (AUC), are approximated by the trapezoidal rule using the function 'trapz' from numpy (Harris et al, 2020). This value represents an integrated metric that quantifies the observed decays and is also representative of the cluster to which a gene will belong (Appendix Fig. S4B). We also examined correlations in AUC values across passage combinations from sample pairs to all passages included to ensure this metric is robust and independent of the passage conditions. In fact, Pearson's correlation coefficient between subsets was always greater than 0.85, suggesting that the same analysis could be efficiently performed with a smaller number of samples (Appendix Fig. S5).

To perform statistical assessments in the calculation of AUC differences between P and T libraries, we calculated the AUC for density/reads normalized by NE regions, separating each replicate (1 and 2) and library type (P and T). This generates a couple of measures for each condition that are then compared by a one-tailed *T*-test and assuming the differences between P and T are significant if $P < 0.05$, while a fold-change indicates the direction of the change (Dataset EV5). It is important to remark that ignoring repeated positions would inflate the apparent LDs for genes with a higher percentage of repetition, as very few numbers of positions, generally

inserted, will be considered. Thus, the k-means cluster prediction algorithm overestimates the evolution of the trajectories of these genes, affecting the definition of coherent clusters. To alleviate this, we discard these 15 genes during the clustering step: *mpn091*, *mpn128*, *mpn130*, *mpn203*, *mpn204*, *mpn367*, *mpn371*, *mpn410*, *mpn413*, *mpn463*, *mpn467*, *mpn468a*, *mpn486*, *mpn501*, and *mpn502*.

## Local essentiality estimation by a sliding window approach

Essentiality was assessed in an annotation-independent manner using a sliding window to capture potential functional units and explore small genomic loci (Zhang et al, 2012). To this end, we defined a minimum sliding window length by comparing windows of different sizes (from 3 to 150 bp) sampled from E and NE windows from genes with known essentiality (Dataset EV2), and applied a Gaussian Mixture Model to differentiate between E and NE populations of windows in an unsupervised manner. Using windows sampled from the reference gold set of genes, we observed that the probability of wrongly assigning the NE category to a window from an E gene is less than 5%, when using a window size of 31 bp (Appendix Fig. S15). Then, we extracted different metrics associated with each 31 bp region in the *M. pneumoniae* M129_LSR genome and defined NE domains by considering contiguous windows together when they were classified as NE. Unassigned nucleobases were labeled as E (Dataset EV6). To make the analysis stricter, we only considered DNA domains covering ≥ 5 bp and with less than 10% of the segment repeated.

To identify possible NE regions within E genes, we mapped predicted domains against different annotations to obtain both the overlap (general) and the percentage they represent relative to gene annotations (Dataset EV7). DNA domains with different essentiality profiles were also predicted by separating P and T libraries, in addition to computing AUC values (passages 1 to 8) for the predicted domains. For measuring the extensions from an evolutionary perspective, we ran BLASTP against 109 bacterial reference genomes (Miravet-Verde et al, 2019) and counted the number of additional or reduced amino acids in each gene based on the alignment positions (Dataset EV7).

To assess possible differences in transposon coverage across DNA domains known to bind to transcriptional repressors or other DNA-binding proteins (POD data previously published) (Yus et al, 2019), we calculated different metrics (number of insertions, LD, total read count and average number of reads per insertion) for the length of domain under inspection and compared to the same number of bp before and after the domain (Datasets EV8, EV9).

## Transposon-insertion enrichment analysis

To identify insertions presenting a selected enrichment after 10 passages, we only considered insertions that were conserved in passages 1 and 10 in both replicates for each P and T condition (considered as "valid" positions in the enrichment analysis). Then, we took the read count observed in each condition and replicate and applied a TMM (trimmed mean of *M*-values) normalization to allow grouping by passage and enable comparison. The TMM method is commonly used to normalize sequencing read counts in RNA-Seq experiments and to ensure that the overall distribution of read counts is comparable across samples by removing biases that may arise due to differences in library sizes and gene-specific effects. It is important to note that TMM normalization assumes that most insertions are not differentially selected across samples. With the normalized read counts, we calculated the fold-change observed between the last and first passage and assessing the statistical differences by the *p* value given by a one-tail *T*-test and corrected by the Bonferroni method (Dataset EV10). These insertion positions were analyzed in relation to the annotations in Dataset EV2 and assigned in one or more of the following categories: overlap E/F1 gene, overlap F2/NE gene, upstream E/F1 gene, upstream F2/NE gene, overlap UTR5, upstream UTR5, overlap with iGiO, overlap with no transcribed region, between two gene ends, and not classified (not present in any of the previous; Dataset EV10).

Finally, for insertions located upstream of a gene, we also incorporated gene expression information recalculated for the updated *M. pneumoniae* M129_LSR genome with the E-MTAB-6203 dataset and COG categories (Dataset EV11). For statistical tests in the COG enrichment, after grouping genes by COG category, we counted the number of genes that are affected and not affected by an enriched insertion located up to 250 bp upstream of their start codon. Then, we calculated the enrichment *p* value for each COG category using a hypergeometric test and adjusted it using the Benjamini–Hochberg method (Dataset EV12).

# Data availability

Raw sequencing files corresponding to the promoter library (P) can be accessed in the ArrayExpress database at EMBL-EBI, under accession number E-MTAB-8918, and are accessible from the following link: http://www.ebi.ac.uk/arrayexpress/experiments/E-MTAB-8918. Raw sequencing files corresponding to the terminator library (T) have been deposited in the same database under accession number E-MTAB-14533 and are accessible from the following link: http://www.ebi.ac.uk/arrayexpress/experiments/E-MTAB-14533.

The genome assembly of *M. pneumoniae* M129_LSR strain can be found at the European Nucleotide Archive (ENA) under the study accession number PRJEB80886 (http://www.ebi.ac.uk/ena/data/view/PRJEB80886) and associated samples SAMEA116129577, SAMEA116129576, SAMEA116129575, SAMEA116129574, and SAMEA116129573.

The bioinformatic functions required to reproduce the presented analyses and figures are part of the analysis framework ANUBIS, available via https://github.com/CRG-CNAG/anubis.

The source data of this paper are collected in the following database record: biostudies:S-SCDT-10_1038-S44320-025-00133-1.

# Peer review information

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

## Acknowledgements

We acknowledge the support provided by the Genomics Unit at the Centre for Genomic Regulation sequencing the samples. This project has received funding from the European Research Council (ERC) under the European Union's Horizon 2020 research and innovation program MYCOCHASSIS (670216) and ERC LUNG-BIOREPAIR (101020135) ERC Advanced Grants (AdG). We also acknowledge support of the Spanish Ministry of Science and Innovation through the Centro de Excelencia Severo Ochoa (CEX2020-001049-S, MCIN/ AEI/10.13039/501100011033), the Generalitat de Catalunya through the CERCA program, and the EMBL partnership. We are grateful to the CRG Core Technologies Program for their support and assistance in this work.

## Author contributions

**Samuel Miravet-Verde**: Conceptualization; Data curation; Software; Formal analysis; Validation; Investigation; Visualization; Methodology; Writing—original draft; Project administration; Writing—review and editing. **Raul Burgos**: Conceptualization; Data curation; Formal analysis; Validation; Investigation; Visualization; Methodology; Writing—original draft; Project administration; Writing—review and editing. **Eva Garcia-Ramallo**: Validation. **Marc Weber**: Formal analysis; Investigation. **Luis Serrano**: Conceptualization; Resources; Supervision; Funding acquisition; Investigation; Methodology; Writing—original draft; Project administration; Writing—review and editing.

Source data underlying the figure panels in this paper may have individual authorship assigned. Where available, figure panel/source data authorship is listed in the following database record: biostudies:S-SCDT-10_1038-S44320-025-00133-1.

## Disclosure and competing interests statement

LS is the cofounder of Pulmobiotics SL, and RB is currently an employee of this company. MW is currently an employee of Flomics Biotech SL. LS is also a member of the Editorial Advisory Board of Molecular Systems Biology. This has no bearing on the editorial consideration of this article for publication.

# Expanded View Figures

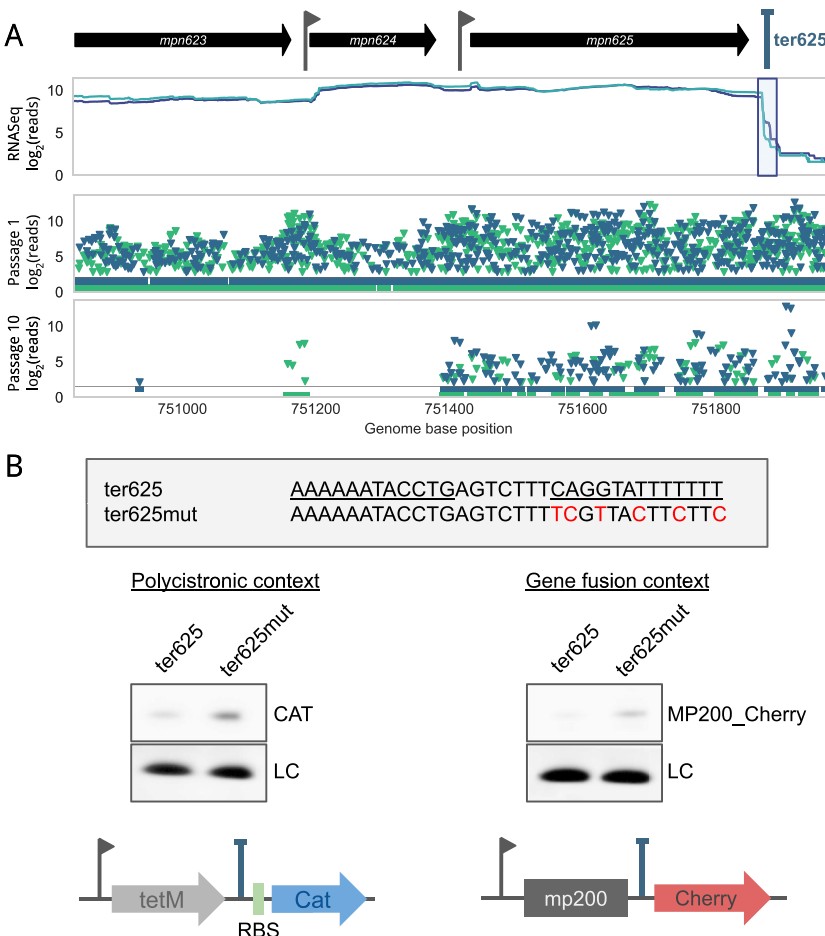

**Figure EV1.  Genomic context and activity of the endogenous ter625 terminator.**

(**A**) Scheme showing the position of the ter625 terminator sequence (downstream blue box, with a blue vertical line representing the start of the hairpin) in its natural genomic context. Genes appear as black arrows, and gray flags indicate the position of predicted promoters and TSS. Below is shown the RNAseq expression profile ($n = 2$), and transposon insertion mapping at passages 1 and 10 are shown along the genomic region. Green and blue triangles represent transposon insertions containing promoters or terminators, respectively. Note the enrichment of transposons containing terminators after ten passages next to the position of the endogenous ter625 terminator. (**B**) Termination activity of the ter625 sequence in two different genetic contexts using gene reporters expressed as polycistronic or fusion transcripts. On top of the panel is shown the wild-type ter625 sequence compared to a mutated version affecting the terminator hairpin structure (underlined). The expression of *cat* or *cherry* gene reporters was assessed by Western blot analysis in the two genetic configurations shown below the panel. The gray flags indicate the promoter of the transcriptional unit, while the position of the ter625 sequence is shown as a blue T, just before a ribosome binding site (RBS, left panel) or after the mp200 polypeptide fusion (right panel).

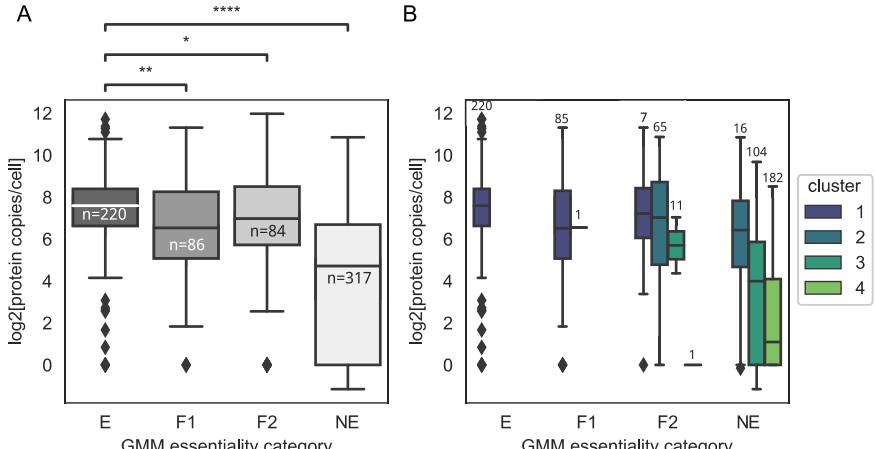

**Figure EV2. Box plots showing the relation between GMM class and k-means clusters in relation to protein copy numbers.**

(A) For each predicted essentiality category at PT1 (X-axis), we relate the log₂-transform copies per cell of the proteins in each group (Y-axis). E genes present significantly higher copies/cell levels than F1, F2, and NE genes ('*' Mann–Whitney $P < 0.05$; '****' Mann–Whitney $P < 0.0001$). NE genes present the lowest copies per cell. Box plots show the median (center line), the 25th and 75th percentiles (box bounds), and the minimum and maximum values (whiskers), or display outliers as diamonds, following the default settings of the Seaborn's boxplot function. The sample size of each category is labeled within the box. (B) Same representation as panel (A) but separating genes by the cluster assigned by the k-means applied on the LD decays between PT1 and PT8. It can be observed that F2 and NE genes assigned to higher clusters (*i.e.*, more stable insertions) present lower copies per cell. Box plots show the median (center line), the 25th and 75th percentiles (box bounds), and the minimum and maximum values (whiskers), or display outliers as diamonds, following the default settings of the Seaborn's boxplot function. On top of each box, the sample size is labeled. Note that for the clustering analysis, we excluded 15 NE genes with a high percentage of sequence repetition, as these interfere with the clustering prediction (see Methods). In addition, we assigned a value of 0.0 copies/cell to proteins not detected (labeled as "-" in Dataset EV2).

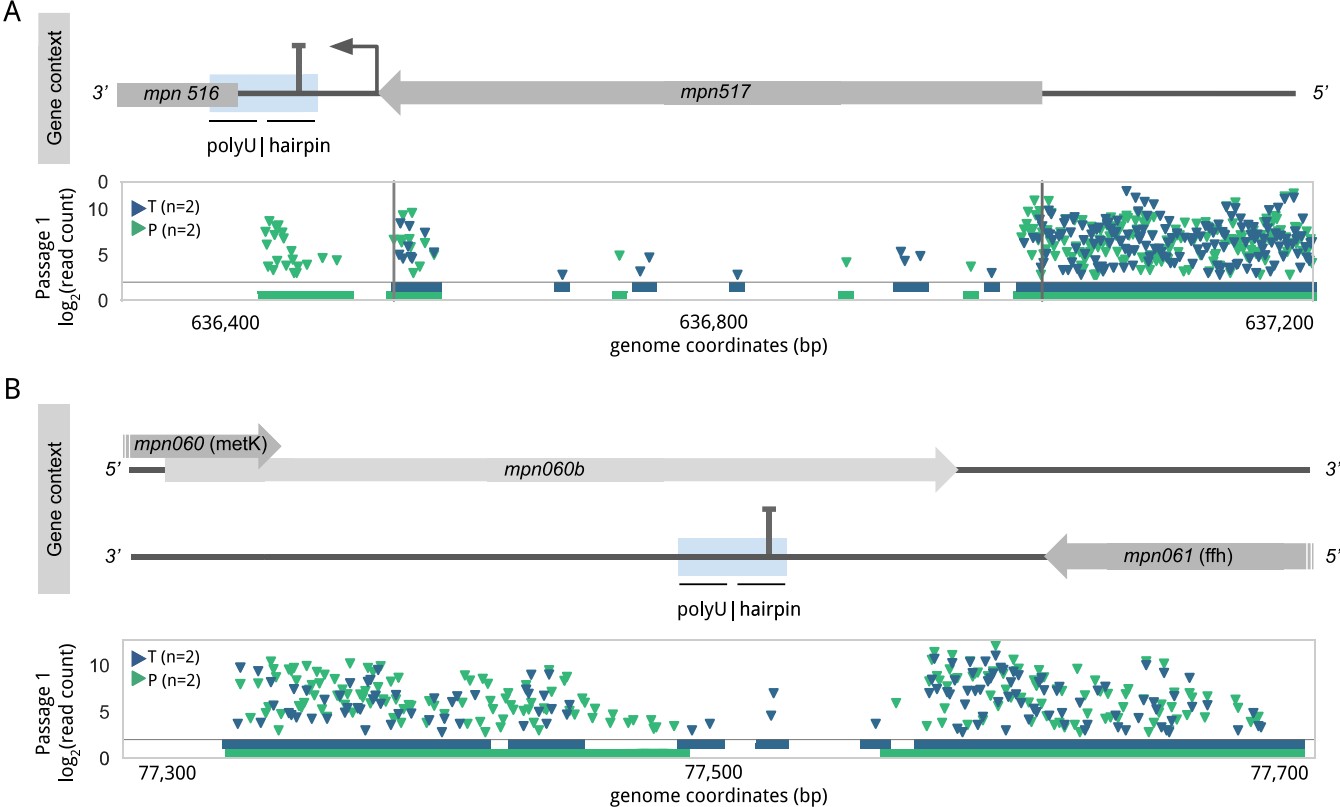

**Figure EV3.   Examples of essential 3′UTR and terminator signals.**

Representation of the insertion profiles and genomic context of (**A**) *mpn517* and (**B**) *mpn061*, two genes presenting essential 3′UTRs. For each panel, the insertion profile of the region of interest is shown. Transposons containing promoters or terminators are depicted in blue and green triangles, respectively. The genetic context is also shown on top of each panel, including the orientation of the genetic elements. Predicted intrinsic terminator sequences are labeled in blue boxes, showing hairpin and poly-U sequences. The 3′UTR are defined from stop codons to the terminators. It should be noticed that the 3′UTR region of *mpn517* partially overlaps with the 5′UTR of *mpn516*, which is essential in the T library but not in the P library, thus making it difficult to assess whether it is essential the terminator or the 5′UTR.

Local insertion frequency of regulatory elements associated to:

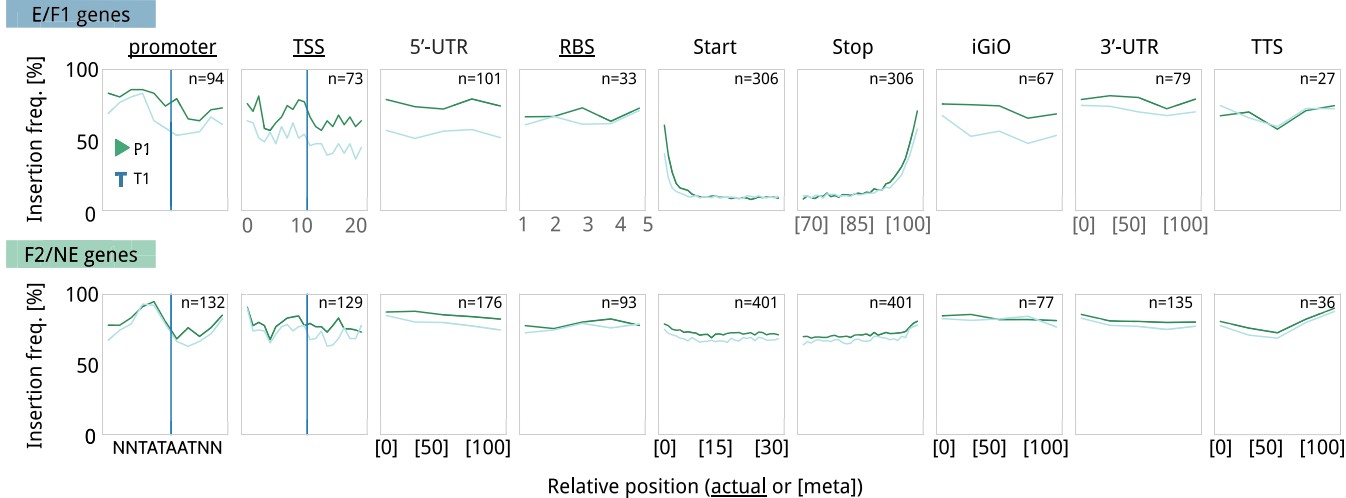

**Figure EV4. Local transposon insertion frequency in different regulatory elements.**

The local frequency of transposon insertions containing promoters (green) or terminators (blue) at passage 1 is shown for different regulatory elements (including start and stop codons) associated with E/F1 (top panel) or F2/NE genes (bottom panel). Regulatory elements with a fixed sequence length are underlined and have a solid blue line representing the center of the element. For regulatory elements with a variable length, the X-axis represents the relative percentage of the covered region (with a minimum of 5 bins for regulatory elements and 100 bins for genes, and then we represent the first 30 and last 30 bins to get the start and stop regions). While no difference is observed for F2/NE genes, E and F1 genes are characterized by an enrichment of transposons containing promoters in upstream regulatory elements, except for RBS. Downstream regulatory elements (3′-UTR and TTS) do not show significant global differences regardless of the essentiality of the associated gene. Note that the N- and C-terminal regions of E genes differ in the length of transposon disruption tolerance, with C-terminal regions showing a higher tolerance for extended regions.

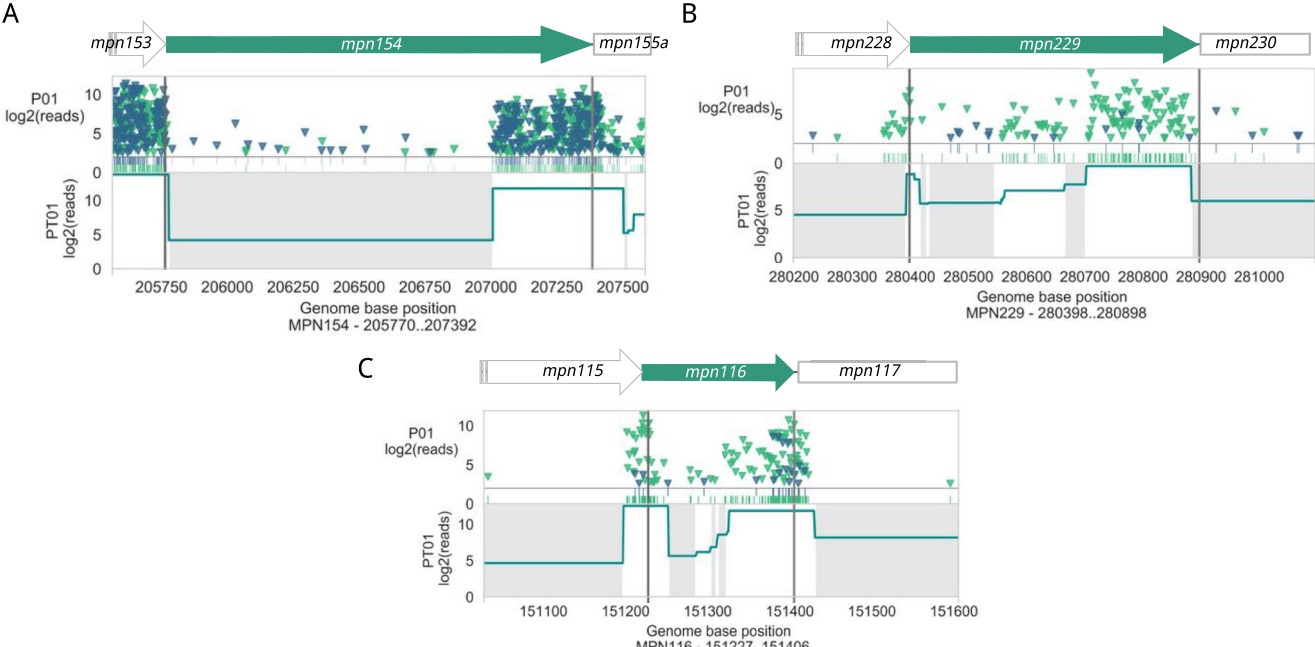

**Figure EV5. Examples of extended termini regions in E genes.**

Panels correspond to genes *mpn154* (**A**), *mpn229* (**B**), and *mpn116* (**C**). For each panel, the insertion profile of the region of interest at passage 1 is shown. Transposons containing promoters or terminators are depicted in blue and green triangles, respectively. Vertical lines show start and stop codons of the gene, also including their genomic context on top.

