## [Peer Review File · Molecular Systems Biology]

Quantitative essentiality in a reduced genome: a functional, regulatory and structural fitness map

Samuel Miravet-Verde, Raul Burgos, Eva Garcia-Ramallo, Marc Weber, and Luis Serrano

Corresponding author(s): Luis Serrano (luis.serrano@crg.eu), Samuel Miravet-Verde (smiravet@ethz.ch), Raul Burgos (raul.burgos@pulmobio.com)

Review Timeline:

Submission Date:	5th Feb 25
Editorial Decision:	3rd Mar 25
Revision Received:	30th May 25
Editorial Decision:	3rd Jul 25
Revision Received:	8th Jul 25
Accepted:	12th Jul 25

Editor: Jingyi Hou

Transaction Report:

3rd Mar 2025

Manuscript Number: MSB-2025-12901

Title: Quantitative essentiality in a reduced genome: a functional, regulatory and structural fitness map

Author: Samuel Miravet-Verde

Raul Burgos

Eva Garcia-Ramallo

Marc Weber

Luis Serrano

Dear Luis,

Thank you for submitting your work to Molecular Systems Biology. We have now heard back from the three reviewers who agreed to evaluate your manuscript. As you will see from the reports below, the reviewers find the study interesting and relevant. They raise, however, a series of concerns, which we would ask you to address in a major revision.

I think the reviewers' recommendations are relatively clear, so there is no need to reiterate the points listed below. All the issues raised by the reviewers need to be satisfactorily addressed. As you may already know, our editorial policy allows in principle a single round of major revision, and it is therefore essential to provide responses to the reviewers' comments that are as complete as possible. Please feel free to contact me in case you would like to discuss in further detail any of the issues raised by the reviewers.

On a more editorial level, we would ask you to address the following issues:

- Please provide a .docx formatted version of the manuscript text (including legends for main figures, EV figures and tables). Please make sure that the changes are highlighted to be clearly visible.
- Please provide individual production quality figure files as .eps, .tif, .jpg (one file per figure).
- Please provide a .docx formatted letter INCLUDING the reviewers' reports and your detailed point-by-point responses to their comments. As part of the EMBO Press transparent editorial process, the point-by-point response is part of the Review Process File (RPF), which will be published alongside your paper.
- Please note that all corresponding authors are required to supply an ORCID ID for their name upon submission of a revised manuscript.
- We replaced Supplementary Information with Expanded View (EV) Figures and Tables that are collapsible/expandable online (see examples in <http://msb.embopress.org/content/11/6/812>). A maximum of 5 EV Figures can be typeset. EV Figures should be cited as 'Figure EV1, Figure EV2' etc... in the text and their respective legends should be included in the main text after the legends of regular figures.

Additional Tables/Datasets should be labeled and referred to as Table EV1, Dataset EV1, etc. Legends have to be provided in a separate tab in case of .xls files. Alternatively, the legend can be supplied as a separate text file (README) and zipped together with the Table/Dataset file.

For the figures and tables that you do NOT wish to display as Expanded View figures, they should be bundled together with their legends in a single PDF file called *Appendix*, which should start with a short Table of Content. Each legend should be below the corresponding Figure/Table in the Appendix. Appendix figures and tables should be referred to in the main text as: "Appendix Figure S1, Appendix Figure S2, Appendix Table S1" etc. See detailed instructions regarding expanded view here: <https://www.embopress.org/page/journal/17444292/authorguide#expandedview>.

- Before submitting your revision, primary datasets (and computer code, where appropriate) produced in this study need to be deposited in an appropriate public database (see <http://msb.embopress.org/authorguide-dataavailability> <https://www.embopress.org/page/journal/17444292/authorguide#dataavailability>). Please remember to provide a reviewer password if the datasets are not yet public. The accession numbers and database should be listed in a formal "Data Availability" section (placed after Materials & Method) that follows the model below (see also <https://www.embopress.org/page/journal/17444292/authorguide#dataavailability>). Please note that the Data Availability Section is restricted to new primary data that are part of this study.

Data availability

- RNA-Seq data: Gene Expression Omnibus GSE46843 (<https://www.ncbi.nlm.nih.gov/geo/query/acc.cgi?acc=GSE46843>)
 - [data type]: [name of the resource] [accession number/identifier/doi] ([URL or identifiers.org/DATABASE:ACCESSION])
- *** Note - All links should resolve to a page where the data can be accessed. ***

-At EMBO Press we ask authors to provide source data for the main figures. Our source data coordinator will contact you to discuss which figure panels we would need source data for and will also provide you with helpful tips on how to upload and organize the files.

- Our journal encourages inclusion of *data citations in the reference list* to directly cite datasets that were re-used and obtained from public databases. Data citations in the article text are distinct from normal bibliographical citations and should directly link to the database records from which the data can be accessed. In the main text, data citations are formatted as follows: "Data ref: Smith et al, 2001". In the Reference list, data citations must be labeled with "[DATASET]". A data reference must provide the database name, accession number/identifiers and a resolvable link to the landing page from which the data can be accessed at the end of the reference. Further instructions are available at .

- We updated our journal's competing interests policy in January 2022 and request authors to consider both actual and perceived competing interests. Please review the policy <https://www.embopress.org/competing-interests> and update your competing interests if necessary.
Please use the heading "Disclosure statement and competing interests".

- All Materials and Methods need to be described in the main text using our 'Structured Methods' format. According to this format, the Methods section includes a Reagents and Tools Table (listing key reagents, experimental models, software and relevant equipment and including their sources and relevant identifiers) followed by a Methods and Protocols section describing the methods, ideally using a step-by-step protocol format. The aim is to facilitate adoption of the methodologies across labs. Please download and fill our Reagents and Tools Table template (.docx), which you can find in our author guidelines: <https://www.embopress.org/page/journal/17444292/authorguide#structuredmethods>.

An example of a Method paper with Structured Methods can be found here:
<https://www.embopress.org/doi/10.15252/msb.20178071>.

- Regarding data quantification:

Please ensure to specify the name of the statistical test used to generate error bars and P values, the number (n) of independent experiments (please specify technical or biological replicates) underlying each data point and the test used to calculate p-values in each figure legend. Discussion of statistical methodology can be reported in the materials and methods section, but figure legends should contain a basic description of n, P and the test applied.

Graphs must include a description of the bars and the error bars (s.d., s.e.m.).

- Please provide a "standfirst text" summarizing the study in one or two sentences (approximately 250 characters, including space), three to four "bullet points" highlighting the main findings and a "synopsis image" (550px width and 400-600 px height, PNG format) to highlight the paper on our homepage.

Here are a couple of examples:

<https://www.embopress.org/doi/10.15252/msb.20199356>

<https://www.embopress.org/doi/10.15252/msb.20209475>

<https://www.embopress.org/doi/10.15252/msb.209495>

When you resubmit your manuscript, please download our CHECKLIST (<https://www.embopress.org/pb-assets/embo-site/EMBO%20Press%20Author%20Checklist-1642513524327.xlsx>) and include the completed form in your submission.

Please note that the Author Checklist will be published alongside the paper as part of the transparent process (<https://www.embopress.org/page/journal/17444292/authorguide#transparentprocess>).

If you feel you can satisfactorily deal with these points and those listed by the referees, you may wish to submit a revised version of your manuscript. Please attach a covering letter giving details of the way in which you have handled each of the points raised by the referees. A revised manuscript will be once again subject to review and you probably understand that we can give you no guarantee at this stage that the eventual outcome will be favorable.

I look forward to receiving your revised manuscript soon.

Kind regards,
Jingyi

Jingyi Hou, PhD
Senior Editor
Molecular Systems Biology

We realize that it is difficult to revise to a specific deadline. In the interest of protecting the conceptual advance provided by the work, we recommend a revision within 3 months (1st Jun 2025). Please discuss the revision progress ahead of this time with the editor if you require more time to complete the revisions.

IMPORTANT: When you send your revision, we will require the following items:

1. the manuscript text in LaTeX, RTF or MS Word format
2. a letter with a detailed description of the changes made in response to the referees. Please specify clearly the exact places in the text (pages and paragraphs) where each change has been made in response to each specific comment given
3. three to four 'bullet points' highlighting the main findings of your study
4. a short 'blurb' text summarizing in two sentences the study (max. 250 characters)
5. a 'thumbnail image' (550px width and max 400px height, Illustrator, PowerPoint or jpeg format), which can be used as 'visual title' for the synopsis section of your paper.
6. Please include an author contributions statement after the Acknowledgements section (see <https://www.embopress.org/page/journal/17444292/authorguide>)
7. Please complete the CHECKLIST available at (<https://bit.ly/EMBOPressAuthorChecklist>). Please note that the Author Checklist will be published alongside the paper as part of the transparent process (<https://www.embopress.org/page/journal/17444292/authorguide#transparentprocess>).
8. When assembling figures, please refer to our figure preparation guideline in order to ensure proper formatting and readability in print as well as on screen:
<https://bit.ly/EMBOPressFigurePreparationGuideline>
See also figure legend guidelines: <https://www.embopress.org/page/journal/17444292/authorguide#figureformat>
9. Please note that corresponding authors are required to supply an ORCID ID for their name upon submission of a revised manuscript (EMBO Press signed a joint statement to encourage ORCID adoption). (<https://www.embopress.org/page/journal/17444292/authorguide#editorialprocess>)
Currently, our records indicate that the ORCID for your account is 0000-0002-5276-1392.

Link Not Available

11. Include a Reagents and Tools Table as part of the Methods section, which can be downloaded from our author guidelines (<https://www.embopress.org/page/journal/17444292/authorguide#structuredmethods>)

*** PLEASE NOTE *** As part of the EMBO Press transparent editorial process initiative (see our Editorial at <https://dx.doi.org/10.1038/msb.2010.72>), Molecular Systems Biology publishes online a Review Process File with each accepted manuscripts. This file will be published in conjunction with your paper and will include the anonymous referee reports, your point-by-point response and all pertinent correspondence relating to the manuscript. If you do NOT want this File to be published, please inform the editorial office at msb@embo.org within 14 days upon receipt of the present letter.

Reviewer #1:

The manuscript "Quantitative essentiality in a reduced genome: a functional, regulatory and structural fitness map" by Miravet-Verde et al is really very interesting expanded and enhanced TnSeq Method with two different libraries for analyzing the essentiality of genes, genome regions and even protein domains encoded by gene

fragments of the genome-reduced bacterium *Mycoplasma pneumoniae* at a high resolution. The authors included a number of passages which selected for growth to add a temporal aspect to their analysis, allowing to follow the fitness contributions of the different elements affected by transposons over time. To analyze their data the authors successfully added and validated different tools and approaches also to categorize the essentiality of their genes including encoded RNA species and genome fragments which can present binding sites for various DNA binding proteins, including transcription factors, RNA Pol, but also SMC proteins or DnaA.

The results of the manuscript suggest an improved appreciation of more complex and more variable properties and characteristics of "essential" genes, depending for example also on different genetic and/or environmental conditions.

The genome-reduced bacterium *Mycoplasma pneumoniae* is of course a kind of special organism, however the authors suggest that their new method should also be applicable to other model organisms. I am wondering whether one could also add different stress conditions instead or during the here applied passages, to study the effect of e.g. heat, cold or antibiotic stress on the distribution, changes or appearances of e.g. new "essential" genes under such conditions.

Minor comments

P11 "affected by the binding of DNA proteins" What are "DNA proteins"?

P12 "the Tn-seq data is sensitive to strong DNA-protein interactions" I would not state that "data" is sensitive. Maybe there could rather be a "competition" for DNA binding? Or a (recognition site) specific interference by DNA binding proteins?

Reviewer #2:

Manuscript Title: "Quantitative essentiality in a reduced genome: a functional, regulatory and structural fitness map"

OVERVIEW

This manuscript presents a novel TnSeq approach to studying gene essentiality in the reduced-genome bacterium *Mycoplasma pneumoniae*, combining two transposon libraries to achieve near single-nucleotide resolution mapping in non-essential coding regions of the genome. The work demonstrates how this method can provide insights into genome organization and function, with implications for synthetic biology and genome engineering.

STRENGTHS

1. Novel dual-library methodology achieving high-resolution resolution and investigation of impact activating and repressing insertions
2. Comprehensive analysis of gene essentiality in a minimal genome organism

REVIEWER COMMENTS

- Page 1: The claimed "near-single-nucleotide precision" needs qualification as resolution varies across genome
- Page 3: The authors acknowledge that "the mariner transposable elements depend on TA dinucleotide targets, limiting the resolution" but don't discuss how this bias is addressed by their method. What is the insertion bias of Tn4001 transposase? How might this impact results?
- Page 5: It is unclear how the 55.6% of genome coverage is defined? Is there a maximum distance between insertions?
- Page 7: K-means requires the setting of cluster numbers. I see a discussion of this in the methods section, but I would make a statement about how this was selected (supplementary fig appears to imply "elbow method" but this is not stated specifically). Why do you say "at least four different clusters"? If you could have used a larger k, what is the implication of picking k=4?
- Page 7: What do you mean by "copy number" of genes? You then refer to this as "protein abundance". How is this measured? If this is an issue, might it not indicate that T1 (~10 cell divisions) is too early to accurately evaluate gene essentiality?
- Page 10: It was initially unclear the sliding window analysis included entire genome and not just "non-annotated regions". How was 31-bp for sliding window selected?
- Page 13: Domain boundaries were defined "through visual inspection of the transposon data" from 16 genes seems subjective and may lack reproducibility. Can you provide a figure to explain method or reference existing figure.
- Page 16: The manuscript mentions COG categories G, D, T, and S in the enrichment analysis, but doesn't provide specific examples of genes/pathways discovered. I am curious what fraction of these are regulators. Please provide some specific examples of genes/pathways.
- Page 17: You state that you work "supports the concept that gene essentiality is a conditional trait that depends on the genetic and environmental context". I don't understand how these specific claims are supported. Only 1 cultivation environment was used and strains were all single KOs (same genetic context). I think this method could be extremely useful in evaluating this in the future.
- Page 20: You claim that that this strategy will be a "very informative and useful to guide genome annotations" but there is no discussion of its application in other strains. How versatile is this system?
- Figure 2C: Figure 2C shows that a large fraction of insertion positions has just 1 read, which seems questionable. These could be artefacts of the TnSeq protocol, or, more likely, remnants of low-abundance non-growing strains. It looks there are many of these within the should-be-essential genes, consistent with either interpretation. So it seems that the linear density should be computed without these 1-read insertions. On a related note, Figure S4B shows that the typical essential protein has insertions at ~15% of its locations, which seems way too high. We suspect that this would drop to a much lower and more reasonable

number if the just-1-read cases were dropped.

A few general comments:

The article uses 3 different operational measures of essentiality! It's a bit confusing, and is it useful to have so many of them? The area under a curve as a measure of fitness looks good, why not use just that?

1 passage = 4 days = ~10 doublings -- so I'm wondering if some of the "essential" genes (with very few insertions after 1 passage) are not quite essential. The authors do have a "known" non-essential set (29 genes) to test on, which is reassuring, but it is not clear if this issue is fully addressed.

A prior study on *Caulobacter* (Christen et al) had ~0.1 insertion per bp. Is ~1 insertion per bp so much better? It seems likely that some of the results wouldn't have been obtained with "only" 0.1 insertion/bp. This should be explicitly addressed in the Discussion.

"Overall, these observations highlight the importance of assessing essentiality at longer growth time points." -- An N-terminal part of MPN119 does appear to be important for fitness, but it may not be exactly essential -- it's probably a subtle growth defect (starting to be apparent after 2 passages ~ = 20 generations).

"Although this C-terminal extension [of NusA] was previously established as essential, our data indicates it is fully disruptable at passage 1" -- The cited reference has a targeted genetic assay. Is there a reason why it found essentiality and the pooled random approach didn't?

Getting back to a comment above: The Methods reports that Figure 6C uses a false discovery rate approach to adjust for multiple testing. This is fine, but it was not apparent from either the figure itself or from the text of the results. It would be better to see some examples of genes with the strongest evidence of a fitness benefit from increasing their expression level, rather than something so abstract as a COG enrichment test.

TYPOGRAPHICAL ERRORS:

- Figure 2C: "Fig. S2C shows a large transcription unit mainly constituted by E ribosomal genes" -- that should be Fig. 2C
- Figure 5: log₂ reads has a different scale, are there no instances of insertions with just (say) 1-2 reads? Or are those filtered out from this display?
- Page 4: "To eliminate death cells"
- Page 6: "Functional RNAs such as ribosomal RNAs (MPNr01, MPNr02, MPNr03), MPNs01 (4.5S RNA), MPNs03 (RNaseP) and MPNs04 (tmRNA), they were all essential for survival"
- Page 13: "transposons carrying promoters identified predicted longer N- and C-terminal"
- Page 13: "For example we o found cases like mpn229"
- Page 20: "We show that the leverage of Tn-Seq data"

CONCLUSION

The manuscript presents a valuable approach to studying gene essentiality with high resolution in a minimal genome. With clearer explanation of the analytical choices, this work could make a significant contribution to our understanding of bacterial genome function and the definition of essential genes beyond binary classifications.

Reviewer #3:

Manuscript Reference: MSB-2025-12901

Authors: Samuel Miravet-Verde, Raul Burgos, Eva Garcia-Ramallo, Marc Weber, and Luis Serrano

Title: Quantitative essentiality in a reduced genome: a functional, regulatory and structural fitness map

This paper improves the random Tn insertion method, a powerful approach for analyzing the essentiality of bacterial genomes, by refining the structure of Tn to be used and the analysis methods. Previous Tn insertion analyses have focused on coding regions, where their essentiality has been debated. In this paper, the authors advance their analysis to include non-coding regions by enhancing the Tn structure and methodology.

The improvement of the Tn structure allows for the insertion of promoters or terminators adjacent to the IR region within the Tn, enabling transcription in both directions regardless of the insertion orientation, while also terminating transcription of any transcript originating from, or passing through the transposon sequence. Using these Tns, the authors conducted a quantitative

analysis with high resolution at the bp level, including non-coding regions, and obtained time-course data with quantitative measures. After standardizing the data, they applied unsupervised k-means clustering for further analysis.

This manuscript is a well-considered study, and considering the reliability of the data, it is believed to provide significant benefits to the readers. I would like you to consider the following questions and comments.

The analysis of this manuscript is a well-considered study, and I believe it provides significant benefits to readers, including the reliability of the data. However, before making a final judgment, I would like the authors to consider the following questions and comments.

Global Points:

Regarding mutations introduced into the genome through Tn insertion or homologous recombination, it is inevitable that differences in physical conditions due to genomic positions or sequences, as well as variations in interactions and modifications with or by proteins, will affect the efficiency of these insertional mutations. One of the standardization in these types of analysis could involve calculating fold changes at the same insertion site between control and samples. This raises the question of whether it is truly possible to compare quantitatively across genomic positions. Differences are also clearly observed in the integration of fragments into the genome via homologous recombination and in amplification efficiency by colony PCR.

Although standardized quantitative data from insertion sites are presented, it may initially seem that such issues do not exist. While detailed analyses and discussions regarding the validity of standardization have been conducted in the past, this time, as the authors were focusing on genomic regions different from the coding regions, I feel it is crucial to consider whether the previous standardization functions equally well functioned. In particular, interactions with other proteins may require more thorough investigation than the coding regions.

Has there been a significant accumulation of data regarding the characteristics of insertion patterns in raw data for both coding and non-coding regions, and whether the same standardization is appropriate? I believe this is a problem that needs to be kept in mind at all times.

Additionally, Tn transposition may not only occur during the log phase of cell growth. While differences in efficiency may exist, transposition is likely to occur even during the stationary phase. What are your thoughts on this? Moreover, in the log phase, the average number of DNA molecules per cell due to genomic positions may differ due to replication; does the raw data reflect such a gradient? It seems that there are still hidden issues.

While it is stated that the absence of Tn insertion does not definitively indicate essentiality, understanding how to make this judgment necessitates considering the presence of proteins that interact with that region, which is well understood and I completely agree. Homologous recombination also shows varying recombination efficiencies based on genomic positions, though it is the case using different bacteria. Differences in PCR amplification efficiencies depending on the genomic position are also observed, and these all seem to stem from common causes. This leads to the possibility that nucleoid structure may also play a role. Has there been an observation made without normalizing insertion frequencies when observing the entire genome? Are there any frequency variations or patterns according to the genomic positions?

Minor Points:

1) Page 4, line 5: Is there a possibility that in the strains with inserted P-type Tn, transcription of downstream genes continues while simultaneously inhibiting transcription of upstream genes? This depends on the orientation of the upstream genes, but particularly in regions with operon structures containing multiple genes, it can be considered that there is a possibility of inhibition for genes that are transcribed in the same orientation as the disrupted gene.

2) Page 5, line 4; If the correlation between P and T is very high, does it mean that even with promoters and terminators inserted at both ends, their overall effect is not significantly influenced?

3) Page 6, first paragraph; In the classification by anti-codon, wouldn't it be clearer to categorize which amino acids have multiple tRNAs and classify them as essential or non-essential? For instance, presenting a table of the codon table for Mycoplasma alongside the number of tRNA genes might clarify the explanation. As it stands, it is a bit hard to understand.

4) Page 6, last two sentences; It is difficult to simply infer whether a certain target gene is dispensable or not. It is necessary to consider the impacts of genetic interactions. While gene may be dispensable in isolation, there are many cases where it becomes essential when combined another.

5) Page 7, line 23: From SupFig7, it appears that the average intracellular copy number of essential gene products seems higher than that of non-essential genes. Is this considered to be specific to Mycoplasma or it is quite common between any microbes analyzed this type of analysis? It is thought that some essential genes may exhibit lethality when overexpressed. Alternatively, could the presence of genes that exhibit lethality when overproduced, such as those requiring large amounts of proteins like ribosomes, obscure this trend? Generally, do most bacterial species analyzed for essentiality and their intracellular product amounts show the same trend?

6) Page 11, line 11; Is it a mistake to refer to mpn688 instead of mpn668? In Fig. 3B, it is indicated as mpn668. Additionally, as gene names are mentioned in the text, including gene names in the figures would help readers. This applies to other figures as

well. However, care must be taken not to make the figures too cluttered.

7) Page 11, line 14 - ; Regarding the essentiality of DnaA boxes, if ncRNA is clearly expressed from this region, it should be discussed alongside the expression of ncRNA. It is conceivable that replication initiation may occur not only due to the relaxation of the double helix caused by DnaA binding but also due to the opening of the double strand caused by ncRNA transcription. Alternatively, has this discussion already been resolved in the past? If replication initiation points are already determined, the positional relationship between those points and the transcription start points of ncRNA is also expected to yield important suggestions. If this has been established, relevant citations are necessary.

8) Page 12, line 15, and Fig. 3C; The target indicated by the green arrow needs improvement. Shouldn't it point to the green box?

9) Page 12, line 18; In this regard, a discussion of the positional relationship between the HrcA binding site and the promoter should be included. Additionally, even if the binding site and the promoter are somewhat separated by Tn insertion, I think there is a possibility that HrcA could still maintain its function as a transcription factor.

10) Page 12, line 24 - ; The statement that the absence of Tn insertion does not definitively indicate essentiality suggests that the presence of proteins interacting with that region needs to be considered, which I understand well and agree with. Not only in Tn insertion but also in homologous recombination, different bacteria exhibit variations in recombination efficiency based on genomic positions. Furthermore, amplification efficiencies by colony PCR method also show variations based on genomic positions. I feel that all of these stem from a common cause. While the structure of the nucleoid is thought to have an influence, when observing the entire genome without normalizing the insertion frequency, wasn't there an observation of frequency variation dependent on genomic positions?

11) Page 14, line 1; The phrase "Due to its unusual genetic code" is hard to grasp. A brief explanation of what is unusual would help clarify this.

12) Page 14, last sentence; Similar to Fig. 4E), it would be easier to understand if the ORF diagrams and domain structures were shown. A consistent presentation method is desirable. Additionally, it seems that there are two areas that clearly appear difficult to enter downstream, though very narrow range. Making these correspond to the ORF regions would enhance understanding.

13) Page 15, line 17; I believe this point is supported by the existence of other species that have split genes identified through homology analysis. Have any such examples been found?

14) Page 16, line 19; Regarding the left diagram of Fig. 6B, I understand that the overall transcription profile of the selected genes seems lower for me. This seems to contradict the statement, "This observation may be indicative that the strong P438 promoter is selected in those positions where it could lead to an increase in expression that could improve fitness." In particular, how should I interpret that the selected mutations, especially the insertions in the NE genes, are lower overall? Am I misunderstanding something? I feel my understanding of the figure is insufficient, so could you please explain this point?

15) Page 19, line 20; Reference 62; An update seems to have been released. L. V. Aseev, L. S. Koledinskaya, I. V. Boni, Extraribosomal Functions of Bacterial Ribosomal Proteins-An Update, 2023, Int J Mol Sci 25, (2024) Pubmed: 38474204.

16) Page 19, line 25; What physiological state does "10 generations" refer to in the context of the Mycoplasma growth profile? Specifically, at which phase-such as lag, log, stationary, or death phase-does refreshing occur? Adding a brief note to the methods section would enhance understanding. It seems that especially during prolonged stationary phases, different selective pressures may be applied compared to logarithmic growth. As supplementary information, providing a growth profile over 10 generations to indicate when medium refresh occurs and how proliferation continues would deepen understanding.

17) Page 20, line 8; Does this imply that expression regulation and coding regions are independent? If so, it has been previously established that there are significant differences in the expression mechanisms of chaperones between gram-positive and gram-negative bacteria, yet despite the differences in molecular mechanisms, the final expression profiles are similar. Citations may be needed for this point. Additionally, there are analyses related to evolution that suggest coding and non-coding regions evolve independently.

18) Page 20, line 21; Could this also simultaneously carry risks? A high mutation rate may facilitate the introduction of adaptive mutations in the early stages, but negative mutations may also increase simultaneously, so it is necessary to discuss how the balance between benefits and drawbacks fluctuates. In the long term, it is expected that the accumulation of disadvantageous mutations might also be accelerate.

19) Fig. 1; I do not fully understand the meaning of "Coverage." I understand it as a library where Tn jumps into all cells, creating insertion disruptions. After determining those positions, is it correct to understand what percentage of the entire genome can be

created at the base sequence level? If this value decreases, should I understand that it narrows down to specific insertion sites? I want to confirm because I might not fully understand; is this percentage the coverage rate of the inserted positions relative to the entire genomic sequence? I am not clear on what 100% signifies, and I may be misunderstanding, so I want to confirm this. However, since I feel that others might also share the same doubts, it may be necessary to provide a description that is easier for readers from other fields to understand.

20) Fig. 4D); I do not recognize a difference in scale from the figure. The scales appear to be the same, but it seems to me that only value=20 is not presented.

21) Fig 5B); I believe it shows the linker portion connecting the two domains of PheT, but it would be clearer if this were explicitly indicated in B).

22) Fig. 6; The text of the enrichment analysis in Figure. 6, results appears low resolution and difficult to read. It seems necessary to pay attention to the format of the figures in the final manuscript. Additionally, regarding the Y-axis scale of D), as pointed out in Figure 4, the description states that the scales are different, but in the figure, they appear to be the same.

23) Fig S1; The high correlation between P and T suggests that even with promoters and terminators inserted at both ends of Tns, their overall effect may not significantly influence the results. Is this incorrect?

24) Fig S16; Regarding the extended terminal regions of the E genes, based on the Tn sequence information, the structure of the additional insertion peptides of the final fused protein inserted into each frame should be able to be shown as a common sequence. I believe that by demonstrating this, it would be possible to explain the "extended termini regions" in a more understandable manner. It could illustrate what peptides are added to the N- and C-termini, displaying the conditions of each frame. Could this be organized as supplementary information?

Editor

Thank you for submitting your work to Molecular Systems Biology. We have now heard back from the three reviewers who agreed to evaluate your manuscript. As you will see from the reports below, the reviewers find the study interesting and relevant. They raise, however, a series of concerns, which we would ask you to address in a major revision.

I think the reviewers' recommendations are relatively clear, so there is no need to reiterate the points listed below. All the issues raised by the reviewers need to be satisfactorily addressed. As you may already know, our editorial policy allows in principle a single round of major revision, and it is therefore essential to provide responses to the reviewers' comments that are as complete as possible. Please feel free to contact me in case you would like to discuss in further detail any of the issues raised by the reviewers.

We thank all the reviewers for their thoughtful feedback and the editorial work for handling this manuscript. We believe the implementation of the provided suggestions have greatly improved both the content and its presentation.

Our detailed point-by-point responses are provided below associated with their relative comments (in blue).

Reviewer #1:

The manuscript "Quantitative essentiality in a reduced genome: a functional, regulatory and structural fitness map" by Miravet-Verde et al is really very interesting expanded and enhanced TnSeq Method with two different libraries for analyzing the essentiality of genes, genome regions and even protein domains encoded by gene fragments of the genome-reduced bacterium *Mycoplasma pneumoniae* at a high resolution. The authors included a number of passages which selected for growth to add a temporal aspect to their analysis, allowing to follow the fitness contributions of the different elements affected by transposons over time. To analyze their data the authors successfully added and validated different tools and approaches also to categorize the essentiality of their genes including encoded RNA species and genome fragments which can present binding sites for various DNA binding proteins, including transcription factors, RNA Pol, but also SMC proteins or DnaA.

The results of the manuscript suggest an improved appreciation of more complex and more variable properties and characteristics of "essential" genes, depending for example also on different genetic and/or environmental conditions.

We thank Reviewer #1 for the reviewing process and the constructive feedback. Please, see below a detailed point-by-point response to the specific comments and questions.

1.1. The genome-reduced bacterium *Mycoplasma pneumoniae* is of course a kind of special organism, however the authors suggest that their new method should also be applicable to other model organisms. I am wondering whether one could also add different stress conditions instead or during the here applied passages, to study the effect of e.g. heat, cold or antibiotic stress on the distribution, changes or appearances of e.g. new "essential" genes under such conditions.

We agree with this reviewer that *Mycoplasma pneumoniae* is a unique organism, with a high transformation efficiency and a reduced genome. Although this certainly facilitates obtaining a high coverage of transposition events, our quantitative approach could be applied in principle to any organism, only limited by this transposon coverage. This limitation is stated both in the introduction (third paragraph - page 3, as “this resolution should approach one insertion per base, with careful consideration given to the design of the transposon used in the analysis”) and in the last paragraph of the discussion (page 23) as

“... we envision that the methodology described here can be applied to other model organisms, only limited by the transposon insertion coverage that can be achieved in each particular cellular system”.

Furthermore, we have extended the references included in the discussion (last paragraph - page 23) to highlight the application of standard Tn-Seq experiments in other model organisms – such as *Escherichia coli*, *Burkholderia pseudomallei*, *Bacillus subtilis*, *Caulobacter crescentus*, or *Mycobacterium abscessus* – in which our approach could be extended:

“Given the widespread use of Tn-Seq in microbiology (Goodall *et al*, 2018; Christen *et al*, 2011; Wong *et al*, 2022; Akusobi *et al*, 2022; Meeske *et al*, 2016)...”.

We also agree that applying this methodology to different growth conditions could reveal a new essentiality status of specific genes under such conditions. Essentiality is in fact a conditional trait, meaning that genes that are non-essential in normal growth conditions may be essential in specific environmental conditions. Although our analysis was implemented under standard laboratory growth conditions, we agree that this essentiality study may be extended in future works under different environmental conditions to reveal conditional essential genes. This is now

commented in the revised manuscript at the end of the first paragraph on the discussion (page 19) and at the end of this same section (page 23).

Minor comments

1.2. P11 "affected by the binding of DNA proteins" What are "DNA proteins"?

We apologize for this typographical error. We have modified the sentence in the revised version as follows (page 12): "affected by the binding of DNA-associated proteins".

1.3. P12 "the Tn-seq data is sensitive to strong DNA-protein interactions" I would not state that "data" is sensitive. Maybe there could rather be a "competition" for DNA binding? Or a (recognition site) specific interference by DNA binding proteins?

Thank you for the suggestion. We have modified the sentence as follows (page 13): "Tn-seq data can be affected by the competition or interference of DNA binding proteins".

Reviewer #2:

Manuscript Title: "Quantitative essentiality in a reduced genome: a functional, regulatory and structural fitness map"

OVERVIEW

This manuscript presents a novel TnSeq approach to studying gene essentiality in the reduced-genome bacterium *Mycoplasma pneumoniae*, combining two transposon libraries to achieve near single-nucleotide resolution mapping in non-essential coding regions of the genome. The work demonstrates how this method can provide insights into genome organization and function, with implications for synthetic biology and genome engineering.

STRENGTHS

1. Novel dual-library methodology achieving high-resolution resolution and investigation of impact activating and repressing insertions
2. Comprehensive analysis of gene essentiality in a minimal genome organism

We thank Reviewer #2 for the reviewing process and the valuable comments. We also appreciate the provided highlights on the strengths of our study. Below, we provide a detailed point-by-point response to the specific comments and questions.

REVIEWER COMMENTS

2.1. Page 1: The claimed "near-single-nucleotide precision" needs qualification as resolution varies across genome

As Reviewer #2 rightly points out, the captured resolution varies considerably across different genomic loci. In theory, every genomic position could be disrupted; however, insertions in essential regions would not be recovered by sequencing, as they prevent cell growth and are eliminated during culture passages. Therefore, given the high density of insertions observed in non-essential genes at the first passage (92.4% of positions with insertions), it is reasonable to conclude that the initial coverage must be uniform and approach one per base.

To make it clear, the claim of near-single-nucleotide resolution assumes only those regions that can be potentially disrupted. For this assumption, we considered a set of known non-essential genes as we describe in the results and method sections:

Page 3. “~55% of the entire genome, reaching a transposon insertion coverage close to absolute saturation for NE genes (92.4% average linear density (LD), calculated as number of insertions / gene length for each genomic locus, ~1 insertion per bp resolution)”.

Page 5. “The combination of both libraries, referred here as ‘PT’, resulted in a total genome LD of 55.6% (453,897 unique insertions of a total of 816,357, with an average of 267.1 reads per insertion), and 92.4% when considering a set of known NE genes (20,385 unique insertions along 22,047 bp; Fig. 1B)”.

Page 32. “a set of known NE genes (n=29; 21,599 bp in total) was used to provide an average of the maximum saturation as the total number of insertions found mapping in these genes and divided by the region covered by these same genes”.

This is also highlighted in Figure 1B, where the top bar plot depicts the genome-wide insertion rate, while the bottom plot shows the same metric restricted to non-essential genes.

Nevertheless, to clarify this claim early in the manuscript, we have modified a sentence in the abstract as follows:

“To address this, we combined transposon libraries containing promoter or terminator sequences to obtain a high-resolution essentiality map of a genome-reduced bacterium, at near-single-nucleotide precision when considering non-essential genes”.

In addition, the reference and source employed as ‘gold set’ or reference set of genes with known essentiality is now included in Dataset EV2.

2.2. Page 3: The authors acknowledge that "the mariner transposable elements depend on TA dinucleotide targets, limiting the resolution" but don't discuss how this bias is addressed by their method. What is the insertion bias of Tn4001 transposase? How might this impact results?

Thank you for this comment. The insertion bias of the Tn4001 transposase was previously evaluated in Miravet-Verde et al., 2020. In this study, we found a slight preference for AT over GC at the base level, yet Tn4001 inserts into GC-rich regions as well. ANUBIS, the dedicated framework for Tn-Seq analyses used in our study, corrects for this slight bias. To clarify this point, we have highlighted this in the results and method section as follows:

Results section (page 3):

“We achieved this by designing and transforming the bacterium with two different Tn4001-based transposons, which insert randomly into the genome with only a slight TA dinucleotide preference that can be computationally corrected (Miravet-Verde *et al*, 2020)”.

Methods section (page 32) under “Metric and essentiality estimation by a GMM model” subsection:

“Repeated regions where the ambiguity prevents efficient mapping of insertion were ignored and GC preferences were corrected as done in previous studies (Miravet-Verde *et al*, 2020)”.

2.3. Page 5: It is unclear how the 55.6% of genome coverage is defined? Is there a maximum distance between insertions?

The combination of the two transposon libraries resulted in 453,897 unique insertions. Considering that the genome has a total of 816,357 bp, this represents a coverage of 55.6% of the total genome. This has been clarified now in the revised manuscript on page 5 by adding the genome size of *M. pneumoniae*.

Regarding the distance between insertions, it will depend on the length of the essential regions found in the genome, assuming transposon saturation as we have in our study. In our case, after segmenting the genome between essential and non-essential regions, the largest essential region, found was 5,594 bp. This is detailed in the results section on page 11:

“We detected in the PT1 library samples a total of 914 E segments with an average size of 328 bp, extending to a maximum of 5,594 bp”.

However, this does not mean that the maximum distance between two insertions is 5,594 bases since we can find some few insertions with a low number of reads scattered over E regions, which are considered background noise and are not taken into consideration by the algorithm to define E regions.

2.4. Page 7: K-means requires the setting of cluster numbers. I see a discussion of this in the methods section, but I would make a statement about how this was selected (supplementary fig appears to imply "elbow method" but this is not stated specifically). Why do you say "at least four different clusters"? If you could have used a larger k , what is the implication of picking $k=4$?

Thank you for your comment. We agree that this point requires clarification. As suggested by Reviewer #2, we used the elbow method to determine the optimal number of clusters, as shown in the previous Fig. S5, now Appendix Figure S4, and described in the "Essentiality decay exploration and k-means clustering" method section. We selected $k = 4$ because it was the point beyond which additional clusters provided diminishing returns in explaining the variance. When we referred to "at least four different clusters," we meant that the data structure supported a minimum of four meaningful groups (each containing at least 50 genes). While higher k values could partition the data further, increasing k did not substantially enhance the interpretability or biological relevance of the clusters, particularly when compared to the Gaussian Mixture Model (GMM) essentiality classification. This is why we chose $k = 4$ for our analysis.

In response to this comment, we have reworded the relevant line in the last paragraph of the "Gene Essentiality Estimation in Highly Saturated Transposon Libraries" section (page 7) to clarify our decision. Finally, we have rephrased and added details on the analysis choices in the "Essentiality decay exploration and k-means clustering" Material and Methods section.

2.5. Page 7: What do you mean by "copy number" of genes? You then refer to this as "protein abundance". How is this measured? If this is an issue, might it not indicate that T1 (~10 cell divisions) is too early to accurately evaluate gene essentiality?

We apologize for the confusion. We mean the copy number of proteins present in a cell resulting from the expression of a specific gene. To clarify this, we have modified the sentence as follows:

“These decay differences could be explained in part by the fact that NE/F2 genes clustered in C2 encode proteins with higher copy numbers in the cell, as compared to genes clustered in C3 and C4”.

Protein levels in average copies per cell were previously measured in *M. pneumoniae* by (Miravet-Verde *et al*, 2019), and the use of these datasets has been clarified now in the Material and Methods section on page 30.

The correlation found in this study between protein abundance and the differences in categorization depending on the passage is an example of possible factors affecting the accuracy of gene essentiality categorization. It is known that essentiality depends on experimental conditions, but here we show that it also depends on the passage number. If we measure essentiality too early, some essential proteins with long half lives could still be present even if the gene is disrupted. But if we measure essentiality after performing too many passages we start losing fitness genes by competition and defining them as E.

Therefore, we agree with the reviewer that the number of cell divisions needs to be taken into account to ensure a proper classification. Indeed, we believe that this is an important aspect supporting that our quantitative approach is much more accurate to define essentiality as compared to previous methods. We have now clarified this idea on page 8 as follows:

“These differences in protein abundance could impact the fitness effect of genes after disruption, as a certain degree of cellular function may persist during the initial passages until the protein is diluted and/or degraded (average protein half life in *M. pneumoniae* is

above 60 hours) (Maier *et al.*, 2011; Burgos *et al.*, 2020). Interestingly, we observed a positive correlation between protein copy numbers and essentiality as previously observed in *Escherichia coli* (Ishihama *et al.*, 2008), suggesting that E genes can be wrongly classified as NE in the first passages. These observations indicate that the number of cell divisions needs to be taken into account to ensure a proper classification of essentiality.”

2.6. Page 10: It was initially unclear the sliding window analysis included entire genome and not just "non-annotated regions". How was 31-bp for sliding window selected?

We apologize for the misunderstanding. The windows analysis is in fact applied at whole-genome level. This is now clearly stated in the text in addition to pointing to Material and Methods sections.

In addition to the already described methodology on the selection of a valid windows size for the presented analysis, we have added the Appendix Figure S15 that displays the accuracy on assigning a categorization of E or NE for windows extracted from the gold set of essential and non-essential genes (now listed as well in the Dataset EV2). This new figure shows the accuracy saturation (with 5% error allowed) to support the defined window size. Note this could be 30 bp as well, but as we defined window locations as the center of a genomic range, it was more convenient to take 31 so the window positions delimits ± 15 bp.

Finally, it is important to remark, as stated in the Material and Methods page 35-36, that 31 bp is just used as an initial step as we then consider contiguous windows together when they were classified as NE, leaving unassigned nucleobases as E (that can be shorter than 31 bp). To make the analysis stricter, we only considered domains covering ≥ 5 bp.

2.7. Page 13: Domain boundaries were defined "through visual inspection of the transposon data" from 16 genes seems subjective and may lack reproducibility. Can you provide a figure to explain method or reference existing figure.

We first performed an automated sliding-window analysis to identify DNA segments across the genome with different profiles of essentiality (see page 11, the results of this analysis can be found in Dataset EV7). As stated in page 11, although the algorithm was efficient to define the boundaries of transposon tolerance of E genes, the algorithm tended to define an excess of E and NE segments, specially across F1 and F2 genes, which is consistent with the discontinuous insertion profile typically found in these genes. We believe that this discontinuous profile of E and NE segments across a coding gene is difficult to interpret from a functional point of view. Moreover, it is important to remark that many of the genes identified presenting protein domains with distinct essentiality profiles were defined as F1 when considering the total length of the gene (See Dataset EV2).

Given the difficulty to automatically define these domain boundaries, we decided to evaluate by eye those genes presenting an apparent distinct profile of essentiality across the length of the coding region. For this, we plotted individually the transposon insertion profile for each gene. We identified a total of 16 genes presenting a distinct profile of essentiality across the length of the coding region, taking into account only those genes presenting distinct domains covering at least more than 15% of the total protein. After visual inspection of the transposon plots, we then defined the domain boundaries based on the position of the transposon insertions that more clearly separated the differential essentiality profile of these domains, taking into account the reads detected for each position. Examples of these plots can be found for example in Figure 4D, and Figure EV5.

Although we agree that a visual inspection analysis is subjective, we also believe that from a functional point of view, it is more reliable than the boundaries defined by the algorithm, as it tends to over-segment these genes. It is true that we may have missed some genes presenting differential essentiality, but we preferred to be more conservative. In addition, although the domain boundaries defined in Dataset EV2 may seem subjective, they are not arbitrary, as they are based only on transposon data. In fact, we found that they used to match well with protein domain signatures detected by blast analysis. On the other hand, we provide quantitative non-subjective data of essentiality of these domains once they have been defined.

Nevertheless, we agree with the reviewer that the exact proposed boundaries need to be considered as an approximation. This has been noted in the revised manuscript on page 14-15. In addition, a more detailed description of the method has been added. Importantly, all the raw transposon data is publically available, allowing the re-definition of these domains based on the investigator's own judgment or criteria.

2.8. Page 16: The manuscript mentions COG categories G, D, T, and S in the enrichment analysis, but doesn't provide specific examples of genes/pathways discovered. I am curious what fraction of these are regulators. Please provide some specific examples of genes/pathways.

M. pneumoniae contains very few transcriptional regulators. We found promoter insertions upstream of the transcriptional regulator *MraZ*, which regulates the cell division operon. Other interesting cases in which we identified enrichment of transposons containing promoters upstream of coding sequences are *RpoE* (MPN024), *RnaseJ1*(MPN280), *Tig* (MPN331) and *GroES* (MPN574), which are involved in transcription modulation and protein folding. We also found enrichment of transposons putatively affecting the expression of several genes belonging to the PTS system, a group of paralog genes with unknown functions of the MPN039 family, and an operon encoding several lipoproteins (MPN640 to MPN648). These findings have been included in the amended version on the second paragraph of the section (page 18) starting with:

“As examples, we identified enrichment of transposons containing promoters upstream the transcriptional regulator *MraZ*, which regulates the cell division operon, *RpoE* (MPN024) and *RnaseJ1* (MPN280)...”

In addition, note that all the results, including every gene showing selected enrichment for P or T, along with their COG assignment, name, and function, are summarized in Dataset EV11.

2.9. Page 17: You state that you work "supports the concept that gene essentiality is a conditional trait that depends on the genetic and environmental context". I don't understand how these specific claims are supported. Only 1 cultivation environment was used and strains were all single KOs (same genetic context). I think this method could be extremely useful in evaluating this in the future.

We apologize for the confusion of the statement, also raised by Reviewer #1 in the first comment. We agree that our data do not include essentiality under different environmental conditions. This statement has been modified as follows (page 19):

“Previous studies have shown that gene essentiality is a conditional trait that depends on the genetic and environmental context (Larrimore & Rancati, 2019). Our study also highlights that genes classified in a particular category have in fact distinct degrees of fitness influence. We also show that the number of cell divisions prior to the sequencing analysis has an impact on determining gene essentiality for fitness genes. This argues for the need for more accurate methods to improve the resolution of categorization.”

In addition, the idea of using this quantitative method to infer essentiality under different environmental conditions is now discussed more clearly at the end of the discussion in page 23 as follows:

“Future work aiming to determine conditional essentiality across different genetic and environmental contexts, such as cell stress or *in vivo* conditions (Danchin & Fang, 2016; Luhua *et al*, 2013; Cain *et al*, 2020; Zhu *et al*, 2018), may reveal a distinct essentiality status of specific genes in such conditions, potentially providing insights on the roles of these genes.”

2.10. Page 20: You claim that that this strategy will be a "very informative and useful to guide genome annotations" but there is no discussion of its application in other strains. How versatile is this system?

The high coverage of transposon insertions allowed us to delineate well the boundaries of essential genes, since the coding regions remain mostly free of transposon insertions, whereas the intergenic areas tend to accept transposon insertions (see for example Fig. 2C). Of course, essentiality data cannot be used to delineate the boundaries of non-essential coding regions. Therefore, when we suggest that Tn-Seq data can guide genome annotations, we refer to better identifying the boundaries of essential coding regions, as well as protein domains when one is E and the other NE, rather than inferring functional information from these genes. In principle, if enough transposon coverage is achieved, this information could be used to guide *in silico* annotation tools in other genomes or strains. Since we acknowledge that this statement can be misleading, we have removed it in the revised version.

Regarding the application on other bacterial species, please refer to our response to comment 1.1 from Reviewer #1. In summary, standard Tn-Seq experiments have been performed in diverse model organisms and the experimental design presented could also be implemented and tested in these models. Regarding the analysis and bioinformatic approach, such as the k-means strategy, these are not conditioned by the transposon design employed, thus applicable in general Tn-Seq methodologies.

2.11. *Figure 2C: Figure 2C shows that a large fraction of insertion positions has just 1 read, which seems questionable. These could be artefacts of the TnSeq protocol, or, more likely, remnants of low-abundance non-growing strains. It looks there are many of these within the should-be-essential genes, consistent with either interpretation. So it seems that the linear density should be computed without these 1-read insertions. On a related note, Figure S4B shows that the typical essential protein has insertions at ~15% of its locations, which seems way too high. We suspect that this would drop to a much lower and more reasonable number if the just-1-read cases were dropped.*

We agree that 1-read insertions can derive from sequencing artifacts. However, we would like to clarify that all the genomic profiles displayed in the graphs do not present these in any case. Please notice that the Y-axis is presented in log scale in every case and there is a gap between the 0 value ($\log_2(1)$) and the height of the insertion. Thus, the insertions visualized have in fact a minimum of 2 insertions. For clarity, we have included this in a sentence at the end of the section ‘Transposon insertion sequencing, insertion calling and visualizations’ in methods.

Regarding the comment on Appendix Figure S3 (formerly S4), we agree that it is not an ideal representation of the distribution, as most of the dots overlap, making it difficult to interpret. A more appropriate way to visualize these data would be through a histogram like the one shown in Fig. 1C. Nevertheless, by selecting only the E genes from Dataset EV4, it can be observed that the actual average linear density for this group is 12%. This value is expected, given that most

genes—regardless of their category—tend to present insertions at their N- and C-terminal regions, which typically accept transposon insertions. To address this, it is common practice to exclude 5% from each terminus of the protein (adding up to 10% of the gene), which aligns with the linear densities observed. However, as shown in this study through the specific analysis of terminal regions, and also in previous work (see Fig. 6 in Miravet-Verde et al., 2020), this correction is somewhat arbitrary and does not accurately represent every case (e.g., some genes have extensions at their C-terminus but not at their N-terminus, and vice versa).

We address this case by case with specific examples studying different essentiality domains as extensions and insertions that could derive in split proteins. Still, and as sanity check, we have run a couple of analysis:

- From Dataset EV1 it can be calculated that 29.3% and 29.4% of 1-read insertions in passage 1 for P and T libraries, respectively, present 2 or more reads in posterior passages.
- The correlation between calculating linear densities with 1-read insertions has a 0.99 coefficient, and these insertions might primarily affect F1 and F2 genes (in between 20% and 80% linear densities) as shown in the plot below:

- This filter would not impact the analysis based on GMM as the relative values between groups remain. For the AUC and k-means analysis, artifactual events will be diluted when integrating multiple passages. For the 1-bp enrichment analysis where these 1-read positions could bias the initial conditions, we considered insertions that were conserved in passage 1 and 10 in both replicates for each P and T condition.

In conclusion, while we understand the point raised by Reviewer #2, we have implemented specific strategies and controls to ensure that the presented and highlighted results are robust and reliable.

A few general comments:

2.12. The article uses 3 different operational measures of essentiality! It's a bit confusing, and is it useful to have so many of them? The area under a curve as a measure of fitness looks good, why not use just that?

We appreciate the observation and recognize that this point was not clearly conveyed throughout the manuscript. We understand that Reviewer #2 is referring to three metrics: linear densities (the number of insertions divided by the locus length, simply referred to as “coverage” when talking at whole-genome level), the AUC (area under the curve) measuring the preservation of insertions within a gene over time, and the read count of individual insertions.

The first point to clarify concerns on the terms *coverage* and *linear density*. While “transposon coverage” has been widely used in the literature to describe the number of unique insertions normalized by the genome size of the model organism, when the same calculation is applied to a specific genomic locus, the metric is often referred to as *linear density*, or simply *density*, which we denote in our study as “LD”. Linear densities are the most widely used metric in Tn-Seq studies for assessing gene essentiality. This metric, which can be calculated from a single sample, remains the primary method in state-of-the-art analyses to predict the essentiality of genomic elements, including Poisson-based methods and the improved Gaussian Mixture Model approach used in this study and supported by our previous work (Miravet-Verde et al., 2020).

In addition, this study introduces an AUC-based approach that leverages multiple passages to provide a quantitative metric capturing not just a static snapshot of essentiality but also fitness, measured as the persistence of insertions over time. The subsequent k-means clustering based on AUC values reveals distinct groups, as shown in Fig. 1E, which do not directly correlate with the essentiality classes predicted by traditional methods.

Thus, both metrics are valuable: linear density because of its established role in the field, and AUC because of the new, orthogonal information it provides. Relying solely on one metric would offer an incomplete view. To address this, we present both estimations in the supplementary material, allowing direct comparison with previous studies, this time at higher resolution, while also introducing the additional dynamic insights provided by the AUC approach. Importantly, the AUC method requires information from multiple passages, which increases both the time and sequencing cost compared to a standard essentiality study. Therefore, we also retained the linear density-based approach to highlight key biological aspects (e.g., the identification of two fitness gene groups, F1 and F2, labelled as quasi-essential and quasi-nonessential, respectively) that can be achieved using single libraries with optimized transformation and insertion saturation.

Regarding the read counts of individual insertions, these are used in the analyses presented in Fig. 6. This metric offers single-nucleotide resolution for assessing the fitness impact of specific insertions, providing a finer level of detail compared to metrics based on linear densities or AUC, which integrate information across broader genomic regions. This approach is used to identify single insertions highly selected after passage selection, information that is not represented by either the linear densities or the AUC metrics.

To improve the clarity, the second paragraph of the discussion (page 19) has been extended to clearly state the orthogonal application of the metrics and methods presented.

Finally, given that Reviewer #3 also raised concerns about the clarity of the coverage and linear density terminology, we have reviewed and updated the content in the new version to:

- Refer to transposon "insertion coverage" when describing the event of transposition generally across the genome.
- Refer to linear density (LD) in all other contexts when referring to a metric, such as when discussing the percentage of insertions normalized by genome size, gene length, etc. This metric is now also introduced and explained earlier in the Introduction.
- Update the terminology in figure representations (e.g., the top plot in Fig. 1B) to avoid possible misunderstandings.

2.13. 1 passage = 4 days = ~10 doublings -- so I'm wondering if some of the "essential" genes (with very few insertions after 1 passage) are not quite essential. The authors do have a "known" non-essential set (29 genes) to test on, which is reassuring, but it is not clear if this issue is fully addressed.

We agree with the reviewer that some genes at passage 1 could, in fact, be less essential if assessed after fewer divisions. However, it is important to note that there is a minimum of cell divisions needed to expand the transformant pool to acquire a representative sample of genomic DNA for TnSeq, and also to dilute E proteins with long half lives as previously discussed. Although we considered that analyzing an even earlier time point (equivalent to a "passage 0") might, in theory, provide additional information, this would likely capture a significant proportion of dead cells and non-viable insertions, which would introduce substantial noise and complicate the interpretation of essentiality. The high saturation achieved on the reference set of non-essential genes (now explicitly listed in Dataset EV2) supports that the conditions used were appropriate to robustly distinguish essential from non-essential genes. Thus, while we acknowledge the theoretical possibility raised, we believe that the chosen protocol provides a good balance between biological signal and practical limitations, as also evidenced by the validation with the non-essential reference set.

2.14. A prior study on *Caulobacter* (Christen et al) had ~0.1 insertion per bp. Is ~1 insertion per bp so much better? It seems likely that some of the results wouldn't have been obtained with "only" 0.1 insertion/bp. This should be explicitly addressed in the Discussion.

There are two main aspects influenced by resolution. First, previous studies, which achieved approximately one insertion every 4 and 3 base pairs (considering only non-essential genes), were able to support only three essentiality categories (Lluch-Senar et al., 2015; Miravet-Verde et al., 2020). Second, we extensively evaluated the impact of insertion saturation on essentiality estimation accuracy in a previous study (Miravet-Verde et al., 2020). From that study, it was evident that, as expected, all approaches tend to overestimate the number of essential genes at lower coverages (see Fig. 3A in Miravet-Verde et al., 2020), and that overall accuracy, when benchmarked against reference sets of essential and non-essential genes, drops below 50% when more than 20% of insertions are removed by subsampling (see Fig. 7 in Miravet-Verde et al., 2020; these gene labels appear now in Dataset EV2). Thus, with an insertion density of one every 10 base pairs, as in the *Caulobacter* study, many non-essential genes could be incorrectly classified as essential, and fitness genes would lack the resolution needed for effective ranking and be grouped together. In addition, this lower resolution would certainly affect the accuracy of the essentiality analysis of small genomic regions. Given the importance of essentiality studies in guiding effective genetic engineering strategies, we believe that the higher resolution achieved in this study will directly impact future experiments.

The improvement obtained with the current high-resolution libraries are now reflected on the discussion as:

- page 19 - “This improvement in resolution is relevant because essentiality studies using lower insertion coverage can lead to overestimation of E and F genes, significantly reducing the accuracy of essentiality predictions (Miravet-Verde *et al*, 2020), especially for small genomic regions...”
- page 20 - “whereas previous essentiality studies in *M. pneumoniae* were able to identify only three categories using ~4 insertions per bp for non-essential genes reported for *M. pneumoniae*”.

2.15. "Overall, these observations highlight the importance of assessing essentiality at longer growth time points." -- An N-terminal part of MPN119 does appear to be important for fitness, but it may not be exactly essential -- it's probably a subtle growth defect (starting to be apparent after 2 passages ~20 generations).

We agree that the results do not necessarily indicate this region is essential, but indicates that it has an important contribution in fitness compared to the rest of the protein. We have modified the sentence as follows (page 16):

“Overall, these observations highlight the importance of considering multiple passages to assess the fitness contribution of genes and protein domains”.

2.16. "Although this C-terminal extension [of NusA] was previously established as essential, our data indicates it is fully disruptable at passage 1" -- The cited reference has a targeted genetic assay. Is there a reason why it found essentiality and the pooled random approach didn't?

The mutational approach used in O'Reilly *et al.*, is based on ssDNA recombineering. In our experience, this system in *M. pneumoniae* is very inefficient. This means that the absence of signal or detection of the intended mutant, does not necessarily mean that the mutant cannot be obtained. The SURE editing method used in our study is more efficient in selecting rare recombineering events and it would be a more appropriate method to attempt this kind of mutation as demonstrated in Piñero-Lambea *et al*, 2022. In any case, in our opinion, an experiment aimed to demonstrate the impossibility to achieve the mutation requires at least a control showing that the mutation in this specific locus is possible. One strategy could be the demonstration that the mutation is feasible in a strain containing two copies of the gene, or using an inducible system. For these reasons, we believe that it is not clearly demonstrated that the C-terminal region of NusA cannot be deleted. In fact, our transposon data clearly shows that the C-terminal region of NusA can be interrupted, at least at first passages. As the obtention of viable transformants requires several generations, it is also possible that such a mutant could not be obtained if the C-terminal region of NusA plays a strong role in fitness. In fact, as mentioned in the text, our results suggest this, since the persistence of the transposons detected in this C-terminal domain is low after consecutive passages.

2.17. Getting back to a comment above: The Methods reports that Figure 6C uses a false discovery rate approach to adjust for multiple testing. This is fine, but it was not apparent from either the figure itself or from the text of the results. It would be better to see some examples of genes with the strongest evidence of a fitness benefit from increasing their expression level, rather than something so abstract as a COG enrichment test.

We apologize for the possible misunderstanding. For clarity, we have included the method and adjustment in the results sections the following sentence (page 18):

“we only detected COG enrichment for the S category after applying a hypergeometric test and adjusted it using the Benjamini-Hochberg method”

Regarding highlighting specific examples, and as commented above, we have now described some examples to close the second paragraph of the section ‘Assessment of transposon insertion events improving cell fitness’ (page 18).

TYPOGRAPHICAL ERRORS:

2.18. Figure 2C: "Fig. S2C shows a large transcription unit mainly constituted by E ribosomal genes" -- that should be Fig. 2C

Thank you for noting this error. This has now been corrected.

2.19. Figure 5: log2 reads has a different scale, are there no instances of insertions with just (say) 1-2 reads? Or are those filtered out from this display?

As explained in comment 2.11, we have now made clear in the Material and Methods section that 1-read insertions are not displayed. In addition, we have reviewed all the figures to ensure the scales are similar between plots.

2.20. Page 4: "To eliminate death cells"

Thank you for noting this error. This has been corrected in the revised version of the manuscript.

2.21. Page 6: "Functional RNAs such as ribosomal RNAs (MPNr01, MPNr02, MPNr03), MPNs01 (4.5S RNA), MPNs03 (RNaseP) and MPNs04 (tmRNA), they were all essential for survival"

We have modified the sentence as follows: “Functional RNAs, including ribosomal RNAs (MPNr01, MPNr02, MPNr03), MPNs01 (4.5S RNA), MPNs03 (RNaseP) and MPNs04 (tmRNA) were essential for survival”.

2.22. Page 13: "transposons carrying promoters identified predicted longer N- and C-terminal"

Thank you for noting this error. This has been corrected in the revised version of the manuscript.

2.23. Page 13: "For example we o found cases like mpn229"

Thank you for noting this error. This has been corrected in the revised version of the manuscript.

2.24. Page 20: "We show that the leverage of Tn-Seq data"

Thank you for noting this error. This has been corrected in the revised version of the manuscript.

CONCLUSION

The manuscript presents a valuable approach to studying gene essentiality with high resolution in a minimal genome. With clearer explanation of the analytical choices, this work could make a significant contribution to our understanding of bacterial genome function and the definition of essential genes beyond binary classifications.

We thank Reviewer #2 for the thorough review and valuable feedback. We sincerely hope that the concerns raised have been properly addressed and that their implementation has improved the presentation of our research.

Reviewer #3:

Manuscript Reference: MSB-2025-12901

Authors: Samuel Miravet-Verde, Raul Burgos, Eva Garcia-Ramallo, Marc Weber, and Luis Serrano

Title: Quantitative essentiality in a reduced genome: a functional, regulatory and structural fitness map

This paper improves the random Tn insertion method, a powerful approach for analyzing the essentiality of bacterial genomes, by refining the structure of Tn to be used and the analysis methods. Previous Tn insertion analyses have focused on coding regions, where their essentiality has been debated. In this paper, the authors advance their analysis to include non-coding regions by enhancing the Tn structure and methodology.

The improvement of the Tn structure allows for the insertion of promoters or terminators adjacent to the IR region within the Tn, enabling transcription in both directions regardless of the insertion orientation, while also terminating transcription of any transcript originating from, or passing through the transposon sequence. Using these Tns, the authors conducted a quantitative analysis with high resolution at the bp level, including non-coding regions, and obtained time-course data with quantitative measures. After standardizing the data, they applied unsupervised k-means clustering for further analysis.

This manuscript is a well-considered study, and considering the reliability of the data, it is believed to provide significant benefits to the readers. I would like you to consider the following questions and comments.

We want to thank Reviewer #3 for evaluating and revising the manuscript. Below, we provide a detailed point-by-point response to the specific comments and questions.

Global Points:

3.1. Regarding mutations introduced into the genome through Tn insertion or homologous recombination, it is inevitable that differences in physical conditions due to genomic positions or sequences, as well as variations in interactions and modifications with or by proteins, will affect the efficiency of these insertional mutations. One of the standardization in these types of analysis could involve calculating fold changes at the same insertion site between control and samples. This raises the question of whether it is truly possible to compare quantitatively across genomic positions. Differences are also clearly observed in the integration of fragments into the genome via homologous recombination and in amplification efficiency by colony PCR.

Although standardized quantitative data from insertion sites are presented, it may initially seem that such issues do not exist. While detailed analyses and discussions regarding the validity of standardization have been conducted in the past, this time, as the authors were focusing on genomic regions different from the coding regions, I feel it is crucial to consider whether the previous standardization functions equally well. In particular, interactions with other proteins may require more thorough investigation than the coding regions.

Has there been a significant accumulation of data regarding the characteristics of insertion patterns in raw data for both coding and non-coding regions, and whether the same standardization is appropriate? I believe this is a problem that needs to be kept in mind at all times.

We agree that this is an important matter to take into account and that is the main reason why our analysis for elements other than genes refer only to either essential or non-essential, and fitness categories are not considered for such cases. However, we believe that several aspects of our study mitigate the potential biases raised:

- First, the extremely high saturation of insertions achieved in our libraries provides the required resolution to explore non-coding regions reliably, using metrics such as linear density. In parallel, we also present an extensive analysis relying on read coverage, that can go to 1-bp resolution, instead of insertion profiles, and they are in fact retrieving examples later validated such as the case of split proteins.
- We have extended the description of the analysis using 31-bp sliding windows, and taking into account a validated set of essential and non-essential genes (see comment 2.6. from Reviewer #2), we provide evidence that the class prediction based on linear densities is accurate even for very short domains (added as Appendix Figure S15). In short, this figure displays the accuracy on assigning a label E or NE for windows extracted from the gold set of essential and non-essential genes (now listed as well in the Dataset EV2). This analysis shows the accuracy in assigning NE elements is close to 100% for all cases and >80% for 5 bp or more.
- Importantly, when examining key non-coding domains such as promoters, terminators, or intergenic regions (which are typically shorter than 20 bp) we observe that these elements are overwhelmingly classified as non-essential (as shown in Fig. 2). This general trend supports the robustness of our standardization across different genomic contexts.
- The few exceptions identified, such as regions overlapping with previously reported DNA-binding domains (Fig. 3), actually reinforce the validity of the analysis: these cases align with biological expectations and demonstrate that when deviations from the non-essential pattern occur, they are biologically meaningful.
- Furthermore, the analytical approaches relying on AUC (Area Under the Curve) scores take into account the persistence of insertions within the same genomic region size, which minimizes the impact of local variations in insertion efficiency.

Finally, we agree with the reviewer that insertion biases related to physical or protein interaction factors should be continuously kept in mind. While accumulating large datasets for systematic comparison between coding and non-coding regions would certainly be valuable, the current data, combined with biological consistency of the findings, suggest that the standardization applied remains appropriate for the non-coding regions analyzed in this study.

3.2. Additionally, Tn transposition may not only occur during the log phase of cell growth. While differences in efficiency may exist, transposition is likely to occur even during the stationary phase. What are your thoughts on this? Moreover, in the log phase, the average number of DNA molecules per cell due to genomic positions may differ due to replication; does the raw data reflect such a gradient? It seems that there are still hidden issues.

In our experimental setting, transposition depends on several events. By one hand, the mini-transposon plasmid needs to be uptaken by the cells after electroporation, and then, the transposase gene needs to be expressed to initiate the transposition event. Both things dictate the transposition efficiency. In our case, we used cells in exponential phase for electroporation, since these are the conditions that maximize the transformation efficiency in mycoplasma (see for example recent publication by Mizutani et al, 2025 in a related mycoplasma species).

We assume that transposition takes place as soon as the transposase is expressed, which can occur during the 2-3h of recovery that is allowed after electroporating the cells for the expression of the selection marker. Although we agree that it would be interesting to know whether the transposition efficiency varies along the cell growth phase, this is extremely difficult to determine as the whole process depends on the transformation efficiency. Nevertheless, we presume that the expression of the transposase may be decreased in the stationary phase. In any

case, our experimental conditions have allowed us to obtain a high-enough transposon insertion coverage to develop this study.

In addition, it is important to mention that our transposon library was conceived to be static, meaning that once the transposon has been inserted in the genome, it cannot mobilize again. Thus, the transposon insertion sites obtained in the library are fixed independently from the moment when the transposition event took place. Therefore, whether the transposition event has occurred during the exponential or stationary phase does not seem to be a confounding factor. Moreover, note also that these transformants are then grown in several passages covering all the growth phases during the selection process.

Finally, the fact that we analyze pools of cells with each sequencing event, minimizes possible biases such as the fact that some cells may contain two copies of the genome at the moment of genomic purification. In fact, this is a minor problem given the slow growth rate of *M. pneumoniae* (8 hours cell division time under the best conditions), which limits the number of cells with multiple genomes. Even if this occurs, the insertion position remains unchanged, and only the read count would be affected, but the analyses presented primarily rely on insertion densities across different genomic loci.

3.3. While it is stated that the absence of Tn insertion does not definitively indicate essentiality, understanding how to make this judgment necessitates considering the presence of proteins that interact with that region, which is well understood and I completely agree. Homologous recombination also shows varying recombination efficiencies based on genomic positions, though it is the case using different bacteria. Differences in PCR amplification efficiencies depending on the genomic position are also observed, and these all seem to stem from common causes. This leads to the possibility that nucleoid structure may also play a role. Has there been an observation made without normalizing insertion frequencies when observing the entire genome? Are there any frequency variations or patterns according to the genomic positions?

We agree that genome structure can be another factor that can potentially affect the transposition profile observed in some cases. However, we have been unable to detect any significant bias based on the genome position. Nevertheless, note that these confounding factors are difficult to detect, especially if the physical constraints are dynamic. In this regard, we have only detected significant variations in transposition events for DNA motifs that are probably protected more permanently.

Nevertheless, as sanity check on potential biases by genomic location, we took into account all genes in *M. pneumoniae* in relation to 1-100 bins from the genome (each representing ~81 kb) and measured the average linear densities by each category (E, F1, F2 and NE) per bin. This reveals that the calculated linear densities are consistent per category independently on their location and it is not observed an increase in regions closer to the origin of replication. We have added a sentence in the discussion as follows in page 21:

“It is also possible that some E regions could be due to nucleoid structural constraints preventing access of the transposase, although we did not find any bias in average density of insertions over the genome.”

Analysis of potential metric biases based on the genomic location: For each 1 to 100 bin of the genome in *M. pneumoniae* we collect the linear densities of genes in that bin and split by essentiality category (blue to green colors). Dashed lines present the total average for all the genes in that category.

If the reviewer thinks it is important to show this data we could add it as a supplementary figure.

Minor Points:

3.4. 1) Page 4, line 5: Is there a possibility that in the strains with inserted P-type Tn, transcription of downstream genes continues while simultaneously inhibiting transcription of upstream genes? This depends on the orientation of the upstream genes, but particularly in regions with operon structures containing multiple genes, it can be considered that there is a possibility of inhibition for genes that are transcribed in the same orientation as the disrupted gene.

We agree that this can be a possibility. With the aim of recovering the maximum of transposon insertions, we prioritized minimizing polar effects, especially in a compact genome as it is *M. pneumoniae*. This includes genetic configurations having genes facing in opposite directions. Although we agree that this kind of transposon design can potentially have a negative effect when the transposon is inserted between genes having the same orientation, we do not think this has affected at all to the final coverage obtained as judged by the results. One clear example is shown in Fig2C, that presents the transposon profile of an E operon containing multiple genes in a co-directional orientation. However, we do recognize that this design may have impaired the fitness of some specific mutants depending on the position of insertion. In this case, we would expect a lower persistence of these insertions after several passages, or see differences when compared to the T library. Despite convergence transcription interference by RNA polymerase collisions has not been analyzed in detail in *M. pneumoniae*, antisense RNA has been shown to have a minor influence in *M. pneumoniae* (Llorens-Rico et al, 2016).

Nevertheless, as a sanity check, we have performed an analysis where we selected the last and second-to-last genes from every operon containing two or more genes in *M. pneumoniae* (n =

166). We then compared the expression levels, in the wild-type strain, of the second-to-last genes between two groups: those followed by a downstream gene with an associated promoter (in green, n=66), and those without (blue, n=100).

It can be observed that no significant differences (T-test two tails, $P=0.91$) in expression are detected between the two groups (genes with a downstream promoter and without).

It is important to emphasize that this analysis represents an initial approach conducted in wild-type *M. pneumoniae*. Ideally, confirming this specific effect would require experimental validation through isolation and characterization of an appropriate mutant strain. Nevertheless, we consider the current evidence to be well substantiated. For instance, we highlight the ribosomal operon, which is known to tolerate P transposons effectively and is therefore expected to exhibit this phenomenon. Whether this operon subsequently disfavors such insertions over longer evolutionary timescales remains an open question. We trust this explanation addresses the reviewer’s concern, and we remain available to provide further clarification if necessary.

3.5. 2) Page 5, line 4; If the correlation between P and T is very high, does it mean that even with promoters and terminators inserted at both ends, their overall effect is not significantly influenced?

We apologize for the confusion. We meant that there exists a good correlation between the replicates for each library, not between the P and T libraries. The sentence has now been clarified.

3.6. 3) Page 6, first paragraph; In the classification by anti-codon, wouldn't it be clearer to categorize which amino acids have multiple tRNAs and classify them as essential or non-essential? For instance, presenting a table of the codon table for Mycoplasma alongside the number of tRNA genes might clarify the explanation. As it stands, it is a bit hard to understand.

A table presenting this information can be found in Dataset EV2. Note that this information is included in the tabs labeled as “tRNA” and “codons” of the excel file.

3.7. 4) Page 6, last two sentences; It is difficult to simply infer whether a certain target gene is dispensable or not. It is necessary to consider the impacts of genetic interactions. While gene may be dispensable in isolation, there are many cases where it becomes essential when combined another.

We completely agree that the essentiality of a gene depends on the genetic context. We have modified this sentence in the revised version to clarify this point (page 7):

“Although gene essentiality depends on genetic context and potential interactions involving the encoded protein, these findings suggest that some of these genes may be dispensable in a synthetic genome or, if deleted, could still yield a viable organism, albeit with impaired growth.”

3.8. 5) Page 7, line 23: From SupFig7, it appears that the average intracellular copy number of essential gene products seems higher than that of non-essential genes. Is this considered to be specific to *Mycoplasma* or it is quite common between any microbes analyzed this type of analysis? It is thought that some essential genes may exhibit lethality when overexpressed. Alternatively, could the presence of genes that exhibit lethality when overproduced, such as those requiring large amounts of proteins like ribosomes, obscure this trend? Generally, do most bacterial species analyzed for essentiality and their intracellular product amounts show the same trend?

Thank you for noting this. Similar results, with abundant proteins tending to be predominantly essential have been described in *E. coli* (Ishihama et al. 2008). We have now added this reference to our manuscript (page 8).

Considering that a controlled expression without incurring lethality exists, it is reasonable to think that essential genes constitute the major building blocks of a cell. We presume that this is especially true for a reduced-genome organism. Of course, this observation refers to a general trend, as there are also essential proteins like MPN525 (a regulatory replication factor) with a low or regulated copy number. In addition, it has been shown that in general essential genes tend to participate in a larger number of protein-protein interactions than non-essential genes (Rancati et al., 2017). This could imply a higher protein abundance requirement, or that these proteins may be more abundant as a consequence of a major protein stability due to the fact of being part of protein complexes.

Nevertheless, as sanity check, we repeated the analysis presented in the mentioned figure excluding all genes from COG category J (corresponding to Translation, ribosomal structure and biogenesis) and while the E category is slightly reduced, the trend described is maintained:

3.9. 6) Page 11, line 11; Is it a mistake to refer to *mpn688* instead of *mpn668*? In Fig. 3B, it is indicated as *mpn668*. Additionally, as gene names are mentioned in the text, including gene names in the figures would help readers. This applies to other figures as well. However, care must be taken not to make the figures too cluttered.

Thank you for noting this error. The correct gene name is *mpn688*. We have now corrected the labels of Fig. 3B as well as the figure legend.

3.10. 7) Page 11, line 14 - ; Regarding the essentiality of *DnaA* boxes, if ncRNA is clearly expressed from this region, it should be discussed alongside the expression of ncRNA. It is conceivable that replication initiation may occur not only due to the relaxation of the double helix caused by *DnaA* binding but also due to the opening of the double strand caused by ncRNA transcription. Alternatively, has this discussion already been resolved in the past? If replication initiation points are already determined, the positional relationship between those points and the transcription start points of ncRNA is also expected to yield important suggestions. If this has been established, relevant citations are necessary.

Thank you for this suggestion. We agree that it is intriguing the presence of these two ncRNAs within the *oriC* region and their possible involvement in the initiation of replication. Non-coding RNAs have been shown to have a role in regulating different steps of DNA replication in some bacterial plasmids, during the recruitment of the origin recognition complex in the protozoa *Tetrahymena thermophila* (Mohammad *et al.*, 2007), as well as regulating cell division in *E. coli* (Faubladier *et al.* 1990). However, the specific role of these ncRNA in *M. pneumoniae* DNA replication remains unclear. To emphasise this point, we have added at the end of the section 'Mapping local-level essentiality using an annotation-independent approach' of the revised version a comment on the possible role of these ncRNAs.

3.11. 8) Page 12, line 15, and Fig. 3C; The target indicated by the green arrow needs improvement. Shouldn't it point to the green box?

Thank you for noting this error. We have fixed the position of the arrow to its right location in the revised version.

3.12. 9) Page 12, line 18; In this regard, a discussion of the positional relationship between the *HrcA* binding site and the promoter should be included. Additionally, even if the binding site and

the promoter are somewhat separated by Tn insertion, I think there is a possibility that HrcA could still maintain its function as a transcription factor.

The MPN332 coding region shares a transcript with the upstream gene (MPN331). However, MPN332 has its own promoter regulated by HcrA. What we have found is that the intergenic region between MPN331 and MPN332, which includes the regulated promoter and the HrcA binding site can be deleted (Burgos et al., 2020), demonstrating that this region is not essential under standard growth conditions. Therefore, we conclude that the absence of transposons within the HrcA binding site is probably due to the interference of HrcA when it is bound to this DNA domain. Our study does not intend to evaluate the function of HrcA in a context of transposon insertions surrounding this DNA binding domain. This has been clarified in the section 'Impact of genome accessibility on Tn-Seq data' in the revised version of the manuscript.

3.13. 10) Page 12, line 24 - ; The statement that the absence of Tn insertion does not definitively indicate essentiality suggests that the presence of proteins interacting with that region needs to be considered, which I understand well and agree with. Not only in Tn insertion but also in homologous recombination, different bacteria exhibit variations in recombination efficiency based on genomic positions. Furthermore, amplification efficiencies by colony PCR method also show variations based on genomic positions. I feel that all of these stem from a common cause. While the structure of the nucleoid is thought to have an influence, when observing the entire genome without normalizing the insertion frequency, wasn't there an observation of frequency variation dependent on genomic positions?

As commented and analyzed in comment 3.3., we have been unable to detect any significant bias based on the genome position. Nevertheless, note that these confounding factors are difficult to detect, especially if the physical constraints are dynamic. This of course does not mean that there could be key regions involved in nucleoid structure that we find essential due to steric constraints. In addition and specifically for this comment, we have added a small note highlighting this in page 13 when describing the impact of regulatory binding proteins (page 13):

“Similarly it could be that some of the E regions in the ori could be due to nucleoid organization by the SMC complex”.

3.14. 11) Page 14, line 1; The phrase "Due to its unusual genetic code" is hard to grasp. A brief explanation of what is unusual would help clarify this.

M. pneumoniae employs the UGA codon to encode for tryptophan, instead of a stop codon (codon table 4, instead of 11, commonly employed by other bacterial species). Therefore, only two codons (TAA and TAG) are used in *M. pneumoniae* to terminate translation. Consequently, *M. pneumoniae* only possesses the release factor RF1, which recognizes TAA and TAG codons. As RF1 is sufficient to recognize all stop codons, the release factor RF2 that recognizes TAA and UGA codons has been lost through evolution. This has been clarified in the revised version of the manuscript.

3.15. 12) Page 14, last sentence; Similar to Fig. 4E), it would be easier to understand if the ORF diagrams and domain structures were shown. A consistent presentation method is desirable. Additionally, it seems that there are two areas that clearly appear difficult to enter downstream, though very narrow range. Making these correspond to the ORF regions would enhance understanding.

We apologize for the inconsistent representations. We have now updated all figures to use a consistent format, considering schemes, ranges, labels and axes, to facilitate understanding.

Regarding Fig. 4E, we have added the genetic context including the PFAM domains mapped and the putative small ORFs that could also be involved in the observed profile.

3.16. 13) Page 15, line 17; I believe this point is supported by the existence of other species that have split genes identified through homology analysis. Have any such examples been found?

Thank you for this comment. Although we have not found exactly the same examples, we did find some studies that have designed orthogonal aminoacyl-tRNA synthetases/tRNA pairs that do not interact with the native translation system, using split aminoacyl-tRNA synthetases. We have added a reference in the revised version of the manuscript:

“Supporting this, orthogonal aminoacyl-tRNA synthetases/tRNA pairs have been designed by generating split forms of these enzymes (Jiang et al, 2023)”.

3.17. 14) Page 16, line 19; Regarding the left diagram of Fig. 6B, I understand that the overall transcription profile of the selected genes seems lower for me. This seems to contradict the statement, "This observation may be indicative that the strong P438 promoter is selected in those positions where it could lead to an increase in expression that could improve fitness." In particular, how should I interpret that the selected mutations, especially the insertions in the NE genes, are lower overall? Am I misunderstanding something? I feel my understanding of the figure is insufficient, so could you please explain this point?

We apologize for the confusion and making us notice the caption was unclear due to some misleading indications on the referred guides. Figure 6B compares, for essential and non-essential genes (X-axis), the RNA expression levels in wild-type *M. pneumoniae*, separating genes based on whether a transposon insertion enrichment is detected (darker color) or not (lighter color). From this analysis, we observe that while essential genes show similar expression levels regardless of having enrichment of P transposon insertions, non-essential genes enriched for P transposon insertions tend to have lower expression levels in the wild-type strain. Given that P438 is a strong promoter in *M. pneumoniae*, leading to significantly higher-than-average gene expression, we hypothesize that insertions of P438 upstream of these lowly expressed non-essential genes may actually confer a positive fitness effect.

We have rephrased the caption in the new version to properly depict how we interpret the figure and updated the legends.

3.18. 15) Page 19, line 20; Reference 62; An update seems to have been released. L. V. Aseev, L. S. Koledinskaya, I. V. Boni, Extraribosomal Functions of Bacterial Ribosomal Proteins-An Update, 2023, Int J Mol Sci 25, (2024) Pubmed: 38474204.

We thank Reviewer #3 for noting this. We have updated the reference in the revised manuscript.

3.19. 16) Page 19, line 25; What physiological state does "10 generations" refer to in the context of the *Mycoplasma* growth profile? Specifically, at which phase-such as lag, log, stationary, or death phase-does refreshing occur? Adding a brief note to the methods section would enhance understanding. It seems that especially during prolonged stationary phases, different selective pressures may be applied compared to logarithmic growth. As supplementary information, providing a growth profile over 10 generations to indicate when medium refresh occurs and how proliferation continues would deepen understanding.

M. pneumoniae is a slow growing bacterium with a doubling time of approximately 8h at exponential phase. In this study, we performed 10 serial passages, each passage consisting of 96h of culture, which represents approximately 10 generations (please note that 8 hours division time happens at the beginning of the cell culture and then it slows down). During this culture setting,

cells grow in exponential phase most of the time, reaching to stationary phase at 96h, which is the moment where refreshing occurs. As suggested, we have clarified this in the Material and Methods section and included a new Supplementary figure (Appendix Fig. S14) showing a representative growth curve.

3.20. 17) Page 20, line 8; Does this imply that expression regulation and coding regions are independent? If so, it has been previously established that there are significant differences in the expression mechanisms of chaperones between gram-positive and gram-negative bacteria, yet despite the differences in molecular mechanisms, the final expression profiles are similar. Citations may be needed for this point. Additionally, there are analyses related to evolution that suggest coding and non-coding regions evolve independently.

Our interpretation of the results is that *M. pneumoniae* is resistant to transposon perturbations of expression as long as essential genes are expressed. In other words, the majority of promoters or regions controlling the expression of E genes could be substituted by transposons containing promoters. This implies that in general no strict regulation is necessary for the majority of the genes. Furthermore, and as the reviewer comments, coding and non-coding regions could evolve independently as shown in prokaryotes and eukaryotes. We have edited the sentence and extended with the references as follows (page 22):

“ We found that gene expression in *M. pneumoniae* is in general resistant to transposon perturbations, indicating a lack of strict regulation in normal growth conditions. This would also suggest coding and non-coding regions could evolve independently, as shown in prokaryotes (Rogozin *et al*, 2002) and eukaryotes (Kowalczyk *et al*, 2022)”

3.21. 18) Page 20, line 21; Could this also simultaneously carry risks? A high mutation rate may facilitate the introduction of adaptive mutations in the early stages, but negative mutations may also increase simultaneously, so it is necessary to discuss how the balance between benefits and drawbacks fluctuates. In the long term, it is expected that the accumulation of disadvantageous mutations might also be accelerate.

We agree that a high mutational rate can potentially generate both disadvantageous and advantageous mutations. However, the growth selective pressure imposed across several passages would probably select only advantageous mutations conferring a major fitness growth compared to their competitors. To clarify this, we have modified this sentence as follows:

“Thus, although MPN294 may play a direct role, it is tempting to speculate that its disruption may cause increased mutation rates, leading to a larger repertoire of mutants. In this scenario, mutations providing a competitive advantage during growth selection are expected to be positively selected along with MPN294 inactivation”

3.22. 19) Fig. 1; I do not fully understand the meaning of "Coverage." I understand it as a library where Tn jumps into all cells, creating insertion disruptions. After determining those positions, is it correct to understand what percentage of the entire genome can be created at the base sequence level? If this value decreases, should I understand that it narrows down to specific insertion sites? I want to confirm because I might not fully understand; is this percentage the coverage rate of the inserted positions relative to the entire genomic sequence? I am not clear on what 100% signifies, and I may be misunderstanding, so I want to confirm this. However, since I feel that others might also share the same doubts, it may be necessary to provide a description that is easier for readers from other fields to understand.

We apologize for any misunderstanding that may have arisen regarding this point. First, we would like to acknowledge that Reviewer #3 correctly interpreted the term. The term "transposon coverage" has been widely used in the literature to describe the number of unique insertions normalized by the genome size of the model organism. However, when the same calculation is applied to a specific genomic locus, the metric is often referred to as linear density, or simply density, which we denote in our study as "LD."

Given that Reviewer #2 also raised concerns about the clarity of this terminology, we have reviewed and updated the content in the new version to:

- Refer to transposon "insertion coverage" when describing the event of transposition generally across the genome.
- Refer to linear density (LD) in all other contexts when referring to a metric, such as when discussing the percentage of insertions normalized by genome size, gene length, etc. This metric is now also introduced and explained earlier in the Introduction.
- Update the terminology in figure representations (e.g., the top plot in Fig. 1B) to avoid possible misunderstandings.

3.23. 20) Fig. 4D); *I do not recognize a difference in scale from the figure. The scales appear to be the same, but it seems to me that only value=20 is not presented.*

We apologize for the unclear scales on this representation. The scales were in fact not the same per passage as it happened for the left gene on the same panel. This has been fixed in the new version.

3.24. 21) Fig 5B); *I believe it shows the linker portion connecting the two domains of PheT, but it would be clearer if this were explicitly indicated in B).*

The linker connecting the two domains is labeled in red. This is indicated in the figure 5 legend, panel B, as follows:

“Domains B1-B2-B3 (in green) and B3-B8 (in blue) are separated by a linker (in red) that accepts preferential insertions of the P library. Within this linker, we show residue L191 (in yellow), which corresponds to residue M208 of *M. pneumoniae* PheT protein based on sequence homology.”

3.25. 22) Fig. 6; *The text of the enrichment analysis in Figure. 6, results appears low resolution and difficult to read. It seems necessary to pay attention to the format of the figures in the final manuscript. Additionally, regarding the Y-axis scale of D), as pointed out in Figure 4, the description states that the scales are different, but in the figure, they appear to be the same.*

We acknowledge that the resolution of this figure appears insufficient when embedded in the document. However, this issue arises from the way Google Docs renders embedded vector figures, and unfortunately, we were unable to fully resolve it (while the current version is initially better). While the resolution was appropriate when checking the figure in the provided PDF/SVG formats, we noticed that some fonts were rendering incorrectly. To address this, we have updated the figures to use scalable fonts and expect that the problem will not occur when figures are accessed directly from the main source files.

Regarding the scale, we thank Reviewer #3 for pointing this out. The caption was indeed incorrect and originated from an earlier version of the figure, which used different scales. This has now been corrected.

Finally, we conducted an extensive review and curation of all figures, ensuring proper rendering in scalable vector formats, harmonizing the figure styles as also suggested by Reviewer #3, and proofreading all figure captions. We hope that the figures are now much clearer and consistent.

3.26. 23) Fig S1; The high correlation between P and T suggests that even with promoters and terminators inserted at both ends of Tns, their overall effect may not significantly influence the results. Is this incorrect?

We believe Reviewer #3 is referring to the former Supplementary Figure S2, now labelled as Appendix Figure S1, which compares the library replicates based on gene linear densities and read values. As already mentioned in comment 3.5. of this document, we apologize for the confusion. We intended to convey that there is a good correlation between replicates within each library, based on general source metrics (gene linear densities or reads), not between the P and T libraries. In addition to rephrasing the sentence from comment 3.5. for clarity, we have updated the color palette in the figure so that it matches the scheme used throughout the manuscript (green for P, blue for T; previously, different patterns were used to separate replicates within libraries, which we now recognize was not ideal). We have also updated the figure labels accordingly.

3.27. 24) Fig S16; Regarding the extended terminal regions of the E genes, based on the Tn sequence information, the structure of the additional insertion peptides of the final fused protein inserted into each frame should be able to be shown as a common sequence. I believe that by demonstrating this, it would be possible to explain the "extended termini regions" in a more understandable manner. It could illustrate what peptides are added to the N- and C-termini, displaying the conditions of each frame. Could this be organized as supplementary information?

We apologize if we have not understood well what the reviewer suggests. First, the inverted repeats of the transposon are designed to include stop codons for the three possible frames of insertion. Therefore, all the insertions generate protein fusions that depending on the frame can include the addition of only 1 amino acid (W or R or G), 4 amino acids (IKSV) or 7 amino acids (DKVRIIV). This information has been included in the Material and Methods section on page 24. Second, we mention in the manuscript that essential genes have some non-essential short sequences at their beginning and end as noted by different studies. These sequences are different for every gene, tend to be poorly conserved in sequence and length and normally are non structured. These non essential tails are independent of our transposon insertions.

3rd Jul 2025

Manuscript Number: MSB-2025-12901R

Title: Quantitative essentiality in a reduced genome: a functional, regulatory and structural fitness map

Author: Samuel Miravet-Verde

Raul Burgos

Eva Garcia-Ramallo

Marc Weber

Luis Serrano

Dear Luis,

Thank you for submitting your revised manuscript to Molecular Systems Biology. We have now received the enclosed report from the reviewer who agreed to re-assess your work. Since the original Reviewer #2 is unable to re-review the manuscript, we asked Reviewer #3 to also review the your responses to Reviewer #2's main points.

Reviewer #3 thinks that both their concerns and those raised by Reviewer #2 have been adequately addressed. Therefore, I am pleased to inform you that we are able to accept your manuscript, pending the following revision:

1. Please remove the figures from the manuscript file, leaving only the figure legends listed below the References.
2. Please remove the Authors' contribution section from the manuscript file.
3. Please reduce the keyword number to five.
4. Please ensure that all figure callouts appear in sequential order.
5. Add the missing funding information- The Generalitat de Catalunya through the CERCA programme and to the EMBL partnership)-to the submission system.
6. Data availability: The section heading should be renamed to "Data Availability".
7. EV datasets:
 - source file names, titles, legends and manuscript callouts all need to be updated to Dataset EV1-EV12.
 - They should be uploaded individually as Dataset files.
 - Dataset EV1 should be uploaded along with the "Description of dataset EV1" (which should be removed from the Appendix and saved as a separate README file), combined in a single zipped folder. This zipped file should then be uploaded as a standalone Dataset file.
8. APPENDIX:
 - The title page should include "Appendix for [Manuscript Title]" and a Table of Contents listing all included items with corresponding page numbers.
 - The section titled "Description of dataset EV1" should be removed from the Appendix PDF.
9. During our routine figure check, we noticed that the resolution of the blots images in Figure 5, Figure EV1, and Appendix Figure S12 appears to be somewhat low. If available, please provide higher-resolution versions of these images."
10. Please ensure that the manuscript sections are presented in the following order: Title page - Abstract - Keywords - Introduction - Results - Discussion - Methods - Data Availability - Acknowledgements - Disclosure and Competing Interests Statement - References - Figure Legends - Table(s) - Expanded View Figure Legends.

Kind regards,
Jingyi

Jingyi Hou, PhD
Senior Editor
Molecular Systems Biology

*** PLEASE NOTE *** As part of the EMBO Press transparent editorial process initiative (see our Editorial at <https://dx.doi.org/10.1038/msb.2010.72> , Molecular Systems Biology will publish online a Review Process File to accompany accepted manuscripts. When preparing your letter of response, please be aware that in the event of acceptance, your cover letter/point-by-point document will be included as part of this File, which will be available to the scientific community. More information about this initiative is available in our Instructions to Authors. If you have any questions about this initiative, please contact the editorial office (msb@embo.org).

Reviewer #3:

All of my concern was cleared by the authors' revision and their explanation.
I think this manuscript is ready for the publication.

Thank you for submitting your revised manuscript to Molecular Systems Biology. We have now received the enclosed report from the reviewer who agreed to re-assess your work. Since the original Reviewer #2 is unable to re-review the manuscript, we asked Reviewer #3 to also review the your responses to Reviewer #2's main points.

Reviewer #3 thinks that both their concerns and those raised by Reviewer #2 have been adequately addressed.

We thank the editor for coordinating the re-review process and the reviewers for their careful assessment of our revised manuscript. We are pleased to know that our responses have adequately addressed the concerns raised, and we appreciate the constructive feedback that helped us improve the quality and clarity of our work.

Therefore, I am pleased to inform you that we are able to accept your manuscript, pending the following revision:

1. Please remove the figures from the manuscript file, leaving only the figure legends listed below the References.

Figures have been removed leaving only the figure legends, under the section 'FIGURE LEGENDS', right after the REFERENCES section.

2. Please remove the Authors' contribution section from the manuscript file.

The authors' contribution section has been removed.

3. Please reduce the keyword number to five.

We have reduced the keywords to five: essentiality, transposon sequencing, high-resolution, regulatory elements, Mycoplasma

4. Please ensure that all figure callouts appear in sequential order.

We have checked that the main, appendix, and EV figures are listed in the correct order. We have also carefully verified the same for the tables and supplementary datasets.

5. Add the missing funding information- The Generalitat de Catalunya through the CERCA programme and to the EMBL partnership)-to the submission system.

We have added the missing funding bodies in the submission system.

6. Data availability: The section heading should be renamed to "Data Availability".

The section has been renamed.

7. EV datasets:

- source file names, titles, legends and manuscript callouts all need to be updated to Dataset EV1-EV12.

We have confirmed that all datasets are properly referred to as Dataset EVX and are presented in the correct order.

- They should be uploaded individually as Dataset files.

All datasets have been uploaded independently.

- Dataset EV1 should be uploaded along with the "Description of dataset EV1" (which should be removed from the Appendix and saved as a separate README file), combined in a single zipped folder. This zipped file should then be uploaded as a standalone Dataset file.

Dataset EV1 has been uploaded as a zip file including the original tsv file and a README with the description of the information included.

8. APPENDIX:

- The title page should include "Appendix for [Manuscript Title]" and a Table of Contents listing all included items with corresponding page numbers.

We have updated the title of the document and added a Table of Contents listing the page of all the included Appendix Figures.

- The section titled "Description of dataset EV1" should be removed from the Appendix PDF.

The description of Dataset EV1 has been removed from the Appendix and is now provided as a README file, packaged together with the original dataset in a single compressed ZIP file.

9. During our routine figure check, we noticed that the resolution of the blots images in Figure 5, Figure EV1, and Appendix Figure S12 appears to be somewhat low. If available, please provide higher-resolution versions of these images."

We have updated the blot images in Figures 5, EV1 and S12. This last figure has also been reorganized Figure S12 to optimize their quality. In addition, we have compiled a new Appendix file to ensure all the figures preserved the maximum quality.

10. Please ensure that the manuscript sections are presented in the following order: Title page - Abstract - Keywords - Introduction - Results - Discussion - Methods - Data Availability - Acknowledgements - Disclosure and Competing Interests Statement - References - Figure Legends - Table(s) - Expanded View Figure Legends.

We have reviewed the section order and titles. However, we were unsure where to place the Expanded View Table Legends for Tables EV1–EV5, so we have listed their legends between the Figure Legends and the Expanded View Figure Legends.

12th Jul 2025

Manuscript number: MSB-2025-12901RR

Title: Quantitative essentiality in a reduced genome: a functional, regulatory and structural fitness map

Dear Luis,

Thank you again for sending us your revised manuscript. We are now satisfied with the modifications made and I am pleased to inform you that your paper has been accepted for publication.

Kind regards,
Jingyi

Jingyi Hou, PhD
Senior Editor
Molecular Systems Biology
